# Uncertainty estimation with prediction-error circuits

Loreen Hertäg ®[1] ✉, Katharina A. Wilmes ®[2] & Claudia Clopath ®[3]

Neural circuits continuously integrate noisy sensory stimuli with predictions that often do not perfectly match, requiring the brain to combine these conflicting feedforward and feedback inputs according to their uncertainties. However, how the brain tracks both stimulus and prediction uncertainty remains unclear. Here, we show that a hierarchical prediction-error network can estimate both the sensory and prediction uncertainty with positive and negative prediction-error neurons. Consistent with prior hypotheses, we demonstrate that neural circuits rely more on predictions when sensory inputs are noisy and the environment is stable. By perturbing inhibitory interneurons within the prediction-error circuit, we reveal their role in uncertainty estimation and input weighting. Finally, we link our model to biased perception, showing how stimulus and prediction uncertainty contribute to the contraction bias.

To survive in an ever-changing environment, animals must flexibly adapt their behavior based on previously encoded and novel information. This adaptation is reflected in the information processing of neural networks underlying context-dependent behavior. For instance, when walking down an unfamiliar staircase in a well-lit basement, your brain may rely almost entirely on feedforward (bottom-up) sensory input (Fig. 1A, left), gradually forming a model of the step sizes. As the step sizes become more predictable, the brain can increasingly rely on this model. However, if the step sizes suddenly change, it will need to revert to the sensory input for guidance. Later, when walking down the same staircase and the lights suddenly turn off, your brain may rely entirely on feedback (top-down) signals derived from the staircase model you previously formed (Fig. 1A, middle), as the sensory information becomes too noisy to trust. But how do neural networks switch between a feedforward-dominated and a feedback-dominated processing mode in an ever-changing environment? For instance, if you hike down an unexplored mountain in very foggy conditions, your brain receives unreliable visual information. In addition, it can only draw on a shaky prediction about what to expect (Fig. 1A, right).

A common hypothesis is that the brain weights different inputs according to their reliabilities. A prominent example of this hypothesis is Bayesian multisensory integration (for example, ref. 1). According to this theory, neural networks represent information from multiple

modalities by a linear combination of the uncertainty-weighted single-modality estimates. Multisensory integration is supported by several observations showing that animals can combine information from different modalities in a fashion that minimizes the variance of the final estimate[2–8]. Here, we propose that the same concepts could be employed for the weighting of sensory inputs and predictions thereof[4,9]. A central point in the weighting of inputs is the estimation of their variances as a measure of uncertainty. However, how the variance of both the sensory input and the prediction can be computed on the circuit level is not resolved yet.

We hypothesized that prediction-error (PE) neurons provide the basis for the neural computation of variances. PEs are an integral part of the theory of predictive processing which states that the brain constantly compares incoming sensory information with predictions. If those predictions are wrong, the resulting PEs allow the network to revise the model of the world, thereby ensuring that the predictions become more accurate[10]. Experimental evidence suggests that these PEs may be represented in the activity of distinct groups of neurons, termed PE neurons[11–14]. Moreover, these neurons may come in two types when excitatory neurons exhibit near-zero, spontaneous firing rates[10,15]: negative PE (nPE) neurons only increase their activity when the sensory input is weaker than the prediction, while positive PE (pPE) neurons only increase their activity when the sensory input is stronger than the

[1]Modeling of Cognitive Processes, TU Berlin, Berlin, Germany. [2]Department of Physiology, University of Bern, Bern, Switzerland. [3]Bioengineering Department, Imperial College London, London, UK. ✉e-mail: loreen.hertag@tu-berlin.de

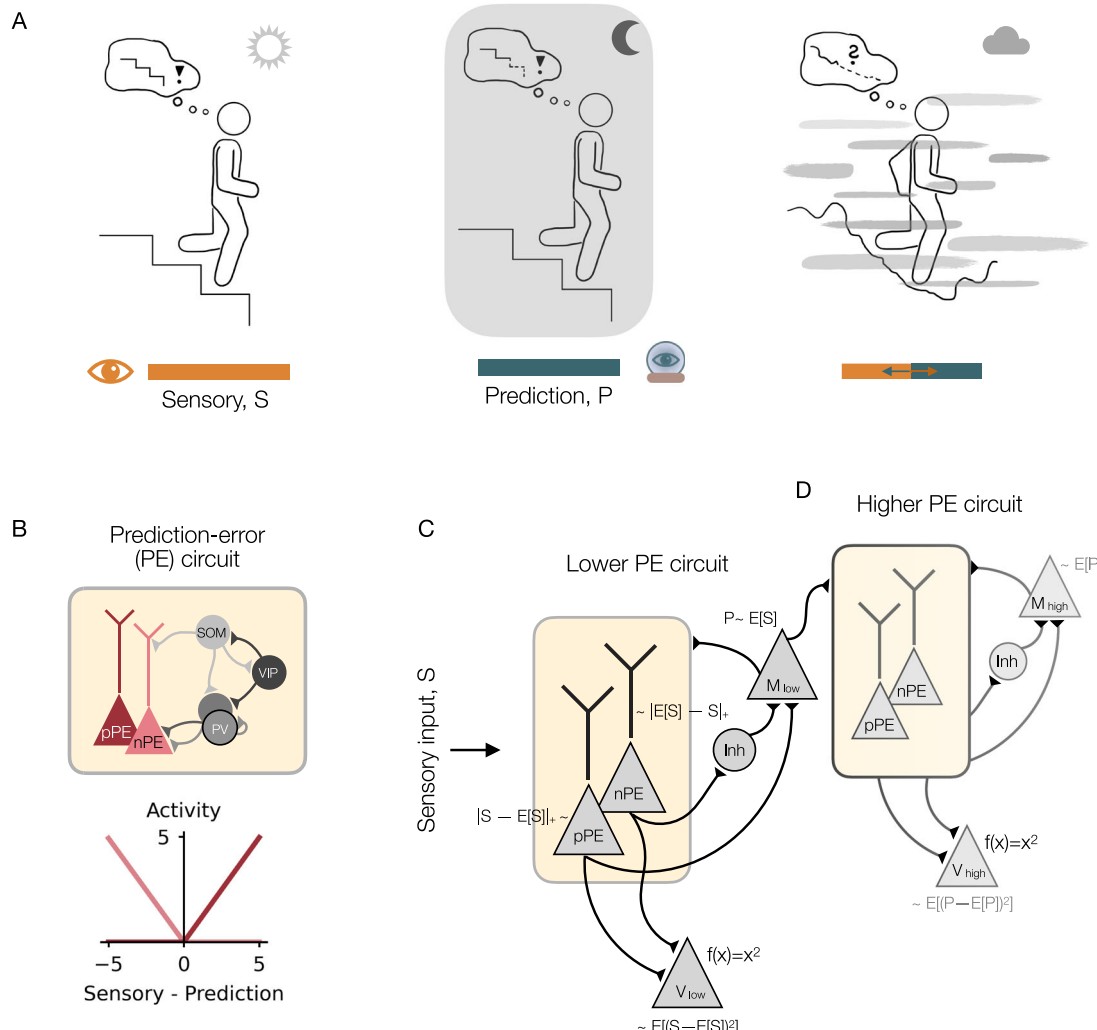

**Fig. 1 | Neural network model to track both the uncertainty of sensory inputs and predictions. A** Example illustration for context-dependent integration of information. Left: when walking down an unfamiliar staircase that is visible, the brain might rely solely on external sensory information. Middle: when walking down the same stairs without visual information, the brain might rely on predictions formed by previous experience. Right: when climbing down an unexplored mountain in foggy conditions, the brain might need to integrate sensory information and predictions simultaneously. **B** Top: illustration of a prediction-error (PE) circuit with both negative and positive PE (nPE/pPE) neurons that receive inhibition from three different inhibitory interneuron types: parvalbumin-expressing (PV), somatostatin-expressing (SOM), and vasoactive intestinal peptide-expressing (VIP) interneurons. Local excitatory connections are not shown for clarity. Bottom: Responses of an nPE and pPE neuron. The nPE neuron only increases its activity relative to a baseline when the sensory input is weaker than predicted, while the pPE neuron only increases its activity relative to a baseline when the sensory input is stronger than predicted. **C** Illustration of the network model that estimates the mean and variance of the external sensory stimuli. The core of this network model is the PE circuit shown in (**B**). The lower-level V neuron encodes the variance, while the lower-level M neuron encodes the mean of the sensory input. **D** Same as in (**C**) but the feedforward input is the activity of the lower-level M neuron.

prediction. Indeed, it has been shown that excitatory neurons in rodent primary sensory areas can encode negative or positive PEs[14,16–18].

Here, we show that the unique response patterns of nPE and pPE neurons may provide the backbone for computing both the mean and the variance of sensory stimuli. Furthermore, we suggest a network model with a hierarchy of PE circuits to estimate the variance of the prediction, in addition to the variance of the sensory inputs. We show that in line with the ideas of multisensory integration, predictions are weighted more strongly than the sensory stimuli when the environment is stable (that is, predictable) and the sensory inputs are noisy. Moreover, we find that predictions are taken into account more at the beginning of a new trial than at the end, especially when the new sensory stimulus is reliable. In addition, we unravel the mechanisms underlying a neuromodulator-induced shift in the weighting of sensory inputs and predictions. In our model, these neuromodulators activate groups of inhibitory neurons such as parvalbumin-expressing

(PV), somatostatin-expressing (SOM), and vasoactive intestinal peptide-expressing (VIP) interneurons[19–24]. These interneurons have been suggested to establish a multi-pathway balance of excitation and inhibition that is the basis for nPE and pPE neurons[25,26]. By perturbing this balance, the PE neurons change their baseline firing rate and gain, leading to a biased variance estimation. Finally, we show that this weighting can be understood as a neural manifestation of the contraction bias, that is, the magnitude of the represented sensory input is biased towards the mean of the past stimuli experienced[27–32].

## Results

### A circuit model for uncertainty estimation

We hypothesize that the distinct response patterns of negative and positive prediction-error (nPE/pPE) neurons can act as a backbone for estimating the mean and the variance of sensory stimuli. An nPE neuron only increases its activity relative to a baseline when the sensory

input is weaker than predicted, while a pPE neuron only increases its activity relative to a baseline when the sensory input is stronger than predicted. Moreover, both nPE and pPE neurons remain at their baseline activities when the sensory input is fully predicted (Fig. 1B).

To test our hypothesis, we study a rate-based mean-field network: the core network contains two excitatory neurons, two inhibitory PV interneurons, one inhibitory SOM, and one inhibitory VIP interneuron (Fig. 1B, also see Supplementary Fig. 1). While the excitatory neurons are simulated as two coupled point compartments to emulate the soma and dendrites of elongated pyramidal cells, respectively, all inhibitory cell types were modeled as point neurons. In line with experimental findings[23], we assume that the PV neurons target the somatic compartment, while the SOM neuron targets the dendritic compartment of the excitatory cells. Moreover, the SOM neuron inhibits both the PV and VIP neurons, while the VIP neuron inhibits both the PV and the SOM neurons[23] (Fig. 1B, also see Supplementary Fig. 1). In addition, all neurons receive local connections from the excitatory neurons (Supplementary Fig. 1).

We chose the connection strengths in line with our previous work on prediction-error neurons (for example, see refs. 25,26). In that work, we showed that response patterns of excitatory cells resemble those of PE neurons when a number of excitatory (E) and inhibitory (I) pathways onto the pyramidal cells were balanced. This multi-pathway E/I balance results in an E/I balance of the inputs to excitatory neurons when the stimulus is perfectly predicted. Depending on the network connectivity, for some excitatory cells, this input E/I balance was maintained for over-predicted stimuli (sensory input < prediction), but temporarily shifted toward excitation for under-predicted stimuli (sensory input > prediction). In contrast, other excitatory cells exhibited the opposite pattern, with responses for over- and under-predicted stimuli reversed. The former group corresponds to pPE neurons, while the latter represents nPE neurons.

The multi-pathway E/I balance required for PE neurons to emerge was established through the different interneurons. These interneurons provide compartment-specific inhibition to balance the feedforward sensory inputs and the feedback predictions, respectively (for a more detailed discussion on the role of these interneurons in PE circuits, please see Supplementary Discussion). In the present work, we use a PE circuit in which the soma of the excitatory cells, the SOM neuron, and one of the PV neurons receive the feedforward sensory input, while the other cells/compartments receive the prediction thereof. This is in line with experimental work showing that feedback connections hypothesized to carry information about expectations or predictions[33–35] target the apical dendrites of pyramidal cells[34] and interneurons located in superficial layers of the cortex (for example, ref. 23).

We reasoned that if a prediction of a stimulus is the mean of the previously experienced stimuli, it can be modeled through a perfect integrator (here denoted memory neuron) that receives connections from the PE neurons (Fig. 1C). More precisely, following Keller and Mrsic-Flogel[10], we assume that the pPE neuron excites the memory neuron, while the nPE neuron inhibits this neuron (for instance, through lateral inhibition, here not modeled explicitly). If the activity of the memory neuron is below the sensory input, the pPE neuron is active while the nPE neuron is silent (Supplementary Fig. 2). Hence, the memory neuron receives more excitation. If the activity of the memory neuron is above the sensory input, the nPE neuron is active while the pPE neuron is silent (Supplementary Fig. 2). As a consequence, the memory neuron receives more inhibition. When the memory neuron is roughly at the mean of the sensory inputs, occasionally being below or above, the effect of nPE and pPE neuron cancels. Hence, the PE neurons ensure that the memory neuron's activity does not drift too far from the mean (see Box 1).

The memory neuron in our network projects back to the PE neurons it receives inputs from. We, therefore, call the input from the memory neuron to all other neurons in the PE circuit feedback input. While we consider the activity of the memory neuron as a prediction of the current sensory input, it could also be interpreted as a prior of the sensory mean at the next time step.

If the prediction equals the mean of the sensory stimulus, the activity of the nPE and pPE neurons encode the deviation from the mean. Thus, the squared sum of nPE and pPE neuron activity represents the variance of the feedforward input (provided that the PE neurons are silent without sensory stimulation). We, therefore, simulate a downstream neuron (termed V neuron), modeled as a leaky integrator with a quadratic activation function, that receives excitatory synapses from the PE neurons (see the lower-level subnetwork in Fig. 1C, the higher-level circuit is described later).

## Estimating the mean and variance of sensory stimuli with prediction-error neurons

To show that this network can indeed represent the mean and the variance in the respective neurons, we stimulate it with a sequence of step-wise constant inputs drawn from a uniform distribution (Fig. 2A).

---

## BOX 1

Following the definition of nPE and pPE neurons, their idealized activity, $r_{nE}$ and $r_{pE}$, can be written as

$$\mathbf{r}_{nE} = \left[\mathbf{r}_M - \mathbf{S}\right]_+$$
$$\mathbf{r}_{pE} = \left[\mathbf{S} - \mathbf{r}_M\right]_+$$

with $S$ denoting the time-dependent feedforward input and $r_M$ the activity of the M neuron that can be summarized as

$$\tau_M \cdot \frac{dr_M}{dt} = r_{pE} - r_{nE}.$$

Inserting the idealized activity of nPE and pPE neurons and solving the differential equation yields

$$r_M = \frac{1}{\tau_M} \int_0^t e^{-(t-x)/\tau_M} \cdot S(x) \, dx$$

for zero activity at time $t = 0$. In the steady state (that is, $t \to \infty$), this is the exponential moving average of the feedforward input. Similarly, the activity of the V neuron can be described by

$$\tau_V \cdot \frac{dr_V}{dt} = -r_V + (r_{pE} + r_{nE})^2 = -r_V + (r_M - S)^2.$$

Solving the differential equation after replacing the activity of nPE and pPE neurons with their respective definitions yields

$$r_V = \frac{1}{\tau_V} \int_0^t e^{-(t-x)/\tau_V} \cdot \left[r_M(x) - s_{FF}(x)\right]^2 dx.$$

In the limit of $t \to \infty$, $r_V$ approaches the variance of the feedforward input.

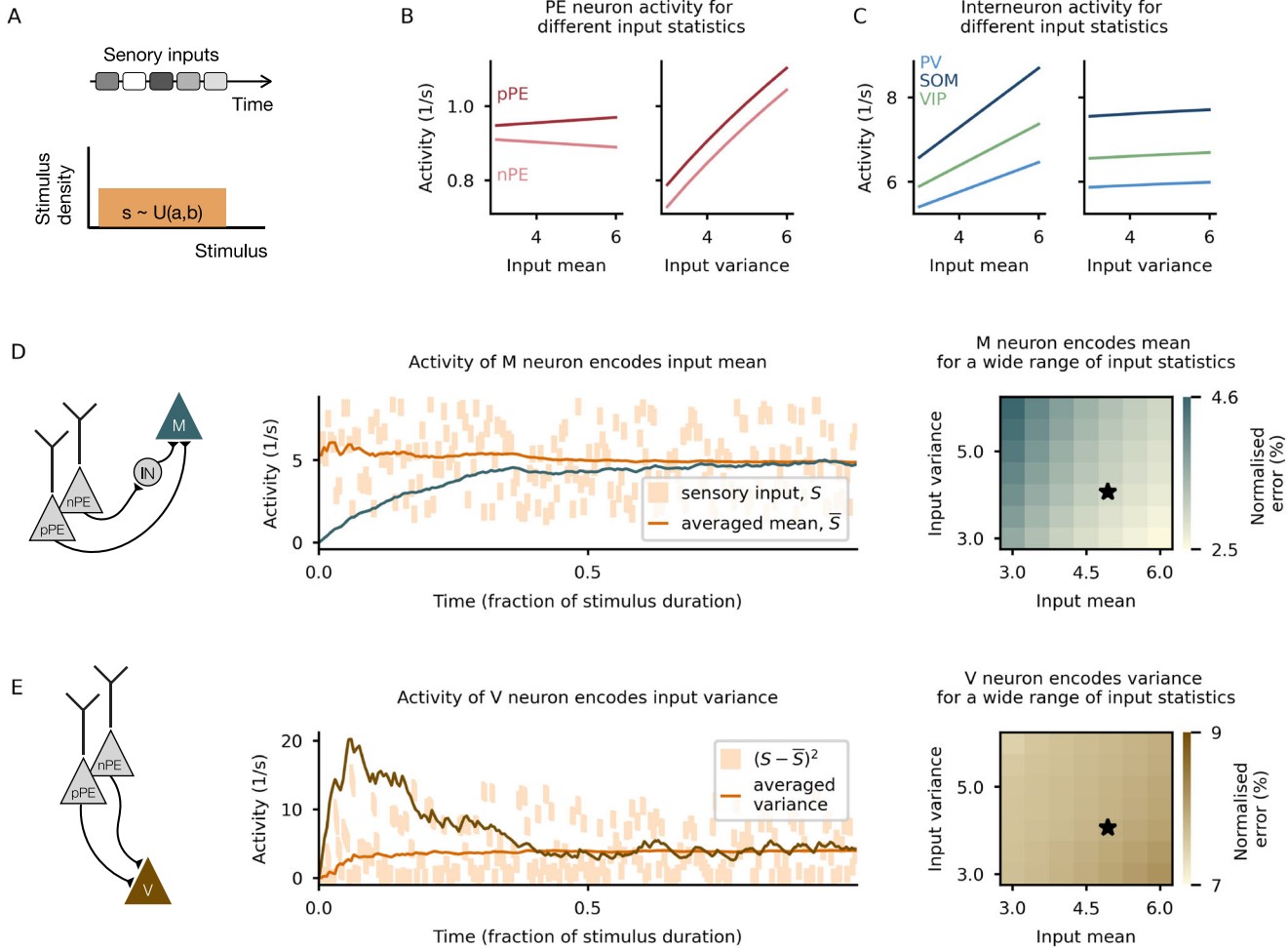

**Fig. 2 | Prediction-error neurons as the basis for estimating mean and variance of sensory stimuli. A** Illustration of the inputs with which the network (Fig. 1C) is stimulated. Network is exposed to a sequence of constant stimuli drawn from a uniform distribution. The gray shaded boxes symbolize different values from the distribution. **B** PE neuron activity hardly changes with stimulus strength (left) but strongly increases with stimulus variability (right). **C** Interneuron activity strongly changes with stimulus strength (left) but hardly changes with stimulus variability (right). **D** M neuron correctly encodes the mean of the sensory stimuli. Left: Illustration of the input synapses onto the M neuron. Middle: Activity of the M neuron

(dark green line) over time for one example distribution (black star in right panel). Right: Normalized absolute difference between the averaged mean and the activity of the M neuron in the steady state for different parametrizations of the stimulus distribution. **E** V neuron correctly encodes the variance of the sensory stimuli. Left: Illustration of the input synapses onto the V neuron. Middle: Activity of the V neuron (dark brown line) over time for one example distribution (black star in right panel). Right: Normalized absolute difference between the averaged variance and the activity of the V neuron in the steady state for different parametrizations of the stimulus distribution.

We, hence, assume that the sensory stimulus varies over time. In line with the distinct response patterns for nPE and pPE neurons, these neurons change only slightly with increasing stimulus mean but increase strongly with input variance (Fig. 2B). In contrast, the three interneurons strongly increase with stimulus mean and only moderately increase with stimulus variance (Fig. 2C). The activity of the memory neuron gradually approaches the mean of the sensory inputs (Fig. 2D, middle), while the activity of the V neuron approaches the variance of those inputs (Fig. 2E, middle). We show that this holds for a wide range of input statistics (Fig. 2D, E, right) and input distributions (Supplementary Fig. 3). Small deviations from the true mean occur mainly for large input variances, while the estimated variance is fairly independent of the input statistics tested. Moreover, using a continuously changing signal instead of a piecewise constant stimulus yields similar results, where small deviations can be attributed to the PE neurons not reaching their steady state (Supplementary Fig. 4).

While the results do not strongly depend on the stimulus statistics and distribution, they are affected by the baseline activities of the PE neurons that were assumed to be zero in our network, in line with the low baseline firing rates reported for neurons in primary visual cortex

of rodents[36,37]. When the baseline rate of the nPE neuron is increased, the memory neuron underestimates the mean of the sensory input (Supplementary Fig. 5A). In contrast, when the baseline rate of the pPE neuron is increased, the memory neuron overestimates the mean of the sensory input (Supplementary Fig. 5A). However, increasing the baseline for both PE neurons by the same amount does not affect the estimation of the stimulus mean (Supplementary Fig. 5A). In contrast, a non-zero baseline in any of the PE neurons yields an overestimation of the stimulus variance (Supplementary Fig. 5B). This suggests inhibitory interneurons must cancel the baseline activity to ensure an unbiased uncertainty estimation in networks with high-baseline PE neurons. While the baseline activity of PE neurons can bias the estimation of mean and variance, other neuron properties and network connection strengths play a less pivotal role (see Supplementary Fig. 6, discussed in the Supplementary Discussion).

We verified the main results in a heterogeneous network, where each neuron type of the PE circuit was represented by a distinct population of neurons, and the synaptic connection strengths from each PE neuron onto the M and V neuron are different (see SI Methods, Supplementary Fig. 7A). As before, the network can correctly estimate

the mean and the variance of the sensory stimuli (Supplementary Fig. 7B). Furthermore, we show that the errors with which the M and V neurons encode the stimulus statistics are independent of uncorrelated modulations of the connection strengths (Supplementary Fig. 7C) and the sparsity of the network (Supplementary Fig. 7E). When all connection strengths are collectively shifted to higher values, the error increases for the variance neuron, while it remains unaffected for the memory neuron.

While our mean-field network was designed to track the mean and the variance of stimuli that vary in time, we reasoned that the same principles apply to stimuli that vary across space. To show that, we simulated a population network that consists of unconnected replicates of the mean-field network described above (Supplementary Fig. 8A). Each mean-field network receives a short, constant input from a different part of the receptive field. If the connection strengths from the PE neurons to the M and V neurons are adjusted accordingly (see Methods), the network correctly estimates the stimulus average and spatial uncertainty (Supplementary Fig. 8B, C).

In summary, nPE and pPE neurons can serve as a basis to estimate the mean and the variance of sensory stimuli which vary over time and space.

## Estimating the uncertainty of both the sensory input and the prediction requires a hierarchy of PE circuits

Following the ideas of Bayesian multisensory integration, the weighting of sensory stimuli and predictions would require knowledge about their uncertainties. As we have shown in the previous section, the variance of the sensory stimulus can be estimated using PE neurons. We hypothesize that the same principles apply to computing the variance of the prediction. To show this, we augment the network with a higher PE circuit that receives feedforward synapses from the memory (M) neuron of the lower PE circuit (Fig. 1D, and a more detailed network diagram in Supplementary Fig. 1). Both subnetworks are identical except for the M neuron in the higher PE circuit which is modeled with slower dynamics than the one in the lower PE circuit.

To evaluate the network's ability to accurately estimate variances, we conducted tests using a sequence of inputs varying on two different timescales. Specifically, in each trial, the network receives a stimulus consisting of $N_{in}$ constant values. Each value is drawn from a normal distribution and presented over $N_{step}$ consecutive time steps. The variance of this normal distribution indicates the level of stimulus noise. Additionally, to simulate changes in the environment, the stimulus mean is re-drawn from a uniform distribution (Fig. 3A) after $N_{in} \cdot N_{step}$ time steps (that is, in each trial). This setup aligns with a change detection task and has been previously studied (for e.g., see ref. 38,39).

Following the formalism of multisensory integration (for e.g., see ref. 40), we assume that the network's output is a weighted sum of the feedforward sensory input and the feedback prediction. The weights assigned to each input stream are functions of the uncertainties, that is, the activities of the V neurons. The sensory weight captures how much the network relies on the sensory input (Fig. 2B). For the sake of simplicity, we assume that the weighted output is encoded in a separate class of neurons not explicitly modeled here and only compute the sensory weight arithmetically.

To test our network, we first consider two limit cases. In the first limit case, we show a low-variance stimulus that differs in each trial (low stimulus uncertainty, high trial-to-trial uncertainty, see Fig. 3C, left). According to the theory, the network should follow the sensory inputs closely and ignore the predictions. When we arithmetically calculate the weighted output (Fig. 3C, middle) and the sensory weight (Fig. 3C, right), the network shows a clear preference for the sensory input, indicating that the network estimated the uncertainty of the sensory input to be lower than that of the prediction. In the second

limit case, we show a high-variance stimulus, the mean of which does not change from trial to trial (high stimulus uncertainty, low trial-to-trial uncertainty, see Fig. 3D, left). According to the theory, the network should downscale the sensory feedforward input and weight the prediction more strongly. As expected, the weighted output of the network shows a clear tendency to the mean of the stimuli (Fig. 3D, middle), also reflected in the low sensory weight (Fig. 3D, right). Hence, the network estimated the uncertainty of the sensory input to be higher than that of the prediction.

To validate the network responses fully, we systematically varied the trial and stimulus variability independently. If both variances are similar, the sensory weight approaches 0.5, reflecting the equal contribution of the sensory input and the prediction to the weighted output. Only if both variances are zero, the network represents the sensory input perfectly. In line with the limit case examples above, if the stimulus variance is larger than the trial variance, the network weights the prediction more strongly than the sensory input (Fig. 3E). Because the network dynamically estimates the sensory and prediction uncertainty, the sensory weight changes when the input statistics shifts (Supplementary Fig. 9).

Inspecting closely the dynamics of our network, we noticed that the prediction is typically weighted higher at the beginning of a new trial than in the steady state. This is particularly pronounced in a sensory-driven input regime (see Fig. 3C), and further confirmed in simulations in which the trial duration was shortened from 5s to 1s (Fig. 3F). This observation highlights that the approach is suboptimal immediately after a change point, as predictions based on previous sensory inputs become incorrect following an environmental change. In such cases, the system should promptly adapt to the new sensory input. Alternative approaches that detect potential change points and allow the system to prioritize sensory input after a change are possible and have been discussed (for e.g., see ref. 39). However, identifying change points can be challenging, especially when the level of sensory noise varies. While it is common to focus on changes in the environment (i.e., $\mu$), changes in sensory noise levels (i.e., $\sigma$) can also occur. In this work, we considered a spectrum of scenarios encompassing both environmental changes and fluctuations in noise levels. Although the weighting strategy used here is less accurate immediately after a change point, it performs well in steady-state conditions (Supplementary Fig. 10A). Furthermore, the reliability-weighted input approach effectively handles scenarios where sensory noise undergoes abrupt changes (Supplementary Fig. 10A). In contrast, simpler methods designed to minimize inaccuracies after a change point may struggle with high-noise scenarios. For example, while approaches based solely on sensory input variance (Supplementary Fig. 10B) or the disparity between the lower and the higher memory neuron (Supplementary Fig. 10C) improve the output estimate in low-noise conditions, they struggle in high-noise conditions.

It has been hypothesized, that some symptoms in psychiatric disorders like autism and schizophrenia can be ascribed to a pathological weighting of sensory inputs and predictions[9]. We thus wondered which network properties might bias the estimation of the variances, and, consequently, the weighting of different input streams. We identified the effective timescales at which the memory neurons incorporate new information as a decisive factor in the integration of inputs. To show this, we varied the weights from the PE neurons onto the lower-level memory neurons. If the weights are too small (the memory neuron updates too slowly), the system relies too much on feedback predictions. In contrast, if the weights are too large (the memory neuron updates too fast), the system relies too much on the feedforward sensory information (Supplementary Fig. 11A). However, scaling the respective weights onto both the lower-level and the higher-level memory neuron by the same amount only has a minor effect on the sensory weight (Supplementary Fig. 6B). While the speeds

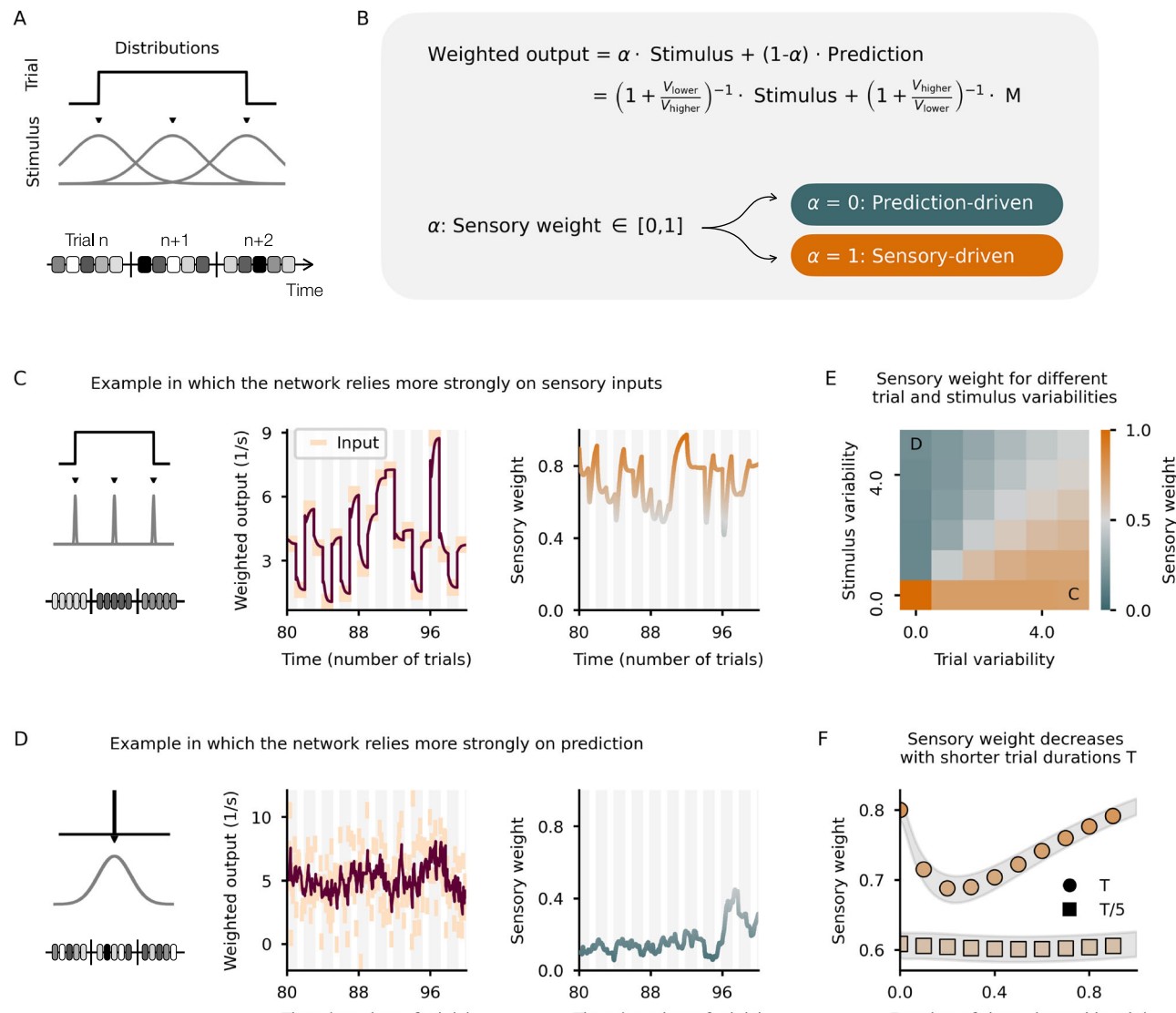

**Fig. 3 | Estimating the uncertainty of both the sensory input and the prediction.**
**A** Illustration of the stimulation protocol. The network is exposed to a sequence of stimuli (one stimulus per trial). To account for stimulus variability, each stimulus is represented by 10 stimulus values drawn from a normal distribution. To account for the volatility of the environment, in each trial, the stimulus mean is drawn from a uniform distribution (denoted trial-to-trial variability). **B** Illustration of how the weighted output is calculated. The sensory weight α lies between zero (system relies perfectly on prediction) and one (system relies solely on the sensory input). **C** Limit case example in which the stimulus variability is zero but the trial-to-trial variability is high. Left: Illustration of the stimulation protocol. Middle: Weighted output follows closely the sensory stimuli. Right: Sensory weight (function of the

variances, see B) close to 1, indicating that the network ignores the prediction. Input statistics shown in (**E**). **D** Limit case example in which the stimulus variability is high but the trial-to-trial variability is zero. Left: Illustration of the stimulation protocol. Middle: Weighted output pushed towards the mean of the sensory stimuli. Right: Sensory weight close to zero, indicating that the network ignores the sensory stimuli. Input statistics shown in (**E**). **E** Sensory weight for different input statistics. Predictions are weighted more strongly when the stimulus variability is larger than the trial-to-trial variability. **F** Sensory weight, averaged over many trials, for two different trial durations (circles: trial duration = 5s, squares: trial duration = 1s). Gray shading denotes the SEM. Predictions are weighted more strongly at the beginning of a new trial.

at which the activity of the memory neurons evolve influence the weighting of inputs, the precise activation function of the variance neurons is less pivotal. When we replaced the quadratic activation function with a linear, rectified function, the V neurons did not encode the variance but the average absolute deviation of the sensory stimuli. However, the sensory weight is only slightly shifted to larger values for low trial/high stimulus variability (Supplementary Fig. 11B).

We have shown that the baseline activity of PE neurons can affect the ability of the M and V neurons to encode the mean and the variance of the feedforward input, respectively. In the full network, these biases manifest in a sensory weight that is slightly pushed towards 0.5 (Supplementary Fig. 5C). That is, in an initially sensory-driven regime, the

dependence on the sensory inputs is slightly weakened. In contrast, in an initially prediction-driven regime, the dependence on the sensory inputs is slightly strengthened. Similarly, while other properties like the connectivity between the PE neurons and the M/V neurons can affect the estimation of the mean and the variance, the sensory weight is only slightly affected if the changes occur in both the lower- and the higher-level circuit (Supplementary Fig. 6).

In summary, we show that the variances of both the sensory inputs and predictions thereof can be dynamically computed in networks comprising a lower and higher PE circuit. In such a network, predictions are given more weight at the beginning of a new stimulus, and if the sensory inputs are noisy while the environment is stable.

## Biasing the weighting of sensory inputs and predictions by neuromodulators

The brain's flexibility and adaptability are supported by a plethora of neuromodulators that influence the activity of neurons in a variety of ways[41]. A prominent target of neuromodulatory inputs is inhibitory neurons[42–44]. Moreover, distinct interneuron types are differently (in-) activated by those neuromodulators[43–45]. Given that the interneurons in our network play a crucial role in establishing the PE neurons that, in turn, are the backbone for computing the uncertainties, we wondered if and how the weighting of sensory inputs and predictions may be biased when neuromodulators activate distinct interneuron types.

To this end, we modeled the presence of a neuromodulator by injecting an additional excitatory input into an interneuron type (while a neuromodulator can also suppress neuronal activity, we focus on the more common excitatory effects that have been described). Given that the interneurons are embedded in a network and establish an E/I balance in the PE neurons through multiple pathways, we reasoned that not only the interneuron type but also the connections it receives/ makes determine the effect of a neuromodulator. Hence, testing only one instantiation of our network may yield effects that do not generalize to other parameterizations of the network. To avoid that, we tested three different mean-field networks derived in[26]. These networks differ in the distribution of sensory inputs and predictions onto the interneurons, and, hence, the underlying connectivity. They cover a broad range of possible PE circuits. The only commonality across those networks is that they exhibit an E/I balance of excitatory and inhibitory pathways onto the PE neurons.

Across the different mean-field networks tested, increasing the activity of PV neurons biases the network's output toward predictions (Fig. 4A, B, light blue line). In contrast, increasing VIP activity forces the networks to weigh both inputs more equally. As a consequence, predictions are overrated in a sensory-driven input regime (Fig. 4A, green line), and, sensory inputs are overrated in a prediction-driven input regime (Fig. 4B, green line). Increasing SOM neuron activity, while qualitatively similar to increasing VIP neuron activity, depends on the mean-field network tested and the strength of activation (Fig. 4A, B, dark blue line).

Neuromodulators are most likely increasing the activity of more than one interneuron type. To account for the co-activation of interneurons, we injected an excitatory input into two interneuron types at the same time and varied the strength with which each interneuron was modulated. If SOM and VIP neurons are equally stimulated, the weighting of sensory inputs and predictions remains largely unaffected (Fig. 4A, B, Supplementary Figs. 15 and 16, dashed beige line), suggesting that the individual effects cancel out. If PV neurons are the major target of a neuromodulator, the network is still biased toward predictions (Supplementary Figs. 15 and 16). While some results depend not only on the interneuron type targeted but also the connections it makes/receives, as well as the strength of the neuromodulation, there are consistent effects across the mean-field networks tested (illustrated in Fig. 4C): neuromodulators increasing the PV neuron activity bias the weighting towards predictions. If, however, VIP neurons are the major target of a neuromodulator, the sensory weight is pushed towards 0.5. This effect is reversed when SOM neurons are equally targeted (see Discussion for more details and a comparison with experiments).

What are the network mechanisms underlying these observations? The sensory weight is a function of the lower and higher variance (V) neuron activity. Hence, any changes to the sensory weight result from changes to the neurons encoding the variances. In our network, the V neurons only receive excitatory synapses from PE neurons. As a consequence, any changes in the sensory weights upon activation of interneurons must be due to changes in the PE neurons. This suggests that to understand the effect of neuromodulators on sensory weight, we need to unravel the effect of interneuron activation on PE neurons. Increasing interneuron activity leads to changes in the baseline and gain of PE neurons (Supplementary Fig. 12, see Methods). In all three networks tested, activating PV neurons decreases both the baseline and gain of the PE neurons, leading to a decrease in the estimated variance (Fig. 4D and Supplementary Fig. 13). Stimulating the SOM or VIP neuron decreases the gain in either nPE or pPE neuron. However, the baseline of those neurons can either decrease or increase depending on the connectivity with other neurons in the network. The summed effect over nPE and pPE neurons (Supplementary Fig. 13) suggests that whether the activity of the V neuron increases or decreases depends on the input statistics: for low-mean stimuli, the elevated baseline activity dominates the changes in the variance, while for high-mean stimuli the changes in the gain dominate. Furthermore, we note that the presence of a neuromodulator can also affect the predictive state of the network. That is, the M neuron activity can also be modulated by changes in the baseline and gain of the PE neurons.

Altogether, we show that neuromodulators increasing the activity of interneurons bias the weighting of sensory inputs and predictions by changing the gain and baseline of PE neurons. Whether the sensory weight increases or decreases depends not only on the interneuron it targets but also on the network it is embedded in and the input regime.

## Explaining the contraction bias with the weighting of sensory inputs and predictions

We hypothesized that the weighted integration of sensory inputs and predictions manifests in everyday behavior as contraction bias. This phenomenon describes the tendency to overestimate sensory stimuli from the lower end of a distribution and underestimate those from the upper end, reflecting a bias toward the mean observed across species and modalities[27–32].

First, we investigated whether the network's output can be interpreted as a neuronal manifestation of the contraction bias (see Methods for an illustrative analysis). We define contraction bias as the trial-averaged difference between the weighted output and the sensory stimulus. The bias is positive for stimuli below the mean of the input distribution and negative for stimuli above the mean (Fig. 5A), consistent with a bias toward the mean. We quantify the bias using the slope of the linear fit between bias and trial stimulus; a larger absolute slope indicates a larger bias.

What network factors contribute to the neuronal contraction bias? When we increase the stimulus uncertainty, the bias increases as well (Fig. 5B). In contrast, when we increase the trial-to-trial uncertainty, the bias decreases (Fig. 5B). To further disentangle the different sources of the bias, we simulated a network without stimulus uncertainty (variance set to zero) under two trial-to-trial variances (environmental volatility). In this case, the emerging contraction bias is independent of the volatility of the environment (Fig. 5C). We show mathematically that the bias results from the network output not reaching its new steady state within the trial duration (see SI Methods). How fast the new steady state is reached depends only on the time constants in the network and not the trial-to-trial variability. Next, we considered high stimulus uncertainty with zero trial-to-trial uncertainty. In this case, the contraction bias is largely independent of the stimulus variance (Fig. 5D). Our mathematical analysis reveals that the bias is well described by the difference between the prediction (that is, the mean stimulus over the history of all stimuli shown) and the current stimulus, weighted by a function of the trial duration.

The analysis of both limit cases suggests that the bias also depends on the trial duration. To confirm this, we extended the trial duration for either limit case. As expected from the analysis, the bias decreases steadily in the simulations (Fig. 5E). We, therefore, predict that the contraction bias can be reduced for sufficiently long trials.

We assumed that the stimulus variance is independent of the stimulus mean. Consequently, the bias at both ends of the input

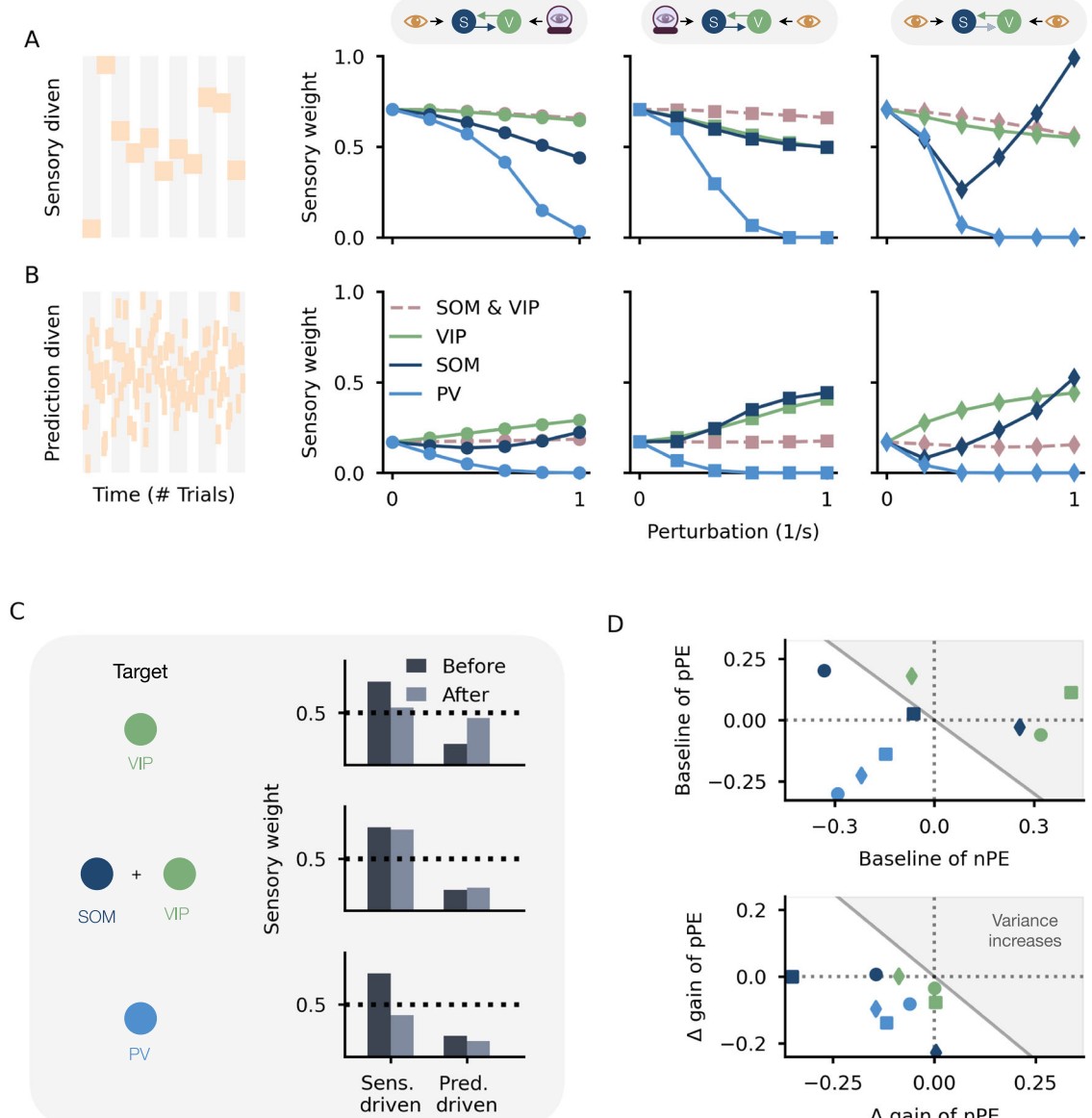

**Fig. 4 | Neuromodulator-based shifts in the weighting of sensory inputs and predictions. A, B** Neuromodulators acting on the interneurons can shift the weighting of sensory inputs and predictions. The changes depend on the type of interneuron/s targeted, the modulation strength (here simulated through an additional excitatory input), and the network's connectivity. Three mean-field networks were tested that differed in terms of the inputs to the SOM and VIP neuron, and, hence, the underlying PE circuit connectivity (see Methods). The first mean-field network is the one used in the other figures: The SOM neuron receives the feedforward input (depicted by an orange eye), while the VIP neuron receives the prediction thereof (depicted by an eye in a crystal ball). In the second MFN, the inputs onto the SOM and VIP neurons were swapped. In the third MFN, both the SOM and the VIP neuron receive the feedforward input. Two input regimes were tested: a sensory-driven (**A**) and a prediction-driven (**B**) regime before modulation.

**C** The take-home messages from the simulation results shown in **A** and **B** are summarized. A neuromodulator that acts mainly on the VIP neuron pushes the sensory weight towards 0.5. When SOM and VIP neurons are equally modulated, the sensory weight remains unaffected. A neuromodulator that acts mainly on the PV neurons reduces the impact of the sensory stimuli. **D** The V neuron activity, and hence the sensory weight, changes as a result of the modulated PE neuron activity. The PE neuron activity, on the other hand, changes as a result of the interneurons being modulated. The interneurons change the baseline (left) and the gain (right) of the PE neurons. Whether an interneuron increases or decreases the estimated variance depends on both factors. To estimate changes in baseline and gain of nPE and pPE neurons, we fitted a linear function to the PE neuron activity for the input range [0, 2.5].

distribution is similar but reversed in sign. However, behavioral data (for example, ref. 46) shows that the bias increases for stimuli from the upper end of the distribution, a phenomenon usually attributed to scalar variability. Modeling the stimulus standard deviation as linearly increasing with the stimulus mean, we also observed an increased bias for higher trial means (Supplementary Fig. 14).

In summary, we show that the weighted integration of sensory inputs and predictions can be interpreted as a neural manifestation of the contraction bias. While the stimulus and trial-to-trial variability

shape the contraction bias, their contributions differ. Moreover, we reveal that the trial duration contributes to the bias.

## Discussion
Our work has been driven by the puzzling question of how the brain integrates top-down feedback predictions with the sensory feedforward inputs it constantly receives during behavior. This task is particularly challenging when predictions and sensory information differ[47]. Conflicting information may arise from noise in the sensory inputs or

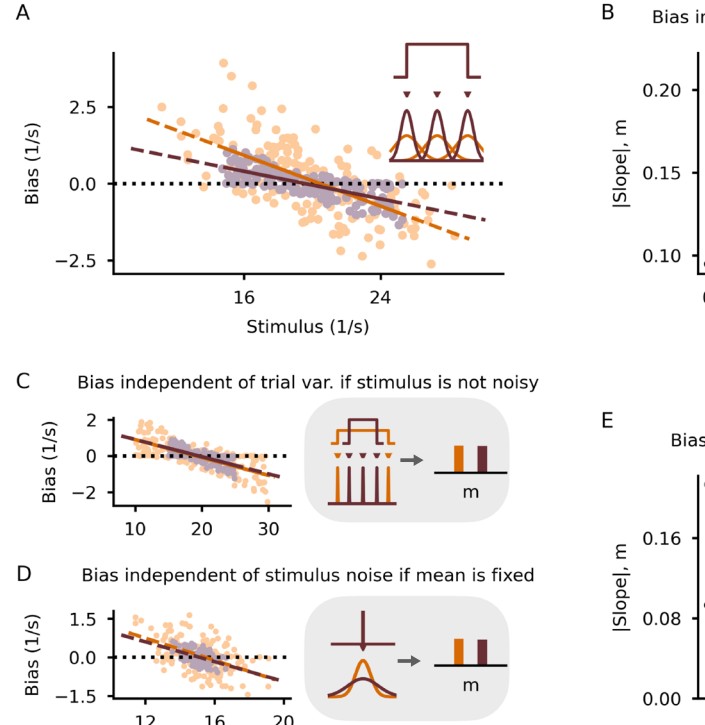

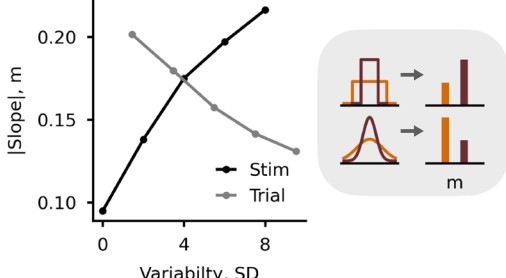

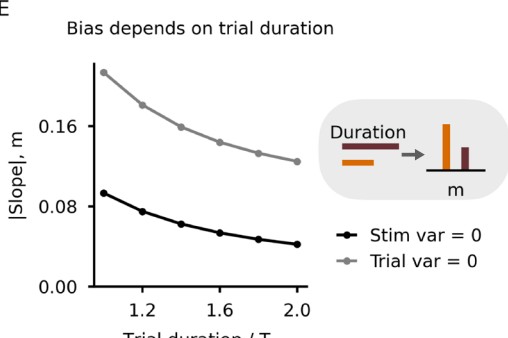

**Fig. 5 | Mechanisms underlying the contraction bias. A** Contraction bias in the model for two different stimulus uncertainties depicted in the inset. Bias is defined as the weighted output minus the stimulus mean. The absolute value of the slope of the linear fit, m, is a measure of the bias. The larger the slope, the larger the bias. **B** As a consequence of the sensory weight, the slope increases with stimulus variability (bias increases) and decreases with trial-to-trial variability (bias decreases). **C** Bias is independent of the trial-to-trial variability when the stimulus variability is zero. **D** Bias is independent of the stimulus variability when the trial-to-trial variability is zero. **E** The slope depends on the trial duration.

from unpredictable changes in the environment. A prominent hypothesis suggests that the degree to which we rely on predictions versus new sensory evidence is determined by an intricate balance based on the reliability of each source (for example, see refs. 4,9).

This idea aligns with Bayesian theories on the optimal integration of multiple sensory cues (multisensory integration). Ernst and Banks[2] demonstrated that humans estimate the height of a bar by combining visual and haptic information in a manner that minimizes the variance of the final estimate. Similar studies have confirmed that animals can optimally combine multiple sensory information by taking into account their uncertainties[3–8]. These behavioral findings were accompanied by neural recordings identifying populations of neurons that can form the basis of multisensory integration[7,8,48].

Here, we show that PE neurons can serve as the backbone for estimating the uncertainty of both the feedforward sensory inputs and the feedback predictions (Figs. 2 and 3). Our model proposes a hierarchy of PE circuits connected through the lower-order memory neuron, whose activity encodes the mean of the sensory bottom-up inputs. This local prediction is fed back to the lower-order circuit and simultaneously feed-forwarded to the higher-order subnetwork (Fig. 1). With this architecture, we show that we rely more strongly on our internal signals when the perceived sensory cues are noisier than the predictions. Moreover, our work suggests that predictions modulate neural activity more at the onset of a new sensory input, even if the stimulus is not noisy.

Relying more on predictions at the onset of a new trial, immediately after a change point, can be suboptimal. It was found that subjects tended to discard their predictions immediately following a change point in a sound-localization task where subjects were asked to predict the next stimulus after observing a series of stimuli[38,39]. However, as noted in the study, participants were informed about the

nature of the task, which could have influenced their responses. In situations where the underlying task is not explicitly known, the strategy may be less clear. This is particularly true when changes occur in sensory noise rather than in the environment (that is, $\sigma$ than $\mu$). In such cases, a reliability-based weighting of sensory input and predictions might offer an advantage. Nevertheless, our model could be extended to include a change-point detection mechanism (for e.g., see ref. 39), which could help reduce the observed discrepancy immediately after a change point.

We show that the weighting of sensory inputs and predictions can be biased by neuromodulators, as previously suggested (for example, see ref. 9). In our model, these modulatory signals act through interneurons[42] whose activities increase in the presence of neuromodulators. When PV neuron activity increases, the network weighs predictions stronger than without modulation. In contrast, when VIP neuron activity increases, the network underestimates the uncertainty of the prediction in a sensory-driven regime, and it underestimates the uncertainty of the sensory input in a prediction-driven regime. This results in the system weighting sensory inputs and predictions more equally (Fig. 4A). When SOM and VIP neuron activities are modulated to the same degree, the weighting remains unaffected, suggesting that the individual contributions cancel (Fig. 4B). These findings can be explained by changes in the baseline and gain of PE neurons arising through the modulation of interneuron activity (Fig. 4D). The results can be tested experimentally by optogenetically or pharmacologically stimulating specific interneuron types. Finally, we illustrate that the weighted integration of feedforward and feedback inputs can be interpreted as a neural manifestation of the contraction bias.

What could be the biological basis for our network model? Sensory information is commonly believed to be channeled through the thalamus and initially arrives in layer 4 of the neocortex[49,50]. Neurons in layer

4 subsequently relay the information to layer 2/3[49], where it is further integrated with inputs from higher-order cortical areas entering layer 1[51]. From layer 2/3, the information is subsequently forwarded to layer 5 neurons[49,52], which integrate it with direct inputs from the thalamus[53].

The core hypothesis of our model is the presence of sensory PE neurons that have been predominately found in layer 2/3, in different brain areas of various species[11,12,14,16–18]. While we assume these neurons encode PEs in their activity, it remains an active research area whether PEs are encoded in the (spiking) activity of separate neurons and/or in the local voltage dynamics of dendrites[54]. Recent findings by Gillon et al.[55] indicate that pattern-violation signals are processed differently at the soma and dendrites over time, suggesting a more complex role for excitatory neuron compartments in predictive processing than our simplified model accounts for.

In our network, memory neurons could correspond to a subset of excitatory L2/3 neurons. Some L2/3 neurons have been shown to develop predictive responses to expected visual stimuli[56]. Additionally, a group of L2/3 neurons has been shown to integrate both negative and positive prediction errors[57], which aligns with our assumption. The weighted output of our network aligns with the concept of internal representation neurons in predictive processing theories[10,58], hypothesized to be deep-layer 5 (L5) neurons[58,59]. These large pyramidal cells in L5 are ideally situated to integrate top-down information (e.g., predictions) arriving at their apical dendrites in layer 1 with bottom-up information (e.g., sensory inputs) arriving at their basal dendrites in deeper layers[34,60].

Neurons encoding variance in primary sensory areas remains a prediction of our model that requires validation. However, it has been shown that stimulus uncertainty can be encoded in the gain variability of individual neurons in V1 and V2 of macaques[61]. Evidence also indicates that populations of neurons can encode uncertainty[62]. For instance, neurons in the parietal cortex of monkeys encode confidence in perceptual decisions[63], and neurons in the orbitofrontal cortex encode confidence regardless of sensory modality[64]. Neural signatures of uncertainty have also been found in regions of the prefrontal cortex[65], the rat insular and orbitofrontal cortex[66], and the dorsal striatum in monkeys[67]. Additionally, the accuracy of memory recalls is encoded in single neurons of the human parietal and temporal lobes[68,69].

In our computation, the relative weights with which the sensory input and the prediction are integrated depend on the activities of the lower-level and higher-level variance neurons. While it is unlikely that these variance neurons can directly modulate the weights, they might trigger the release of neuromodulators that then, in turn, affect the synaptic plasticity of those weights[70,71]. For example, deep L5 neurons, which have been hypothesized to act as internal representation neurons[58,59] could be modulated in this manner. Depending on the receptor types present in the apical and basal dendrites, neuromodulators could either decrease or increase the synaptic weights connecting the sensory input and the prediction to these neurons.

Alternatively, the integration of sensory inputs and predictions could occur without changes in synaptic weights, implemented through a network of neurons encoding different aspects of the computation via their activities. For instance, an inhibitory neuron could encode the sum of the variances and exert divisive inhibition on another neuron, which is driven by the sensory input and the higher-order variance neuron in a multiplicative manner. Interneurons such as PV or SOM neurons have been shown to exert divisive inhibition[72,73]. Moreover, these computations could also be carried out by different compartments within the same neuron. For example, a deep L5 pyramidal cell may receive the sensory input at its basal dendrites and the activity of the higher-order variance neuron at its apical dendrites.

Neuromodulators correlate with uncertainty and influence the weighting of sensory inputs and their predictions[9]. Theoretical work by Yu and Dayan[74] suggests that acetylcholine (ACh) correlates with expected uncertainty, while noradrenaline (NA) correlates with unexpected uncertainty. Expected uncertainty is usually interpreted as known cue-outcome unreliabilities, whereas unexpected uncertainty relates to the changes in the environment that produce large PEs outside the expected range of uncertainties[74]. While in our network, the stimulus and trial-to-trial variability can only be loosely interpreted as 'expected' and 'unexpected' uncertainty, we discuss the conditions under which our network aligns with findings on ACh and NA.

NA is believed to increase in more volatile environments and enhance bottom-up processes[9,75]. In line with this idea, NA blockade impairs cognitive flexibility[76,77]. Recent work by Lawson et al.[78] shows that humans receiving propranolol (blocking NA) rely more on expectations and are slower to update their predictions despite new sensory evidence[9]. A main target for noradrenergic inputs is SOM neurons, whose activity increases in the presence of NA (reviewed in refs. 43,44,79). In our model, activating SOM neurons does not enhance sensory bottom-up input. In a volatile environment, that is, a sensory-driven regime, the system takes into account predictions slightly more than without SOM modulation (Fig. 4A and Supplementary Fig. 15).

However, we assumed that neuromodulators act globally, that is, on the interneurons in both the lower and the higher PE circuit. While this agrees with the view that neuromodulators can control network states globally, there is also evidence that they can have a more local, finely adjusted impact on neural circuits[80]. In our model, increasing SOM activity only in the lower-order circuit slightly enhances the sensory weight (Supplementary Fig. 16), that is, the bottom-up inputs.

Similarly to NA, ACh has also been shown to enhance bottom-up, feedforward inputs (reviewed in refs. 74,81). For instance, subjects relied more on prior beliefs when given cholinergic receptor antagonists[81]. A major target for cholinergic inputs is VIP neurons, whose activity increases in the presence of ACh (reviewed in refs. 43–45). In our model, globally activating VIP neurons enhances bottom-up input in stable environments for noisy stimuli. However, increasing VIP activity only in the higher-order PE circuit generally enhances sensory bottom-up inputs (Supplementary Fig. 16). This suggests that whether a neuromodulator biases the network toward feedforward bottom-up or feedback top-down inputs depends on its spatial and temporal scale of influence.

Our model suggests one potential neuronal circuit mechanism for the uncertainty estimation of sensory inputs and predictions. Modeling specific neurons that encode the variance of feedforward sensory inputs and predictions aligns with the concept that neurons can explicitly represent parameters of a probability distribution, such as the mean or variance (see also[82–84]). However, the representation of variances in the brain is still not comprehensively understood, and several alternative models have been proposed. For instance, uncertainty might be decoded from the collective activity of neuron populations[85–88]. Some theories suggest that uncertainty is represented by the amplitude[87], the width[89] or the variability of a neuron's response[87,90]. Another prominent theory is the neural sampling hypothesis, which suggests that neural circuits encode probability distributions rather than precise values. In this framework, the variability in neural responses is interpreted as samples drawn from these distributions[91–93].

Many normative models have been proposed for state estimation and prediction under uncertainty[62], ranging from the classical Kalman filter to more recent models like Bayes Factor Surprise[94]. For instance, the Bayes factor surprise formularizes the trade-off between integrating new observations in an existing belief system and resetting this belief system with novel evidence. The surprise factor captures how much an animal's current belief deviates from the new observation.

In recent years, normative models have also been squared with biological constraints. For instance, Kutschireiter et al.[95] showed that a Bayesian ring attractor model can encode uncertainty in the amplitude of the network activity and match the performance of a circular

Kalman filter when the recurrent connections are tuned appropriately. In other seminal work, it has been proposed that Bayesian inference in time can be linked to the dynamics of leaky integrate-and-fire neurons with spike-dependent adaptation[96].

Furthermore, in our model, we assume that the lower-level prediction is not only fed back to neurons in the lower-level PE circuit but is also forwarded to the higher-level subnetwork. Unlike classical hierarchical predictive coding frameworks, where PEs are sent up the hierarchy, we propose that PEs are processed locally. However, we are not the first to consider an alternative model. For instance, Spratling[97] reformulated the predictive coding model by Rao and Ballard[15] within the context of a biased competition model. In Spratling's model, cortical borders are also redefined so that prediction neurons in the lower-level cortical area connect to error neurons in the higher-level cortical area. However, we note that these views are not mutually exclusive and might differ between brain areas or species. We discuss in more detail alternative network models in the Supplementary Discussion.

The notion that neural implementations for the integration of inputs may vary across species and modalities is supported by work on multisensory integration. Wong et al.[98] showed that in Drosophila larva the chosen cue-combination strategy varies depending on the type of sensory information available. Also, humans put typically more weight on visual than auditory cues[3,5], but trust vestibular information more than visual information about head direction[99], a finding also observed for monkeys[100]. Moreover, Summerfield et al.[101] showed that humans diverge from an optimal Bayesian strategy in very volatile environments and act according to their experience in the last trial. It has been suggested that the brain may use different strategies to combine signals depending on the task demands[84].

Here, we propose a view in which PE neurons serve as the backbone for estimating both the uncertainty of the feedforward sensory stimuli arising from the external world and the feedback signals carrying predictions about the same feedforward inputs our brains receive. Our work is an important step toward a better understanding of the brain's ability to integrate these unreliable feedforward and feedback signals that often do not match perfectly.

## Methods
### Network model
The mean-field network model consists of a lower and higher PE circuit (Fig. 1C, D). Each PE circuit contains an excitatory nPE neuron and pPE neuron ($N_{nPE} = N_{pPE} = 1$), as well as inhibitory neurons. The inhibitory neurons comprise PV, SOM, and VIP neurons ($N_{SOM} = N_{VIP} = 1$, $N_{PV} = 2$), further explained in ref. 26. In addition to the core PE circuit, each subnetwork also includes one memory neuron $M$ and one variance neuron $V$.

The excitatory neurons in the PE circuit are simulated as two coupled point compartments, representing the soma and the dendrites of elongated pyramidal cells. All other neurons are modeled as point neurons. The activities of all neurons are represented by a set of differential equations describing the network dynamics.

The dynamics of the neurons in the lower and higher PE circuits ($\underline{r}_{PE}^{low}$ and $\underline{r}_{PE}^{high}$) are given by

$$\begin{aligned} \underline{r}_{PE}^{low} &= \left[\underline{h}_{PE}^{low}\right]_+ \\ \underline{r}_{PE}^{high} &= \left[\underline{h}_{PE}^{high}\right]_+ \end{aligned} \tag{1}$$

with

$$\begin{aligned} T_c \cdot \dot{\underline{h}}_{PE}^{low} &= -\underline{h}_{PE}^{low} + W_{PE\leftarrow PE} \cdot \underline{r}_{PE}^{low} + \underline{w}_{PE\leftarrow M} \cdot r_M^{low} + \underline{w}_{PE\leftarrow FF} \cdot s + \underline{I}_{PE} \\ T_c \cdot \dot{\underline{h}}_{PE}^{high} &= -\underline{h}_{PE}^{high} + W_{PE\leftarrow PE} \cdot \underline{r}_{PE}^{high} + \underline{w}_{PE\leftarrow M} \cdot r_M^{high} + \underline{w}_{PE\leftarrow FF} \cdot r_M^{low} + \underline{I}_{PE}. \end{aligned} \tag{2}$$

We follow the notation that column and row vectors are indicated by letters with an underscore $\bullet$, matrices are denoted by capital letters, and scalars are given by small letters without an underscore. Furthermore, a time derivative (e.g., $\frac{dx}{dt}$) is denoted by a dot above the letter (e.g., $\dot{x}$). The rate vector $\underline{r}_{PE}^{loc} = \left[r_{nE}^{loc}, r_{pE}^{loc}, r_{nD}^{loc}, r_{pD}^{loc}, r_{PV_1}^{loc}, r_{PV_2}^{loc}, r_{SOM}^{loc}, r_{VIP}^{loc}\right]$ with loc $\in$ {low, high} contains the activities of all neurons or compartments in the PE circuit (soma of nPE/pPE neurons: nE/pE, dendrites of nPE/pPE neurons: nD/pD). The network receives time-dependent stimuli $s$ and neuron/compartment-specific external background input $\underline{I}_{PE}$. The connection strengths between the pre-synaptic population and the neurons of the PE circuit are denoted by $W_{PE\leftarrow pre}$ (if $r$ is a vector) or $\underline{w}_{PE\leftarrow pre}$ (if $r$ is a scalar). The activities of the neurons evolve with time constants summarized in the diagonal matrix $T_c$ (entries that correspond to an excitatory cell are set to $\tau_E = 60$ ms, while entries that correspond to the inhibitory neurons are set to $\tau_I = 2$ ms, in line with ref. 26).

The activities of the lower and higher memory (M) neuron evolve according to a perfect integrator. The M neurons receive synapses from both nPE and pPE neurons of the same subnetwork,

$$\begin{aligned} \tau_E \cdot \dot{r}_M^{low} &= \underline{w}_{M\leftarrow PE}^{low} \cdot \underline{r}_{PE}^{low} = w_{M\leftarrow pPE}^{low} \cdot r_{pPE}^{low} - w_{M\leftarrow nPE}^{low} \cdot r_{nPE}^{low} \\ \tau_E \cdot \dot{r}_M^{high} &= \underline{w}_{M\leftarrow PE}^{high} \cdot \underline{r}_{PE}^{high} = w_{M\leftarrow pPE}^{high} \cdot r_{pPE}^{high} - w_{M\leftarrow nPE}^{high} \cdot r_{nPE}^{high}. \end{aligned} \tag{3}$$

Please note that although the time constants for the lower and higher M neurons are identical, their effective time constants differ due to variations in the weights connecting the PE neurons with the M neurons (the effective time constant of the higher subnetwork is between 4 and 64 times larger than that of the lower subnetwork, see 'Connectivity').

The activities of the lower and higher V neuron evolve according to a leaky integrator with quadratic activation function ($\tau_V = 5$ s). The variance neurons receive synapses from both nPE and pPE neurons of the same subnetwork,

$$\begin{aligned} \tau_V \cdot \dot{r}_V^{low} &= -r_V^{low} + \left(\underline{w}_{V\leftarrow PE} \cdot \underline{r}_{PE}^{low}\right)^2 = -r_V^{low} + \left(w_{V\leftarrow pPE} \cdot r_{pPE}^{low} + w_{V\leftarrow nPE} \cdot r_{nPE}^{low}\right)^2 \\ \tau_V \cdot \dot{r}_V^{high} &= -r_V^{high} + \left(\underline{w}_{V\leftarrow PE} \cdot \underline{r}_{PE}^{high}\right)^2 = -r_V^{high} + \left(w_{V\leftarrow pPE} \cdot r_{pPE}^{high} + w_{V\leftarrow nPE} \cdot r_{nPE}^{high}\right)^2. \end{aligned} \tag{4}$$

Details on the model equations for the mean-field and the multi-cell population network, as well as supporting analyses can be found in the supplementary material.

### Connectivity
The connectivity within a PE circuit, $W_{PE\leftarrow PE}$, can be found in ref. 26, in the simulation code provided (see below), and the Supplementary Data 1–3. The vector $\underline{w}_{PE\leftarrow M}$ contains the connection strengths between the memory neuron M and the post-synaptic neurons X in the PE circuit, $w_{X\leftarrow M}$. If a connection exists, $w_{X\leftarrow M} = 1$, $w_{X\leftarrow M} = 0$ otherwise. In all mean-field networks tested, the dendrites of nPE and pPE neurons and one of the two PV neurons receive connections from the memory neuron. Whether the SOM or VIP neurons are the target of the feedback projections depend on the specific mean-field network tested (see Supplementary Table 2 and ref. 26).

The vector $\underline{w}_{PE\leftarrow FF}$ contains the connection strengths between the feedforward input and the post-synaptic neurons X in the PE circuit, $w_{X\leftarrow FF}$. The feedforward input is either the direct sensory input $s$ for the lower PE circuit, or the activity of the lower-level M neuron, $r_M^{low}$, for the higher PE circuit. In general, for the three mean-field networks tested, we chose $w_{X\leftarrow FF} = 1 - w_{X\leftarrow M}$.

The connection strength between the nPE/pPE neuron and the memory neuron $M$ in the mean-field network is $w_{M\leftarrow nE}^{loc} = \frac{\lambda^{loc}}{g_{nPE}}$ and $w_{M\leftarrow pE}^{loc} = \frac{\lambda^{loc}}{g_{pPE}}$, respectively, where $\lambda^{loc}$ denotes the non-normalized weight for the lower or higher-order PE circuit, loc $\in$ {low, high}. In the

**Table 1 | Parameters used to stimulate the network**

| Fig. | | # trials | $N_{in}$ | $N_{step}$ | $\mu_{in}$ | $\sigma^2_{in}$ | (a, b) or $\sigma^2_{trial}$ | pert. stgth | note |
|---|---|---|---|---|---|---|---|---|---|
| 1 | B | 1 | 1 | $10^5$ | [0,10] | 0 | - | - | M = 5 |
| 2 | B,C | 1 | 200 | 500 | [3,6] | [3,6] | - | - | $\mu_{in}$ varied: $\sigma^2_{in}$ = 4.5, $\sigma^2_{in}$ varied: $\mu_{in}$ = 4.5 |
| | D,E | 1 | 200 | 500 | [3,6] | [3,6] | - | - | example: $\mu_{in}$ = 5, $\sigma^2_{in}$ = 4 |
| 3 | C | 100 | 10 | 500 | 5 | 0 | (1,9) | - | - |
| | D | 100 | 10 | 500 | 5 | 5 | (5,5) | - | - |
| | E | 100 | 10 | 500 | 5 | [0,25] | $\sigma^2_{trial} \in [0,25]$ | - | - |
| | F | 100 | 10 | 100, 500 | 5 | 0 | 9 | - | - |
| 4 | A | 200 | 10 | 500 | 5 | 0 | 1 | [0,1] | perturb. |
| | B | 200 | 10 | 500 | 5 | 1 | 0 | [0,1] | in last 100 |
| | D | 200 | 10 | 500 | 5 | 5 | 0 | 0.5 | trials |
| 5 | A | 200 | 10 | 500 | - | 1 or 49 | (15,25) | - | - |
| | B | 200 | 10 | 500 | - | [0,64] or 25 | a = 15 | - | - |
| | | | | | | | b = 25 or [20,48] | | |
| | C | 200 | 10 | 500 | - | 0 | (15,25) or (10,30) | - | - |
| | D | 200 | 10 | 500 | - | 4 or 25 | (15,15) | - | - |
| | E | 200 | 10 | $[0.5,1] \times 10^3$ | - | 0 or 25 | (15,25) or (15,15) | - | - |

To increase readability, we do not include units for the parameters. All units can be deduced from the equations. $N_{in}$: number of values each stimulus is composed of, $N_{step}$: number of consecutive time steps each value is presented, trial duration is given by $N_{in} \cdot N_{step}$, $\mu_{in}/\sigma^2_{in}$: mean/variance of the normal distribution from which the stimulus values are drawn, (a, b): lower and upper bound of the uniform distribution $\mu_{in}$ is drawn from in each trial, $\sigma^2_{trial}$: standard deviation of the uniform distribution $\mu_{in}$ is drawn from in each trial. – indicates that parameter/s is/are not used in the simulation.

lower PE circuit, $\lambda^{low} = 3 \times 10^{-3}$ for the mean-field model in Fig. 2 and $\lambda^{low} = 4.5 \times 10^{-2}$ for Figs. 3–5. In the higher PE circuit, $\lambda^{high} = 7 \times 10^{-4}$. The gain factors, $g_{nPE}$ and $g_{pPE}$ depend on the mean-field network tested and can be found in ref. 26 or the supplementary material. Similarly, the connection strength between the nPE/pPE neuron and a V neuron is given by $w_{V \leftarrow nPE/pPE} = \frac{1}{g_{nPE/pPE}}$.

Details on the multi-cell population networks can be found in the supplementary material.

### Inputs

All neurons of the core PE circuit receive an external background input (summarized in the vector $I_{PE}$) that ensures reasonable baseline firing rates in the absence of sensory inputs and predictions thereof. In line with ref. 26, these inputs were set such that the baseline firing rates are $r_{pE} = r_{pD} = r_{nE} = r_{nD} = 0\ s^{-1}$ and $r_P = r_S = r_V = 4\ s^{-1}$. In addition, the network receives feedforward stimuli $s$ that may vary between trials. To account for noise, each stimulus is composed of $N_{in}$ constant values drawn from a normal distribution with mean $\mu_{in}$ and standard deviation $\sigma_{in}$. Each value is presented for $N_{step}$ consecutive time steps (each time step is 1 ms). To account for changes in the environment, $\mu_{in}$ is drawn from a uniform distribution $U(a, b)$ with mean $\mu_{trial} = \frac{a+b}{2}$ and standard deviation $\sigma_{trial} = \frac{b-a}{\sqrt{12}}$. To test different input statistics, the parameterization of both distributions varies across the experiments (see Table 1). All stimulus/input parameters can also be found in the supplementary material, in addition to the parameters for the Supplementary Figs.

### Simulations

All simulations were performed in customized Python code written by LH. Differential equations were numerically integrated using a 2nd-order Runge-Kutta method. Neurons were initialized with $r = 0/s$.

### Weighting of sensory inputs and predictions

We arithmetically calculated the weighted output of sensory inputs and predictions, $r_{out}$, based on ideas of Bayesian multisensory integration (for e.g., see ref. 40),

$$r_{out} = \alpha \cdot s + (1 - \alpha) \cdot r_M^{low}, \tag{5}$$

where $\alpha$ denotes the sensory weight (that is, the reliability of the sensory input) and is given by

$$\alpha = \left(1 + \frac{r_V^{low}}{r_V^{high}}\right)^{-1}. \tag{6}$$

### Reporting summary

Further information on research design is available in the Nature Portfolio Reporting Summary linked to this article.

## Data availability

All data is generated through simulations, with the corresponding code publicly available (see Code availability statement below).

## Code availability

The Python source code (v1.0.0) for reproducing the simulations, analyses, and figures can be accessed at https://github.com/lhertaeg/weighted_sensory_prediction[102].

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

## Acknowledgements

We thank Inês C. Guerreiro for comments on earlier versions of this manuscript. L.H. is supported by Deutsche Forschungsgemeinschaft (DFG) Grant 46008809. C.C. is supported by Biotechnology and Biological Sciences Research Council (BBSRC) Grants BB/N013956/1 and BB/N019008/1, Wellcome Trust Grant 200790/Z/16/Z, Simons Foundation Grant 564408, and Engineering and Physical Sciences Research Council (EPSRC) Grant EP/R035806/1.

## Author contributions

L.H., K.W., and C.C. conceived the project and designed the experiments. L.H. performed the simulations and analyses. L.H., K.W., and C.C. interpreted the results and wrote the paper.

## Funding

## Competing interests

The authors declare no competing interests.
