## [Transparent Peer Review file · Nature Communications]

Uncertainty estimation with prediction-error circuits

Corresponding Author: Dr Loreen Hertäg

Version 0:

Reviewer comments:

Reviewer #1

(Remarks to the Author)

In this manuscript entitled "Knowing what you don't know: Estimating the uncertainty of feedforward and feedback inputs with prediction-error circuits", the authors propose a microcircuit model for how excitatory neurons and inhibitory interneurons compute prediction errors as well as estimate the reliabilities of the feedforward "sensory" input and the feedback "predictive" signal.

The main building block of this model, extracting positive and negative prediction errors from a randomly time-varying signal, has already been described in previously published studies. This manuscript propose several new extensions. First, the activity of the positive and negative "predictive coding" neurons are combined by two units, one estimating the mean, and the second the variance. Second, two such microcircuits are stacked hierarchically, the second performing the same operation but at a slower time scales on the estimate generated by the first. And finally, the output of the two microcircuits are combined in a "Bayesian" manner by weighting each according to the estimated variance. Note that this last part is analytical, and not performed by the circuit. Finally, they show that neuromodulators acting on different components of this circuit would bias inference differently.

A powerful motivation for this approach is that such microcircuit could be an ubiquitous component for implementing hierarchical Bayesian inference in brain circuits. However, I have some serious issues with the manuscript, as listed below.

First, one essential component of the circuit, computing positive and negative prediction errors, has already been published. Apart from referring to this other article, the authors do not give any specifics on the connection weights or why this architecture was chosen. As a result, we do not have the rational behind the circuit organization (e.g. what is the role of VIP neurons, why two PV neurons, or a separate dendritic compartment for excitatory neurons?). Moreover, one cannot replicate these results based on the information provided.

Second, the variance-computing neurons do not, generally speaking, measure a variance. More exactly, they do so if and only if the sensory signals are generated exactly as the author describe.

Third, despite the authors mentioning feedback in the title, there is no feedback connection in the model (at least between the two hierarchical levels). Similarly, prediction errors are usually understood as the difference between a sensory signal and a top-down prediction. This is not the case here. The estimates from the two hierarchical levels are combined offline, not implemented in the circuit, which is misleading.

More generally, this circuit lacks generality (it works only for a specific type of signal modulation and noise, in the absence of any prior or "top down" information). The title is an overstatement.

I have also more minor issues.

First, while there is experimental evidence for negative and positive prediction error coding neurons in sensory areas, it is not so much the case for variance computing neurons. Neurons specifically tuned to variance have been reported in high level or "associative" areas (such as prefrontal cortex) in tasks where uncertainty plays a crucial role. To my knowledge, variance-tuned neurons have never observed in sensory areas, where the dominant effect of changing stimulus variance is to change response gains through adaptation.

Second, I think the reported simulation results for neuromodulation are not useful. It is unsurprising that perturbing the

circuit leads to biases in variance estimates. However, no simple, testable or non-trivial prediction seems to emerge from that. In addition, the figures are hard to read because of overlap between the different symbols.

(Remarks on code availability)

Reviewer #2

(Remarks to the Author)

Hertäg and colleagues describe a model based around a prediction-error circuit that is an elegant reflection of biological local excitatory and inhibitory circuits that function to estimate the uncertainty in sensory input and determine when the use of sensory input is balanced by predictive processing. The work adds to a growing number of incredibly biologically relevant circuit models that have proven to be highly versatile for forming predictions of how sensory integration and predictive processing would work in the brain. The paper is well written and describes the dynamics of the prediction-error circuits as well as making a number of testable predictions regarding their modulatory dynamics that will be of high interest not only for the field on computational neuroscience, but for sensory neurophysiologists and systems neuroscientists in general. While the complexity of the computational modelling may limit the overall general audience, the discussion is very accessible and relates the model and the findings of the current study to a number of important neuroscientific phenomena and supporting literature. Following are some specific comments largely related to the general limits of the model and the implementation of the initial state. It is possible that these details are available in previous publications developing the basis of the model, the code directly or specified in the methods section, but I would still recommend that some additional brief information be added to the main text in this regard to increase accessibility to a wider audience.

Specific comments:

1. Lines 86-87: 'Thus, the squared sum of nPE and pPE neuron activity represents the variance of the feedforward input (provided that the PE neurons are silent without sensory stimulation)'. It is unclear if this means that the baseline activity would be zero (see related comments below). If the baseline were not zero, is it possible to evaluate how a potential bimodal modulation of the PE elements affect the model?
2. Lines 92-92: 'While the excitatory neurons are simulated as two coupled point compartments to emulate the soma and dendrites of elongated pyramidal cells, respectively, all inhibitory cell types were modeled as point neurons'. Where there any spatial specification of the inhibitory interneuron populations? Perhaps with connectivity of the PV interneurons 'closer' or more highly connected to the soma and SOM to the dendrites for instance? Or did all interneurons influence the same compartments? This is indicated in the schematic in Figure 1B and alluded to in lines 102-105, but then it is unclear in the main text how this is precisely implemented. Similarly, the M and V neurons only had single compartments then without dynamics related to compartments?
3. How was the baseline activity of the pPE and nPE neurons established and is it relevant? Presumably these must be set to the same values to balance some initial state. Also in relation to the interneurons, does the baseline activity have an impact on the modulation via interneurons in the model. Perhaps relatedly, is the baseline activity of the pPE and nPE neuron always the same – can a neuron be considered to have a predictive initial state that changes under different circumstances in the model (e.g. changing M activity after initial encoding, after it approaches the mean of the sensory inputs)?
4. For Figure 3E and the testable finding of the model: 'sensory predictions influence neural activity more significantly in experiments that rely on fast stimulus changes', presumably there is no lower limit integration time in the model – as in, the sensory weight is high at time zero. Also, it is unclear what values/limits correspond to T and T/5. Some intuition about this would be helpful.
5. The justification for the three different PE circuits in Figure 4 (denoted by different markers) is unclear. Why these particular circuit parameters and combination of sensory input? Some additional information comes in the very nicely written discussion – but some clarity in the results would also help the interpretation here.

Minor comment:

- The direction of modulation in lines 55-57 seem to be backwards for the specified PE types.

(Remarks on code availability)

Reviewer #3

(Remarks to the Author)

In this manuscript, Hertag and colleagues use a rate-based circuit model composed of different cell types to propose a possible mechanism explaining how uncertainty of feedforward sensory stimuli and feedback predictions can be estimated within a cortical circuit. Specifically, the circuit model builds on a model proposed in the authors' previous work (Hertag &

Clopath 2022 “Prediction-error neurons in circuits with multiple neuron types: formation, refinement, and functional implications”) which includes an excitatory, negative prediction error neuron (nPE), a positive prediction error neuron (pPE), and inhibitory PV, SOM, and VIP neurons, by adding a memory (M) neuron and a downstream variance (V) neuron. The model is wired and tuned so that the nPE responses increase when the prediction is stronger than the sensory input, and pPE responds strongly when the prediction is weaker than the sensory input. The M neuron represents the top-down prediction and relays feedforward signals from the lower to the higher PE circuit. It receives direct excitatory inputs from pPE neuron while being indirectly inhibited by nPE neuron. The V neuron receives excitatory inputs from both PE neurons.

The key findings of this study can be summarized as follows:

- In this circuit model, the M and V neurons approach the mean and the variance of the sensory inputs, respectively.
- By modulating the trial and stimulus statistics, the circuit weighs either the prediction or the sensory input more in its output.
- Neuromodulator effects simulated on each of the cell types produce different shifts in the weighting of sensory inputs and predictions.
- Contraction bias can be explained by the model’s estimation of stimulus and trial variability.

Understanding the computational roles of diverse cell-types in a biologically motivated circuit has been attracting a lot of attention in neuroscience recently. Furthermore, explicating their functional implications in a predictive coding framework is undoubtedly a very interesting topic in the field. However, there are some concerns regarding novelty and contributions of this work, which need to be addressed before the manuscript can be accepted for publication. The main concern is that it is rather questionable whether this work provides sufficiently more in-depth insights in addition to the authors’ previous papers, Hertag & Clopath 2022 “Prediction-error neurons in circuits with multiple neuron types: formation, refinement, and functional implications” and Hertag & Sprekeler 2020 “Learning prediction error neurons in a canonical interneuron circuit”. Secondly, while the model provides a plausible explanation for how the uncertainty of feedforward and feedback inputs can be computed in a neural circuitry, the circuit model itself is formulated precisely to produce the desired outcomes. For example, based on how the sensory input weight α is defined, it is not surprising that when the variance is high, then the weight on the sensory input decreases. Overall, the contribution of this work is rather limited to providing a circuit model for uncertainty estimation, which can be tested experimentally in the future but for now is still hypothetical. I believe the contribution of the manuscript would increase significantly if the authors also investigate some other alternative circuit models, and systematically evaluate the plausibility of each circuit model. Some additional comments are detailed in the following.

Major comments:

1. How pPE and nPE neurons produce responses that depend on the relative strength of prediction compared to stimulus is not clear in this manuscript. While I see that this is described in the cited, previous work of the authors, it will be helpful to include some brief description on how pPE and nPE neurons in the model produce the desired response patterns.
2. Is there any experimental evidence supporting the existence and the assumed connectivity of the M and V neurons proposed in the model? Are the results from the simulations of neuromodulator effects expected? There are some discussions on biological relevance in Discussions, but it will be helpful to motivate the implementation of M & V neurons and the neuromodulator simulations in the main text.
3. Why do the authors use discrete sequences of sensory stimuli, rather than continuous stimulus signals and modulating the temporal and spatial variability?
4. The Figure 4C is motivated on lines 226-228 “To disentangle the effect of nPE and pPE neurons, we perturbed those neurons individually in both the lower or higher subnetwork by injecting either an inhibitory or excitatory additional input.”, but it seems to show that perturbations of both nPE and pPE introduce exactly the same effect on V neuron, so the claim of “disentangling the effect of nPE and pPE neurons” can be misleading.
5. Is the weighted output hypothesized to be represented by a separate class of neurons?
6. The authors have proposed a mechanism underlying contraction bias using a similar model in their previous work (Hertag & Clopath 2022, Fig 5). Some clarification on the novelty and description that distinguish these two models of contraction bias should be included to signify the result in the manuscript.
7. What is the motivation to use an integrator with indirect inhibition from nPE for M neuron, and a leaky integrator with quadratic activation for V neuron?
8. In the hierarchical model of PE circuits, feedforward signals from the lower PE circuit to the higher PE circuit are relayed only via the M neuron which encodes the mean sensory inputs/expectations. However, in the traditional framework of hierarchical predictive coding, error signals (represented in PE neurons) are transmitted from the lower to the higher areas as feedforward signals. Some comments on this discrepancy would be helpful.

Minor comments:

1. The authors modeled the excitatory PE neurons as two-compartmental models representing the soma and the dendrites of pyramidal cells. In a recent paper by Gillon et al (2024, J Neuroscience), the dendrites and the soma of pyramidal neurons show distinct response patterns to expectation violation, which can be relevant for the two-compartmental model proposed in this work.
2. Figure 4 is a bit confusing. Fig4A markers have different colors noting sensory weights, but this does not add any new information as the y-axis already denotes Sensory Weight.
3. Typo on Line 258, “trail-averaged”->“trial-averaged”
4. Typo on Line 379, “trail-to-trial”->“trial-to-trial”
5. Typo in Supporting Information, Line133, “trail”->“trial”
6. The legend describing colors for “in lower PE circuit” and “in higher PE circuit” of Figure S8 is confusing.
7. While the colors and shapes of the data points in Fig S9 are provided in the main text, it will be nice to have the legend

included in Fig S9 as well.

8. In Fig 5D, the schematics showing different distributions of stimuli seems to be misleading. Shouldn't a schematic of Gaussian distributions with different widths be shown?

(Remarks on code availability)

The Github repository provided in the URL is not publicly accessible.

Reviewer #4

(Remarks to the Author)

The authors considered a scenario in which there is an incoming scalar sensory stimulus, s , which has some mean and variance (over trials). The job of the brain is to compute that mean and variance. The mean is what the authors call the prediction. When a new (noisy) stimulus comes in, the brain can infer its mean by combining the prediction and the current stimulus, weighted by their respective variances.

I think that's right, but I admit that I don't have a huge amount of confidence in. Assuming it is, though, I have a couple of issues with the paper.

1. What the authors call a "prediction" is really just a prior over the sensory stimulus, s . That's fine; it's certainly something the brain uses. But I don't think it's what most people would call prediction. And it confused me for a long time. It would have been a lot easier if they just said the network was computing a prior; then I would have instantly known what the model was supposed to do.

2. Trying to explain the network in words doesn't work, especially given that the words are somewhat vague (see lines 97-108). So I looked at Methods. After a couple hours I was able to sketch what the circuit looked like. Which wasn't all that complicated; it's not clear why the authors didn't do that (Figs. 1B-D are kind of close, but, in my opinion, were missing a lot of important detail).

Based on the equations in Methods, two things jumped out at me. First, the prediction was put into a "memory neuron", r_M , which was a perfect integrator. That seems a bit strange, since it would seem hard for the memory neuron to equilibrate at the mean value of the stimulus. It managed to (see Fig. 2D), but I can't figure out why. An explanation would be very useful here. In particular, did it require fine tuning?

Second, I was expecting the positive and negative prediction error neurons to be differentially driven by the stimulus, s , and the prediction, r_M . That is, positive prediction error neurons should increase their firing rate when $r_M < s$ and negative prediction error neurons should decrease their firing rate when $r_M < s$ (I think; I might have gotten that backwards). However, when I looked at the equations (Eq. 2 in Methods), as far as I could tell, both s and r_M positively modulated both types of prediction error neurons. But maybe that's not the case? It was somewhat hard to tell from the explanation, even when I looked in SI. This should be clear, as it seems critical.

3. Based on Eq. 6, it seems that variance was computed on two timescales. (There were two circuits, low and high, but I never actually got to the high one. Given my uncertainty on the low one, I didn't want to try.) The short timescale was for the stimulus. But that was only 60 ms, which seems way too short; most stimuli last for longer than that. But again, maybe I'm missing something?

4. Finally, it wasn't clear to me what the take-home message was. Because the computational problem was easy (compute priors), I assume the take-home message was how a circuit implemented it. But if that's the case, trying to figure out what the circuit was from the paper was quite difficult. What is needed is a clear circuit diagram, with clear explanations. In particular, I would have liked to know why there were so many inhibitory cell types. Was it, for instance, impossible to build a circuit with only one kind of inhibitory neuron? Or were they just trying to match data?

(Remarks on code availability)

As you can see, I'm not too excited about the paper. There may be something interesting there (although I kind of doubt it), but it was pretty much impossible for me to extract it.

Version 1:

Reviewer comments:

Reviewer #2

(Remarks to the Author)

The authors should be commended for investing a lot of effort into clarifying their model and to making it generally accessible to a wider scientific audience. They have addressed my concerns and have strengthened the manuscript with substantial additional Figures and associated discussion. I especially appreciate the efforts that went into the additional Supp Fig 1 and also the additional discussions and clarifications regarding the influence of the baseline dynamics.

In relation to the minor comment about lines 55-77, indeed the authors are correct – my confusion arose from cross-

referencing this text in the introduction to the accompanying Figure 1B caption text:

Lines 55-57: "negative PE (nPE) neurons only increase their activity when the prediction is stronger than the sensory input, while positive PE (pPE) neurons only increase their activity when the prediction is weaker than the sensory input."

Fig 1B caption: 'The nPE neuron only increases its activity relative to a baseline when the sensory input is weaker than predicted, while the pPE neuron only increases its activity relative to a baseline when the sensory input is stronger than predicted.'

Clearly both are actually correct, but on first reading they seemed contradictory since the prediction and the sensory input are switched in reference to each other. While I see the error now, for a first-time reader this was confusing. Personally, I find the order of the Fig 1B caption text more intuitive, but in general I would just suggest they pick one direction of reference and stick to this at least for the first few references of this relationship, and in the early descriptions of the model, for increased clarity and ease.

(Remarks on code availability)

Reviewer #3

(Remarks to the Author)

The revised manuscript has undergone significant edits for improved clarity and now includes additional simulations. Specifically, some of the major edits for improved clarity include 1) modified visualization of the main Fig 4 and Supplementary Figures 11 & 12, and 2) additional Supplementary Figures 1 & 2 which show the circuit model schematics and the relationship between the M neuron and the nPE/pPE neurons. Regarding additional simulations, the revised manuscript now includes three new Supplementary Figures 4, 5, and 6 on new results. In these newly added Supplementary Figures, the authors performed simulations with continuously varying stimulus signals and tested effects of varying some parameters of the circuit model such as nPE and pPE baselines, V neuron time constant, connection weight scaling, etc. Finally, the revisions include additional discussions as a part of the Supporting Information. These revisions addressed many of the suggestions from the first round of reviews, and the manuscript has improved a lot in clarity. However, in my opinion, the major concerns raised in the first round of reviews remain despite the revisions.

This work is an extension of the authors' previously published papers which formulate the circuit model involving nPE & pPE neurons and inhibitory subtypes. The novel aspects of this manuscript are that 1) the M and V neurons are added to the model which encode the mean and the variance of sensory inputs, respectively, 2) this circuit motifs are hierarchically combined, 3) the trial and stimulus statistics are shown to modulate weights on the prediction or the sensory input, 4) neuromodulator effects on the sensory and prediction weights are simulated, and 5) contraction bias is explained by the model.

While the originally proposed circuit model introduced in the authors' previously published papers provides a valuable framework and the extensions introduced in this manuscript are interesting, I am yet unconvinced whether this work adds a sufficient amount of novelty and more in-depth insights on top of the authors' previously published works, for the following reasons:

1) In my opinion, the proposed model provides one possible explanation of how cortical circuits can estimate the uncertainty of sensory stimuli and predictions. This point was raised in the previous round of reviews, and in response, the authors added a discussion on two other possible minor modifications of the proposed circuit model in the Supplementary Discussion as well as testing the effects of varying parameters such as time constants and connection weights on the M and V neurons in the model.

However, this does not eliminate alternative mechanisms or circuit structures that can carry out similar computations. Testing all different possible circuit mechanisms, of course, may not be realistic, but given that the proposed model is constructed purposefully to output the desired effects without a sufficient ground on experimental data, simply proposing this one circuit mechanism and slight variations of it limits the impact of this work.

2) Relatedly, the proposed circuit model and the conclusions from the model simulations are not sufficiently motivated. First, given how the model is structured, it is not surprising to see the desired effects such as the M and V neurons encoding the mean and the variance, or the stimulus and trial statistics determining the sensory & prediction weights. Secondly, some of the key components of the model lack biological grounds. The motivation of the proposed model will be much stronger, for example, if the suggested neuromodulator effects are indeed hinted in some experimental studies, or if the variance- and mean-encoding neurons and their projections onto other cell populations & higher cortical circuit are supported by biological evidence.

Overall, while this work provides interesting ideas and the revisions have improved it further, the novel contributions made by this work in addition to the authors' previous publications are rather limited.

Here are some other minor suggestions/comments:

1) Supporting Information- B. Supporting analyses:

References to Supplementary Figure 12 in B2 and B3 should be changed to Supplementary Figure 13.

2) Supplementary Figure 5:

Please show the actual mean and the actual variance on panels A & B for clarity.

3) Supporting Discussion:

Line 208 "In addition, we include VIP neurons that are known to receive top-down inputs and to provide disinhibition.":
Please provide a reference supporting this.

(Remarks on code availability)

I haven't reviewed the code extensively but it seems the code is available in the provided Github repository.

Reviewer #4

(Remarks to the Author)

For me, the paper was still quite difficult to read. Equations are still explained in words, which doesn't make sense -- to understand what's going on, one has to turn the words into equations, so why not start with the equations? Or at least make them easily accessible. It's true that there's a figure in SI, but it didn't help me that much.

In addition, it was pretty much impossible, from the main text, for me to figure out what problem they were addressing. The focus was on prediction when they in fact meant prior, which is utterly confusing for a theorist; "prediction" does not equate to prior.

Fortunately, I stumbled across Secs. B1 and A1.3 in SI, and I think (but am still not 100% sure) that I figured out what was going on. (Those sections were in the original submission, but I didn't notice them.) Sec. B1 told us what the network did: it low-pass filtered the sensory input to get the mean, and then subtracted the mean from the sensory input, squared it, and low-pass filtered that to get the variance. The mean was then passed to the next network, where the process was repeated -- thus producing the mean and variance over trials of the sensory input. This formed the prior.

Section A1.3 then told us that on each trial, the means were combined with the variances to produce, on each trial, the minimum variance estimate of the sensory input.

It would have been very helpful to have that in the main text.

Presumably the whole point of the network, which was pretty complicated, was to make this all biologically plausible?

In any case, now that I know what's going on, I can review the science. Here I think there are problems.

1. The goal is standard Bayesian inference: the brain acquires the mean and variance of some sensory signal, and that's combined with the prior (also characterized by a mean and variance) to provide an optimal estimate of the sensory input.

But details matter.

The sensory signal consists of multiple presentations. More precisely, letting x_t be the sensory signal at time t , the sensory signal is x_1, x_2, \dots, x_n . The x_t are drawn from a distribution with mean μ and variance σ^2 . What the circuit does is estimate μ and σ^2 .

That seems problematic to me: given n independent samples from a distribution, the variance is σ^2/n , not σ^2 . It's not totally clear that the brain can do that kind of averaging, but it's almost for sure true that the brain can estimate the mean with lower variance than σ^2 . So in that respect the setup is not very realistic.

Much more realistic is that a single sensory input tells you both the mean and variance (when you hear a sound or see an image, typically you immediately know how uncertain you are about it). For repeated multiple stimuli, figuring out what variance the brain estimates is highly nontrivial, but almost for sure it's not the raw variance of the x_t .

2. Low pass filtering a signal to get the mean works only if the timescale of the filter is long compared to the timescale of the variability in the sensory input. In particular, if the sensory input varies on a timescale long compared to the timescale of the filter, it will track the input almost perfectly and yield no variance at all. Given that the variability is likely to have multiple timescales, having a single filter timescale seems like a bad idea.

3. Combining likelihood with prior (which is essentially what they did) requires the relative weights to depend on the variance. To set those weights, they plugged in the computed variance (Eq. 7 of SI). Biologically, it's pretty unrealistic to have activity affect synaptic weight in just the right way. Given that this paper is all about biological realism, that seems to be a serious problem.

4. The memory neuron has linear tuning, something that's highly unusual -- most neurons have "bump" (as for orientation) or sigmoidal tuning curves. There's a reason for that: neurons don't have much dynamic range (even at the population level). As far as I can tell, their scheme will not apply to more standard population tuning curves.

5. Performing Bayesian inference (on a range of problems, including the simple case here of combining the likelihood with

the prior) has a long history in this field. The authors alluded to it on lines 47-48, where they say "However, how the variance of both the sensory input and the prediction can be computed on the circuit level is not resolved yet". Although it's not resolved, it's not for lack of trying. There's a pretty big literature on this (which mainly consists of sampling -- Langyel and colleagues -- and probabilistic population codes -- Pouget and colleagues). Essentially none of it was referenced. The work in this paper needs to be placed in the context of that previous work.

(Remarks on code availability)

Version 2:

Reviewer comments:

Reviewer #3

(Remarks to the Author)

In this second revision, the authors have mainly updated the text for clarification. In particular, they included an additional discussion on alternative network architectures and biological grounds of the proposed circuit model. The added discussion on the experimental evidence is especially helpful and informative, as they provide speculations on the location and identity of the proposed M and V neurons. This update has alleviated my previous concerns regarding biological relevance. My only remaining concern, however, is on the novelty and additional contributions of this work in comparison to the authors' previous published works (Hertag & Clopath, 2022; Hertag & Sprekeler, 2020) as noted in both the first and the second reviews. As the authors point out, I do recognize that the current manuscript made several new extensions to the previous circuit model, such as incorporating M and V neurons for computing the mean and the variance of the sensory inputs and the model predictions of neuromodulator effects on weighting of sensory inputs and predictions. I am not sure, however, whether these would be sufficient innovations compared to the authors' previous papers; yet, I think this work is a solid and interesting study suggesting a possible circuit mechanisms for computing the uncertainty using prediction-error circuits.

(Remarks on code availability)

I only briefly looked through the code and have not run the code. The repository seems to include all the necessary code though.

Reviewer #4

(Remarks to the Author)

OK, I think I finally understand what's going on. Mainly because I talked to two of the authors, Claudia Clopath and Loreen Hertag. After they explained it, I realized it's all in the paper. But not where I was expecting, and somewhat spread out.

But first, a general comment: this paper was more about the circuit than the algorithm, and that part was good. So, although this seems like kind of a long review, it shouldn't be taken as negative; more as guidance on how to make the paper better.

So let me summarize. Subjects observe a stimulus, $s(t)$, which has the following behavior:

- $s(t)$ consists of a series of steps
- each step lasts 500 ms
- the height of each step is drawn from $N(\mu, \sigma^2)$.
- every 5 seconds μ is redrawn from a uniform distribution.

The goal is to estimate μ . So basically subjects get 10 noisy samples at each μ .

This is a classic change detection task. Which is possibly why I was so confused (and why comment 1 in my previous review was pretty far off): nowhere in the paper is change detection mentioned. In fact, line 40 says "A common hypothesis is that the brain weights different inputs according to their reliabilities". Technically that's true, but it's a bit misleading given the prior literature in the field (OK, I was misled), for which the major emphasis has been on an uncertain signal and a prior.

By the way, almost exactly this task has been considered before; see Krishnamurthy et al., Nature Human Behaviour 2017,

<https://www.nature.com/articles/s41562-017-0107>

And the task was analyzed theoretically by Meijer et al., bioRxiv 2024,

<https://www.biorxiv.org/content/10.1101/2024.10.29.620874v1>).

Both those papers should be referenced.

Given that this is a change detection task, we can ask: is the algorithm used by the authors a reasonable one? For their algorithm, they updated two sets of variables, corresponding to a lower and higher level circuit. Using L and H for lower and

higher, and M and V for mean and variance, the update rules were

lower:

$$\tau_{LM} dr_{LM}/dt = -r_{LM} + s(t)$$

$$\tau_{LV} dr_{LV}/dt = -r_{LV} + (r_{LM}(t) - s(t))^2$$

higher:

$$\tau_{HM} dr_{HM}/dt = -r_{HM} + r_{LM}(t)$$

$$\tau_{HV} dr_{HV}/dt = -r_{HV} + (r_{HM}(t) - r_{LM}(t))^2$$

The estimate of the mean, μ_{hat} , is

$$\mu_{hat} = (s(t)/r_{LV} + r_{LM}(t)/r_{HV}) / (1/r_{LV} + 1/r_{HV}).$$

According to the equation for μ_{hat} ,

- the stimulus, $s(t)$, is weighted more heavily when the variance of the lower circuit is small compared to the variance of the higher circuit.

- the filtered version of the stimulus, $r_{LM}(t)$, is weighted more heavily when it's the other way around.

Given this, if the circuit is close to optimal, we would expect the variance of the lower circuit to be small compared to the variance of the higher circuit right after a change. In fact, I believe it's the other way around: because the higher circuit lags the lower circuit, the variance of the lower circuit goes up faster than the variance of the higher circuit after a change. So this algorithm seems backwards: it weights the stimulus less heavily after a change, not more heavily.

This paper was more about the circuit than the algorithm, and that part was good, so this isn't the end of the world. But I don't think it would be a great idea to pass this off as semi-optimal (on lines 36-7 the authors ask "And how do neural networks in the brain combine both input streams wisely?"). The authors could simply admit that their algorithm is suboptimal. (With a caveat: assuming it is suboptimal; it's possible that I'm totally wrong.) But this doesn't seem like a great idea. A better alternative, in my opinion, is to use a more optimal algorithm. There is (to my knowledge) no closed form solution to this problem, so all reasonable algorithms are approximate. Here are a couple suggestions:

1. Filter the signal at two timescales. The difference between the two filters provides information about whether or not there has been a change: if the two filters are about the same, the output should be a filtered version of the stimulus; otherwise, the output should rely much more heavily on the stimulus. Note, though, that the variance neurons disappear.
2. Use the variance as an indicator of change. When the variance goes up, rely more heavily on $s(t)$; when it's not so large, rely more heavily on $r_{LM}(t)$. Not sure if you need the higher circuit any more, but maybe it can be used.
3. And there are a lot of other options; see Meijer et al., referenced above.

Alternatively, they could show that their algorithm actually is a good one (see caveat above).

In addition, I have a some minor suggestions.

a. In the intro, be clear that you're modeling a change detection task. The stairs example doesn't seem to fall into this category. And, given the timescales, this seems relevant to cognitive tasks; if you framed it as a cognitive task from the beginning, it would make a lot more sense. At least to me.

b. I would strongly suggest having, somewhere in the paper a succinct description of the model, as given above, along with the parameters used in all the simulations. In particular, it was hard to go from the Table 1 to the time constants, which are critical for understanding what's going on. I personally would put this in the main text (you have already started with the Box), but I'll leave that up to you.

c. What are the dark lines in Figs. 2D and E? I don't believe they were described in the figure caption. But I might have missed it.

(Remarks on code availability)

Version 3:

Reviewer comments:

Reviewer #3

(Remarks to the Author)

In this revision, the main update is Supplementary Fig 10 where the authors performed additional simulations comparing

alternative weighted outputs. These additional results show that different strategies of weighing can result in distinct reconstruction results under different noise conditions. This is a nice addition to the manuscript. The authors also mention that the manuscript is novel as it adds a hierarchical extension and introduces M and V neurons to the previously proposed circuits.

I am still not entirely convinced that these are sufficiently significant extensions of the models proposed in the authors' previous works. I do not think further simulations or modifications to the model are needed for this manuscript at this point, as my concerns are more on the general impact and novelty, which of course, is just my opinion. Nevertheless, I agree that the manuscript explores interesting additional mechanisms and simulations that make the circuit model more holistic, which is a valuable contribution.

(Remarks on code availability)

I briefly looked at the Github repository and it includes a README file with instructions and code files. I have not installed and run the code.

Reviewer #4

(Remarks to the Author)

I'm happy with the changes!

(Remarks on code availability)

Thank you very much for the thorough and considerate review of our manuscript. We have made changes to both text and figures (in the main manuscript and the Supplementary Information) according to the comments and suggestions.

To address all comments, we have changed Figs. 4 and 5, as well as Supplementary Figs. 6, 7, 8, and 9 (now Supplementary Figs. 11, 12, 13 and 14), and added five new supplementary figures (now labeled Supplementary Figs. 1, 2, 4, 5 and 6). We also added a new section to our Supplementary Information ('Supplementary Discussion') and the Results ('A circuit model for uncertainty estimation'). Furthermore, to provide easier access to the PE circuits used, we added 5 Supplementary Data files that contain the connectivities of the mean-field networks tested (Supplementary Data 1-3), the connectivity (Supplementary Data 4) and the gain factors (Supplementary Data 5) for the multi-cell population model.

Below, we answer the questions from the reviewers (in bold) and outline in detail the changes we made. To simplify the process for the reviewers, we have copied the major text changes from the main manuscript (italics) and, in contrast to the numerical references of Nature Communications, used an in-text author/year citation style in this response letter, with a reference list at the end of this document. Furthermore, in consideration of the overall comments from all reviewers and following the Nature Communication guidelines, we have made additional modifications where we felt that more clarity or conciseness was needed.

The manuscript has benefited greatly from the reviewers' comments. We thank the reviewers again for their work. We are looking forward to your responses.

Best regards,
Loreen Hertäg, Katharina A. Wilmes & Claudia Clopath

Detailed responses

Reviewer #1

In this manuscript entitled "Knowing what you don't know: Estimating the uncertainty of feedforward and feedback inputs with prediction-error circuits", the authors propose a microcircuit model for how excitatory neurons and inhibitory interneurons compute prediction errors as well as estimate the reliabilities of the feedforward "sensory" input and the feedback "predictive" signal.

The main building block of this model, extracting positive and negative prediction errors from a randomly time-varying signal, has already been described in previously published studies. This manuscript propose several new extensions. First, the activity of the positive and negative "predictive coding" neurons are combined by two units, one estimating the mean, and the second the variance. Second, two such microcircuits are stacked hierarchically, the second performing the same operation but at a slower time scales on the estimate generated by the first. And finally, the output of the two microcircuits are combined in a "Bayesian" manner by weighting each according to the estimated variance. Note that this last part is analytical, and not performed by the circuit. Finally, they show that neuromodulators acting on different components of this circuit would bias inference differently.

A powerful motivation for this approach is that such microcircuit could be an ubiquitous component for implementing hierarchical Bayesian inference in brain circuits. However, I have some serious issues with the manuscript, as listed below.

1. **First, one essential component of the circuit, computing positive and negative prediction errors, has already been published. Apart from referring to this other article, the authors do not give any specifics on the connection weights or why this architecture**

was chosen. As a result, we do not have the rationale behind the circuit organization (e.g. what is the role of VIP neurons, why two PV neurons, or a separate dendritic compartment for excitatory neurons?). Moreover, one cannot replicate these results based on the information provided.

We thank the reviewer for the comment as it highlights that we have not explained the model and the rationale behind it sufficiently. We have extensively revised the main text and added a section to the Suppl. Info to address this comment. More precisely, we have rewritten/phrased and structured the first part of the Results to give more space to the network model. In line 77 of the Results, we start with the section *A circuit model for uncertainty estimation* that contains the information needed to understand the architecture, rationale, and motivation for the circuit model:

We hypothesize that the distinct response patterns of negative and positive prediction-error (nPE/pPE) neurons can act as a backbone for estimating the mean and the variance of sensory stimuli. An nPE neuron only increases its activity relative to a baseline when the sensory input is weaker than predicted, while a pPE neuron only increases its activity relative to a baseline when the sensory input is stronger than predicted. Moreover, both nPE and pPE neurons remain at their baseline activities when the sensory input is fully predicted (Fig. 1B).

To test our hypothesis, we study a rate-based mean-field network: the core network contains two excitatory neurons, two inhibitory parvalbumin-expressing (PV) interneurons, one inhibitory somatostatin-expressing (SOM), and one inhibitory vasoactive intestinal peptide-expressing (VIP) interneuron (Fig. 1B, also see Supplementary Fig. 1). While the excitatory neurons are simulated as two coupled point compartments to emulate the soma and dendrites of elongated pyramidal cells, respectively, all inhibitory cell types were modeled as point neurons. In line with experimental findings (Tremblay et al., 2016), we assume that the PV neurons target the somatic compartment, while the SOM neuron targets the dendritic compartment of the excitatory cells. Moreover, the SOM neuron inhibits both the PV and VIP neurons, while the VIP neuron inhibits both the PV and the SOM neurons (Tremblay et al., 2016) (Fig. 1B, also see Supplementary Fig. 1). In addition, all neurons receive local connections from the excitatory neurons (Supplementary Fig. 1).

We chose the connection strengths in line with our previous work on prediction-error neurons (see Hertäg and Sprekeler, 2020; Hertäg and Clopath, 2022, and Methods). In that work, we showed that response patterns of excitatory cells resemble those of PE neurons when a number of excitatory (E) and inhibitory (I) pathways onto the pyramidal cells were balanced. This multi-pathway E/I balance results in an E/I balance of the inputs to excitatory neurons when the stimulus is perfectly predicted. Depending on the network connectivity, for some excitatory cells, this input E/I balance was preserved for over-predicted stimuli (sensory input < prediction) while it was temporarily broken in favor of excitation for under-predicted stimuli (sensory input > prediction). In contrast, for some excitatory cells, the responses for over- and under-predicted stimuli were reversed. While the former are pPE neurons, the latter are nPE neurons.

The multi-pathway E/I balance required for PE neurons to emerge was established through the different interneurons. These interneurons provide compartment-specific inhibition to balance the feedforward sensory inputs and the feedback predictions, respectively (for a more detailed discussion on the role of these interneurons in PE circuits, please see Supplementary Discussion). In the present work, we use a PE circuit in which the soma of the excitatory cells, the SOM neuron and one of the PV neurons receives the feedforward sensory input, while the other cells/compartments receive the prediction thereof (other networks were tested in Fig. 4). This is in line with experimental work showing that feedback connections hypothesized to carry information about expectations or predictions (Mumford, 1992; Larkum, 2013; Friston, 2008) target the apical dendrites of pyramidal cells (Larkum, 2013) and interneurons located in superficial layers of the cortex (see, e.g. Tremblay et al., 2016).

We reasoned that if a prediction of a stimulus is the mean of the previously experienced stimuli, it can be modeled through a perfect integrator (here denoted memory neuron) that receives connections from the PE neurons (Fig. 1C). More precisely, following Keller and Mrsic-Flogel (2018), we assume that the pPE neuron excites the memory neuron, while the nPE neuron inhibits this neuron (for instance, through lateral inhibition, here not modeled explicitly). If the activity of the memory neuron is below the sensory input, the pPE neuron is active while the nPE neuron is silent (Supplementary Fig. 2).

Hence, the memory neuron receives more excitation. If the activity of the memory neuron is above the sensory input, the nPE neuron is active while the pPE neuron is silent (Supplementary Fig. 2). As a consequence, the memory neuron receives more inhibition. When the memory neuron is roughly at the mean of the sensory inputs, occasionally being below or above, the number of times the nPE and pPE neuron are active balances. Hence, the PE neurons ensure that the memory neuron’s activity does not drift too far from the mean. In our network, the memory neuron is consistent with the idea of internal representation neurons in predictive processing theories (Bastos et al., 2012; Keller and Mrsic-Flogel, 2018). The neural substrate for these internal representation neurons have been hypothesised to be deep-layer 5 (L5) neurons (Bastos et al., 2012; Heindorf and Keller, 2022). Also, more recently, a group of excitatory L2/3 neurons have been shown to integrate over negative and positive prediction errors (O’Toole et al., 2022).

The memory neuron in our network projects back to the PE neurons it receives inputs from. We therefore call the input from the memory neuron to all other neurons in the PE circuit feedback input. Please note that the activity of the memory neuron, here considered a prediction, could also be interpreted as a prior of the sensory stimulus.

If the prediction equals the mean of the sensory stimulus, the activity of the nPE and pPE neuron encode the deviation from the mean. Thus, the squared sum of nPE and pPE neuron activity represents the variance of the feedforward input (provided that the PE neurons are silent without sensory stimulation). We, therefore, simulate a downstream neuron (termed V neuron), modeled as a leaky integrator with a quadratic activation function, that receives excitatory synapses from the PE neurons (see the lower-level subnetwork in Fig. 1C, the higher-level circuit is described later).

We have furthermore added a section in the Supporting Information to discuss the function of the interneurons in our network:

We include three types of inhibitory interneurons in our network: PV, SOM and VIP interneurons. Generally speaking, these interneurons are required to establish nPE and pPE neurons by balancing the multiple pathways the sensory inputs and predictions can take through the network (Hertäg and Clopath, 2022). More precisely, the PV neurons establish an E/I balance at the soma of the excitatory neurons, while the SOM neurons establish an E/I balance at the dendrites of the same neurons. In addition, we include VIP neurons that are known to receive top-down inputs and to provide disinhibition. These VIP neurons and more importantly the connections they make with other interneurons are necessary to ensure that the E/I balance required for PE neurons is met not only for fully predicted sensory inputs but also for one of the mismatches (sensory input < prediction, or sensory input > prediction).

In our mean-field network, we include only one cell per interneuron type except for PV neurons which we assumed to be represented by two neurons. The reason for that is that we have previously shown (Hertäg and Clopath, 2022) that only one source of somatic inhibition is not sufficient to give rise to both nPE and pPE neurons in the same network in which the dendrites are balanced for fully predicted sensory inputs and during one of the two mismatch phases (for more information, please see Hertäg and Clopath, 2022). To account for that, we included two PV neurons, one receiving the sensory inputs while the other one receiving the prediction thereof.

Please note though that other network architectures are possible. For instance, we can develop PE circuits with less inhibitory interneuron types but those networks would require more constraints on the distribution of sensory inputs and predictions among the remaining interneurons. This, however, might not be in line with the rich spectrum of inputs neurons receive in biological networks. Moreover, some of the constraints on the interneuron circuit can be relaxed if we only require nPE and pPE neurons to exhibit an E/I balance for fully predicted sensory inputs and allow them to be over-inhibited in one of the mismatch phases.

Beyond the text changes, we added a new figure that shows in detail the network architecture (Supplementary Fig. 1), and added more information to the Methods (see Table 1, Supplementary Data 1-5). Moreover, we made sure that the GitHub is publicly accessible so that all simulations and figures can be replicated.

2. **Second, the variance-computing neurons do not, generally speaking, measure a variance. More exactly, they do so if and only if the sensory signals are generated exactly as the author describe.**

The reviewer is right that we mainly focus on a specific input scenario. However, we also tried different input distributions (now Supplementary Fig. 3), showed that it holds for a multi-cell population model (now Supplementary Fig. 7), and also considered a case in which the network has to estimate a spatial variance. In addition, we now also validate our network by using a continuous input signal instead of a piecewise constant input (Supplementary Fig. 4), discuss the influence of baseline activities in PE neurons (Supplementary Fig. 5), and investigate the robustness of the results by changing several network parameters (Supplementary Fig. 8).

The effect of a continuous input signal (Supplementary Fig. 4) is addressed on line 163: *Moreover, using a continuously changing signal instead of a piecewise constant stimulus yields similar results, where small deviations can be attributed to the PE neurons not reaching their steady state (Supplementary Fig. 4).*

The effect of baseline activities (Supplementary Fig. 5) and the robustness of the results in general are addressed on line 166:

While the results do not strongly depend on the stimulus statistics and distribution, they are affected by the baseline activities of the PE neurons that were assumed to be zero in our network, in line with the low baseline firing rates reported for neurons in primary visual cortex of rodents (Polack et al., 2013; Xue et al., 2014). When the baseline rate of the nPE neuron is increased, the memory neuron underestimates the mean of the sensory input (Supplementary Fig. 5A). In contrast, when the baseline rate of the pPE neuron is increased, the memory neuron overestimates the mean of the sensory input (Supplementary Fig. 5A). However, increasing the baseline for both PE neurons by the same amount does not affect the estimation of the stimulus mean (Supplementary Fig. 5A). In contrast, a non-zero baseline in any of the PE neurons yields an overestimation of the stimulus variance (Supplementary Fig. 5B). This suggests inhibitory interneurons must cancel the baseline activity to ensure an unbiased uncertainty estimation in networks with high-baseline PE neurons. While the baseline activity of PE neurons can bias the estimation of mean and variance, other network connection strengths and neuron properties play a less pivotal role (see Supplementary Fig. 6 and discussed in the Supplementary Discussion).

Furthermore, these new results are also mentioned on line 260: *We have shown that the baseline activity of PE neurons can affect the ability of the M and V neuron to encode the mean and the variance of the feedforward input, respectively. In the full network, these biases manifest in a sensory weight that is slightly pushed towards 0.5 (Supplementary Fig. 2C). That is, in an initially sensory-driven regime, the dependence on the sensory inputs is slightly weakened. In contrast, in an initially prediction-driven regime, the dependence on the sensory inputs is slightly strengthened. Similarly, while other properties like the connectivity between the PE neurons and the M/V neurons can affect the estimation of the mean and the variance, the sensory weight is only slightly affected if the changes occur in both the lower- and the higher-level circuit (Supplementary Fig. 6)*

Furthermore, in the SI, we discuss the results of the robustness analysis further on line 227:

We showed that the lower-level subnetwork can correctly estimate the mean and the variance of the sensory inputs for different stimulus implementations (Supplementary Fig. 4), statistics (Fig. 2) and distributions (Supplementary Fig. 3). Moreover, the results hold for a multi-cell population model used to estimate the sensory uncertainty over time (Supplementary Fig. 7) or space (Supplementary Fig. 8). Furthermore, we showed that the baseline firing rate of nPE and pPE neurons must rather be small or canceled by respective interneurons to ensure an unbiased estimation of the sensory uncertainty (Supplementary Fig. 5).

We further investigated how the estimation of the mean and the variance is influenced by connection strengths and neuron properties (Supplementary Fig. 6). Scaling the connections from the PE neurons onto the memory neuron only affects the speed at which the mean is reached (Supplementary Fig. 6B). If the connections from the M neuron are scaled, the mean is encoded in the product of the memory neuron's activity times this connection strength (Supplementary Fig. 6C). If additional

arbitrary top-down input targets the neurons in the PE circuit that also receive synapses from the memory neuron, the memory neuron encodes the mean reduced by this top-down input (Supplementary Fig. 6E). In contrast, the V neuron is mainly affected by its time constant and the connection it receives from the PE neurons. While the time constant determines how much the estimated variance fluctuates around the true variance, scaling the connections onto the V neuron biases the variance estimation (Supplementary Fig. 6A & D).

In the full network, these biases manifest in a sensory weight that is slightly pushed towards 0.5 (Supplementary Fig. 6). That is, in a former sensory-driven regime, the dependence on the sensory inputs is slightly weakened. In contrast, in a former prediction-driven regime, the dependence on the sensory inputs is slightly strengthened. Similarly, while other properties like the connectivity between the PE neurons and the M/V neurons can affect the estimation of the mean and the variance, the sensory weight is only slightly affected if the changes occur in both the lower- and the higher-level circuit.

We hope that this convinces the reviewer that the network indeed can estimate the variance more generally than it was depicted in our first version of the manuscript.

3. **Third, despite the authors mentioning feedback in the title, there is no feedback connection in the model (at least between the two hierarchical levels). Similarly, prediction errors are usually understood as the difference between a sensory signal and a top-down prediction. This is not the case here. The estimates from the two hierarchical levels are combined offline, not implemented in the circuit, which is misleading. More generally, this circuit lacks generality (it works only for a specific type of signal modulation and noise, in the absence of any prior or “top down” information). The title is an overstatement.**

We apologize that our writing was not clear and even misleading. To outline our thinking a bit further: We consider the case in which the prediction is the mean of the sensory input. In our network, this prediction/mean is encoded in the memory neuron which feeds back to the neurons in the PE circuit it receives its inputs from. We, hence, decided to use the term feedback when we talked about the inputs from the memory neuron onto the neurons of the lower-order PE circuit. To make that clear in the text, we now revised the introduction of the memory neuron extensively (line 121-139) and add a sentence that clarifies what we mean by “feedback” (line 140): *The memory neuron in our network projects back to the PE neurons it receives inputs from. We therefore call the input from the memory neuron to all other neurons in the PE circuit feedback input. Please note that the activity of the memory neuron, here considered a prediction, could also be interpreted as a prior of the sensory stimulus.*

Moreover, to ensure that our title does not evoke false hopes, we erased the terms “feedforward” and “feedback” from our title because we agree with the reviewer these terms are indeed widely used and their meanings can differ in different contexts. We also note that there is a difference between the terms “uncertainty” (which is the variance in our case) and “reliability” (which is a function of the uncertainties). While we do compute the variance in our network, we do not implement the reliabilities explicitly, and hence combine the sensory input and the prediction (encoded in the lower-level M neuron) offline, as correctly stated by the reviewer. However, our title only claims that we estimate the uncertainties and not the reliabilities. This is why we feel the title after removing the terms “feedforward” and “feedback” is not an oversell anymore and hope the reviewer is ok with it. However, we are happy to discuss alternative titles if the current one is still inadequate.

To highlight even more that the weighted output of the network is computed offline, we added more words/sentences to emphasize this point. Line 215: *For the sake of simplicity, we assume that the weighted output is encoded in a separate class of neurons not explicitly modeled here, and only compute the sensory weight arithmetically.* We also down-toned the statements in the subsequent paragraph, lines 218 - 229.

Regarding the generality and the top-down input: We refer to the reviewer’s concern #2 above where we outline our efforts to show that the network can also perform for different parameters

and input scenarios. Moreover, while we do not consider a "general" top-down input and only a more local prediction encoded in the lower-level M neuron, we now show in the revised manuscript that additional additive input (that could be interpreted as top-down input) targeting the neurons which would also receive the prediction from the M neuron, does not impair the network's ability to estimate the uncertainties (see Supplementary Fig. 6 E). In such a scenario, the M neuron would encode the mean reduced by the amount of top-down input present. Overall, the top-down input and the activity of the M neuron would constitute the mean of the sensory input, and hence, the uncertainty estimation would not be affected. However, we note that this is the case if the top-down input targets the neurons that receive the synapses from the M neuron.

4. **I have also more minor issues. First, while there is experimental evidence for negative and positive prediction error coding neurons in sensory areas, it is not so much the case for variance computing neurons. Neurons specifically tuned to variance have been reported in high level or "associative" areas (such as prefrontal cortex) in tasks where uncertainty plays a crucial role. To my knowledge, variance-tuned neurons have never observed in sensory areas, where the dominant effect of changing stimulus variance is to change response gains through adaptation.**

We do agree with the reviewer that the evidence for variance computing neurons in sensory areas is not present to date. We make that clear in the Discussion on line 457: *However, neurons encoding the variance in primary sensory areas is a prediction of our model that still needs to be validated. It has been shown though that stimulus uncertainty can be encoded in the gain variability of individual neurons recorded from V1 and V2 of macaque monkeys (Hénaff et al., 2020).*

5. **Second, I think the reported simulation results for neuromodulation are not useful. It is unsurprising that perturbing the circuit leads to biases in variance estimates. However, no simple, testable or non-trivial prediction seems to emerge from that. In addition, the figures are hard to read because of overlap between the different symbols.**

We thank the reviewer for this comment as it highlights that we needed to revise the text and the figure on the section about neuromodulators. We have now changed the main figure (Fig. 4) to more clearly show the results and highlight the main messages (see Fig. 4C). In addition, we also changed the corresponding supporting figures (now Supplementary Figs. 11 and 12). We revised the main text extensively (see lines 273 - 338) to capture those changes.

Finally, we also note that not all perturbations lead to biases in the variance estimates. When SOM and VIP neurons are perturbed by the same amount, the variance estimation is not or only slightly affected because of the underlying E/I balance in the PE circuits.

Reviewer #2

Hertäg and colleagues describe a model based around a prediction-error circuit that is an elegant reflection of biological local excitatory and inhibitory circuits that function to estimate the uncertainty in sensory input and determine when the use of sensory input is balanced by predictive processing. The work adds to a growing number of incredibly biologically relevant circuit models that have proven to be highly versatile for forming predictions of how sensory integration and predictive processing would work in the brain. The paper is well written and describes the dynamics of the prediction-error circuits as well as making a number of testable predictions regarding their modulatory dynamics that will be of high interest not only for the field on computational neuroscience, but for sensory neurophysiologists and systems neuroscientists in general. While the complexity of the computational modelling may limit the overall general audience, the discussion is very accessible and relates the model and the findings of the current study to a number of important neuroscientific phenomena and supporting literature. Following are some specific comments largely related to the general limits of the model and the implementation of the initial state. It is possible that these details are available in previous publications developing the basis of the model, the code directly or specified in the methods section, but I would still recommend that some additional brief information be added to the main text in this regard to increase accessibility to a wider audience.

1. **Lines 86-87:** ‘Thus, the squared sum of nPE and pPE neuron activity represents the variance of the feedforward input (provided that the PE neurons are silent without sensory stimulation)’. It is unclear if this means that the baseline activity would be zero (see related comments below). If the baseline were not zero, is it possible to evaluate how a potential bimodal modulation of the PE elements affect the model?

We thank the reviewer for this comment. We indeed did not talk about the baseline activities in the PE neurons sufficiently. To address this, we now include a SI Figure (Supplementary Fig. 5) that shows the impact of the baseline of nPE and pPE neurons on the estimation of mean and variance (and ultimately on the sensory weight). We discuss the baseline now in the text, line 166:

While the results do not strongly depend on the stimulus statistics and distribution, they are affected by the baseline activities of the PE neurons that were assumed to be zero in our network, in line with the low baseline firing rates reported for neurons in primary visual cortex of rodents (Polack et al., 2013; Xue et al., 2014). When the baseline rate of the nPE neuron is increased, the memory neuron underestimates the mean of the sensory input (Supplementary Fig. 5A). In contrast, when the baseline rate of the pPE neuron is increased, the memory neuron overestimates the mean of the sensory input (Supplementary Fig. 5A). However, increasing the baseline for both PE neurons by the same amount does not affect the estimation of the stimulus mean (Supplementary Fig. 5A). In contrast, a non-zero baseline in any of the PE neurons yields an overestimation of the stimulus variance (Supplementary Fig. 5B). This suggests inhibitory interneurons must cancel the baseline activity to ensure an unbiased uncertainty estimation in networks with high-baseline PE neurons.

and also line 260:

We have shown that the baseline activity of PE neurons can affect the ability of the M and V neuron to encode the mean and the variance of the feedforward input, respectively. In the full network, these biases manifest in a sensory weight that is slightly pushed towards 0.5 (Supplementary Fig. 2C). That is, in an initially sensory-driven regime, the dependence on the sensory inputs is slightly weakened. In contrast, in an initially prediction-driven regime, the dependence on the sensory inputs is slightly strengthened.

2. **Lines 92-92:** ‘While the excitatory neurons are simulated as two coupled point compartments to emulate the soma and dendrites of elongated pyramidal cells, respectively, all inhibitory cell types were modeled as point neurons’. Where there any spatial specification of the inhibitory interneuron populations? Perhaps with connectivity of the PV interneurons ‘closer’ or more highly connected to the soma and SOM to the dendrites for instance? Or did all interneurons influence the same compartments? This is indicated in the schematic in Figure 1B and alluded to in lines 102-105, but then it is unclear in the main text how this is precisely implemented. Similarly, the M and V neurons only had single compartments then without dynamics related to compartments?

We thank the reviewer for this comment as it highlights that we needed to explain the rationale and the architecture of the model better. To do that, we have now added a new section in our Results that specifically explains the model (“A circuit model for uncertainty estimation”). We did not have any spatial dependencies in our model but the interneurons do target different compartments. This is now further explained in the first part of the Results, lines 79 - 143:

We hypothesize that the distinct response patterns of negative and positive prediction-error (nPE/pPE) neurons can act as a backbone for estimating the mean and the variance of sensory stimuli. An nPE neuron only increases its activity relative to a baseline when the sensory input is weaker than predicted, while a pPE neuron only increases its activity relative to a baseline when the sensory input is stronger than predicted. Moreover, both nPE and pPE neurons remain at their baseline activities when the sensory input is fully predicted (Fig. 1B).

To test our hypothesis, we study a rate-based mean-field network: the core network contains two excitatory neurons, two inhibitory parvalbumin-expressing (PV) interneurons, one inhibitory somatostatin-expressing (SOM), and one inhibitory vasoactive intestinal peptide-expressing (VIP) interneuron (Fig. 1B, also see Supplementary Fig. 1). While the excitatory neurons are simulated as two coupled point compartments to emulate the soma and dendrites of elongated pyramidal cells, respectively, all inhibitory cell types were modeled as point neurons. In line with experimental findings (Tremblay et al., 2016), we assume that the PV neurons target the somatic compartment,

while the SOM neuron targets the dendritic compartment of the excitatory cells. Moreover, the SOM neuron inhibits both the PV and VIP neurons, while the VIP neuron inhibits both the PV and the SOM neurons (Tremblay et al., 2016) (Fig. 1B, also see Supplementary Fig. 1). In addition, all neurons receive local connections from the excitatory neurons (Supplementary Fig. 1).

We chose the connection strengths in line with our previous work on prediction-error neurons (see Hertäg and Sprekeler, 2020; Hertäg and Clopath, 2022, and Methods). In that work, we showed that response patterns of excitatory cells resemble those of PE neurons when a number of excitatory (E) and inhibitory (I) pathways onto the pyramidal cells were balanced. This multi-pathway E/I balance results in an E/I balance of the inputs to excitatory neurons when the stimulus is perfectly predicted. Depending on the network connectivity, for some excitatory cells, this input E/I balance was preserved for over-predicted stimuli (sensory input < prediction) while it was temporarily broken in favor of excitation for under-predicted stimuli (sensory input > prediction). In contrast, for some excitatory cells, the responses for over- and under-predicted stimuli were reversed. While the former are pPE neurons, the latter are nPE neurons.

The multi-pathway E/I balance required for PE neurons to emerge was established through the different interneurons. These interneurons provide compartment-specific inhibition to balance the feedforward sensory inputs and the feedback predictions, respectively (for a more detailed discussion on the role of these interneurons in PE circuits, please see Supplementary Discussion). In the present work, we use a PE circuit in which the soma of the excitatory cells, the SOM neuron and one of the PV neurons receives the feedforward sensory input, while the other cells/compartments receive the prediction thereof (other networks were tested in Fig. 4). This is in line with experimental work showing that feedback connections hypothesized to carry information about expectations or predictions (Mumford, 1992; Larkum, 2013; Friston, 2008) target the apical dendrites of pyramidal cells (Larkum, 2013) and interneurons located in superficial layers of the cortex (see, e.g. Tremblay et al., 2016).

We reasoned that if a prediction of a stimulus is the mean of the previously experienced stimuli, it can be modeled through a perfect integrator (here denoted memory neuron) that receives connections from the PE neurons (Fig. 1C). More precisely, following Keller and Mrsic-Flogel (2018), we assume that the pPE neuron excites the memory neuron, while the nPE neuron inhibits this neuron (for instance, through lateral inhibition, here not modeled explicitly). If the activity of the memory neuron is below the sensory input, the pPE neuron is active while the nPE neuron is silent (Supplementary Fig. 2). Hence, the memory neuron receives more excitation. If the activity of the memory neuron is above the sensory input, the nPE neuron is active while the pPE neuron is silent (Supplementary Fig. 2). As a consequence, the memory neuron receives more inhibition. When the memory neuron is roughly at the mean of the sensory inputs, occasionally being below or above, the number of times the nPE and pPE neuron are active balances. Hence, the PE neurons ensure that the memory neuron's activity does not drift too far from the mean. In our network, the memory neuron is consistent with the idea of internal representation neurons in predictive processing theories (Bastos et al., 2012; Keller and Mrsic-Flogel, 2018). The neural substrate for these internal representation neurons have been hypothesized to be deep-layer 5 (L5) neurons (Bastos et al., 2012; Heindorf and Keller, 2022). Also, more recently, a group of excitatory L2/3 neurons have been shown to integrate over negative and positive prediction errors (O'Toole et al., 2022).

The memory neuron in our network projects back to the PE neurons it receives inputs from. We therefore call the input from the memory neuron to all other neurons in the PE circuit feedback input. Please note that the activity of the memory neuron, here considered a prediction, could also be interpreted as a prior of the sensory stimulus. [...]

We have furthermore added a section in the Supporting Information to discuss the function of the interneurons in our network. As this is not the main focus of our paper and can also be found in our previous publication, we decided to keep it short and put it in the SI:

We include three types of inhibitory interneurons in our network: PV, SOM and VIP interneurons. Generally speaking, these interneurons are required to establish nPE and pPE neurons by balancing the multiple pathways the sensory inputs and predictions can take through the network (Hertäg and Clopath, 2022). More precisely, the PV neurons establish an E/I balance at the soma of the

excitatory neurons, while the SOM neurons establish an E/I balance at the dendrites of the same neurons. In addition, we include VIP neurons that are known to receive top-down inputs and to provide disinhibition. These VIP neurons and more importantly the connections they make with other interneurons are necessary to ensure that the E/I balance required for PE neurons is met not only for fully predicted sensory inputs but also for one of the mismatches (sensory input < prediction, or sensory input > prediction).

In our mean-field network, we include only one cell per interneuron type except for PV neurons which we assumed to be represented by two neurons. The reason for that is that we have previously shown (Hertäg and Clopath, 2022) that only one source of somatic inhibition is not sufficient to give rise to both nPE and pPE neurons in the same network in which the dendrites are balanced for fully predicted sensory inputs and during one of the two mismatch phases (for more information, please see Hertäg and Clopath, 2022). To account for that, we included two PV neurons, one receiving the sensory inputs while the other one receiving the prediction thereof.

Please note though that other network architectures are possible. For instance, we can develop PE circuits with less inhibitory interneuron types but those networks would require more constraints on the distribution of sensory inputs and predictions among the remaining interneurons. This, however, might not be in line with the rich spectrum of inputs neurons receive in biological networks. Moreover, some of the constraints on the interneuron circuit can be relaxed if we only require nPE and pPE neurons to exhibit an E/I balance for fully predicted sensory inputs and allow them to be over-inhibited in one of the mismatch phases.

Beyond the text changes, we added a new figure that shows in detail the network architecture (Supplementary Fig. 1), and added more information to the Methods (see Table 1, Supplementary Data 1-5).

3. **How was the baseline activity of the pPE and nPE neurons established and is it relevant? Presumably these must be set to the same values to balance some initial state. Also in relation to the interneurons, does the baseline activity have an impact on the modulation via interneurons in the model. Perhaps relatedly, is the baseline activity of the pPE and nPE neuron always the same – can a neuron be considered to have a predictive initial state that changes under different circumstances in the model (e.g. changing M activity after initial encoding, after it approaches the mean of the sensory inputs)?**

Regarding how the baseline activities are set: All neurons of the PE circuit receive an external background input to ensure reasonable firing rates in the absence of any stimulation. These inputs were computed such that the baseline firing rates were 4/s for each interneuron and 0/s for the PE neurons (the PE neurons only fire if there is a mismatch). The exact values for the interneurons are not critical, and were simply chosen in line with our previous paper. However, the baseline of the PE neurons are important and, as pointed out in the #1 comment of the reviewer, are required to be zero to ensure an unbiased mean and variance estimation. To ensure an unbiased mean estimation, nPE and pPE neurons can have a baseline activity greater than zero but must be equal (see new Supplementary Fig. 5). However, the variance estimation would be biased when one or both PE neurons have a baseline activity greater than zero. This suggests that there must be interneurons (targeting the V neuron) that can balance the baseline activity of the PE neurons.

Regarding baselines & neuromodulation: This is a very good question. While the sensory weight might therefore be biased to begin with (that is, without neuromodulation), we do not expect the changes in the sensory weight upon neuromodulation to be *qualitatively* altered by the baseline activities. Baseline activities for the nPE and pPE neurons would increase the baseline activities of the INs (that were set to 4/s in our network). Upon neuromodulation, the targeted interneuron would still change the baseline and the gain of the nPE/pPE neuron/s in the same way. However, we note that the changes in sensory weight may be affected *quantitatively* by the presence of PE neuron baseline activities. Moreover, if the baseline rates would push the network into a regime in which the rate rectifications (that is non-linearities) play a crucial role, the results may indeed also change qualitatively. However, we do not expect this to happen in our network because increasing the baselines of the PE neurons should rather push the network further away from the rectification regime.

Regarding the predictive state: This is an intriguing suggestion. When the neuromodulator targeting the INs changes the baseline and gain of the PE neurons, the activity of the M neuron also changes. Hence, the effect of neuromodulators goes beyond the changes in the sensory weight (that is, the uncertainty estimation) and also affects the predictive state of the network. In other words, the presence of a neuromodulator may also change the prediction itself. We have now addressed this on line 336:

Furthermore, we note that the presence of a neuromodulator can also affect the predictive state of the network. That is, the M neuron activity can also be modulated by changes in the baseline and gain of the PE neurons.

4. **For Figure 3E and the testable finding of the model: ‘sensory predictions influence neural activity more significantly in experiments that rely on fast stimulus changes’, presumably there is no lower limit integration time in the model – as in, the sensory weight is high at time zero. Also, it is unclear what values/limits correspond to T and T/5. Some intuition about this would be helpful.**

This is indeed correct, the statement does only apply to the ”steady state”, that is, a prediction had been established before. At the beginning of an experiment (the very first trial), the sensory input is more important. To account for that, we change the sentence to [...] *sensory predictions, once established, influence neural activity more significantly in experiments that rely on fast stimulus changes. This is because when the stimulus is only present for a short time before changing again in the next trial, these changes can be considered as input noise, so the system relies more strongly on the prediction.*

Moreover, we now explicitly state the trail duration in the text, line 240: *This is further confirmed in simulations in which the trail duration was shortened from 5s to 1s.*

5. **The justification for the three different PE circuits in Figure 4 (denoted by different markers) is unclear. Why these particular circuit parameters and combination of sensory input? Some additional information comes in the very nicely written discussion – but some clarity in the results would also help the interpretation here.**

Thank you for this important comment. We have now completely revised the section on the effect of neuromodulators and Fig. 4 to motivate the three mean-field networks tested. This is now addressed more explicitly at the beginning of the section, lines 273 - 292:

The brain’s flexibility and adaptability are supported by a plethora of neuromodulators which influence the activity of neurons in a variety of ways (Avery and Krichmar, 2017). A prominent target of neuromodulatory inputs is inhibitory neurons (Cardin, 2019; Hattori et al., 2017; Swanson and Maffei, 2019). Moreover, distinct interneuron types are differently (in-)activated by those neuromodulators (Wester and McBain, 2014; Hattori et al., 2017; Swanson and Maffei, 2019). Given that the interneurons in our network play a crucial role in establishing the PE neurons that, in turn, are the backbone for computing the uncertainties, we wondered if and how the weighting of sensory inputs and predictions may be biased when neuromodulators activate distinct interneuron types.

To this end, we modeled the presence of a neuromodulator by injecting an additional excitatory input into an interneuron type (while a neuromodulator can also suppress neuronal activity, we focus on the more common excitatory effects that have been described). Given that the interneurons are embedded in a network and establish an E/I balance in the PE neurons through multiple pathways, we reasoned that not only the interneuron type but also the connections it receives/makes determine the effect of a neuromodulator. Hence, testing only one instantiation of our network may yield effects that do not generalize to other parameterisations of the network. To avoid that, we tested three different mean-field networks derived in (Hertäg and Clopath, 2022). These networks differ in the distribution of sensory inputs and predictions onto the interneurons, and, hence, the underlying connectivity. They cover a broad range of possible PE circuits. The only commonality across those networks is that they exhibit an E/I balance of excitatory and inhibitory pathways onto the PE neurons.

6. **Minor comment: - The direction of modulation in lines 55-57 seem to be backwards for the specified PE types.**

We have re-read carefully the lines 55-57 (*"negative PE (nPE) neurons only increase their activity when the prediction is stronger than the sensory input, while positive PE (pPE) neurons only increase their activity when the prediction is weaker than the sensory input."*) and cannot find the mistakes. Maybe it is a misunderstanding. Could the reviewer let us know more specifically what may be wrong here.

Reviewer #3

In this manuscript, Hertag and colleagues use a rate-based circuit model composed of different cell types to propose a possible mechanism explaining how uncertainty of feedforward sensory stimuli and feedback predictions can be estimated within a cortical circuit. Specifically, the circuit model builds on a model proposed in the authors' previous work (Hertag & Clopath 2022 "Prediction-error neurons in circuits with multiple neuron types: formation, refinement, and functional implications") which includes an excitatory, negative prediction error neuron (nPE), a positive prediction error neuron (pPE), and inhibitory PV, SOM, and VIP neurons, by adding a memory (M) neuron and a downstream variance (V) neuron. The model is wired and tuned so that the nPE responses increase when the prediction is stronger than the sensory input, and pPE responds strongly when the prediction is weaker than the sensory input. The M neuron represents the top-down prediction and relays feedforward signals from the lower to the higher PE circuit. It receives direct excitatory inputs from pPE neuron while being indirectly inhibited by nPE neuron. The V neuron receives excitatory inputs from both PE neurons. The key findings of this study can be summarized as follows:

- In this circuit model, the M and V neurons approach the mean and the variance of the sensory inputs, respectively.
 - By modulating the trial and stimulus statistics, the circuit weighs either the prediction or the sensory input more in its output.
 - Neuromodulator effects simulated on each of the cell types produce different shifts in the weighting of sensory inputs and predictions.
 - Contraction bias can be explained by the model's estimation of stimulus and trial variability.
1. Understanding the computational roles of diverse cell-types in a biologically motivated circuit has been attracting a lot of attention in neuroscience recently. Furthermore, explicating their functional implications in a predictive coding framework is undoubtedly a very interesting topic in the field. However, there are some concerns regarding novelty and contributions of this work, which need to be addressed before the manuscript can be accepted for publication. The main concern is that it is rather questionable whether this work provides sufficiently more in-depth insights in addition to the authors' previous papers, Hertag & Clopath 2022 "Prediction-error neurons in circuits with multiple neuron types: formation, refinement, and functional implications" and Hertag & Sprekeler 2020 "Learning prediction error neurons in a canonical interneuron circuit". Secondly, while the model provides a plausible explanation for how the uncertainty of feedforward and feedback inputs can be computed in a neural circuitry, the circuit model itself is formulated precisely to produce the desired outcomes. For example, based on how the sensory input weight α is defined, it is not surprising that when the variance is high, then the weight on the sensory input decreases. Overall, the contribution of this work is rather limited to providing a circuit model for uncertainty estimation, which can be tested experimentally in the future but for now is still hypothetical. I believe the contribution of the manuscript would increase significantly if the authors also investigate some other alternative circuit models, and systematically evaluate the plausibility of each circuit model. Some additional comments are detailed in the following.

We indeed use the connectivity for the PE circuits from our previous work in which we showed how nPE and pPE neurons can emerge in cortical circuits with several inhibitory interneurons. However, while we make use of the previously derived PE circuit, we here go beyond those networks.

The focus of this work was to suggest and investigate a network that can dynamically estimate the mean of the perceived stimuli (which we interpret as a prediction) and the variance of those inputs by utilizing the unique response patterns of nPE and pPE neurons. Furthermore, we wanted to show that when hierarchically combined, these modules may be able to compute not only the variance of the sensory input but also the prediction thereof, thus, laying the foundation for combining these different inputs flexibly. Hence, we suggest a circuit-level implementation for the dynamic estimation of uncertainties. Moreover, we examine how neuromodulators targeting the different interneurons in the PE circuit may affect uncertainty estimation. This allows us to make experimentally testable predictions: 1) We show that the weighting would not change when a neuromodulator targets both SOM and VIP neurons. 2) The network would be biased towards prediction if a global neuromodulator targets predominately PV neurons, while 3) the network tends to weight the sensory input and prediction more equally when VIP neurons are predominately targeted by a global neuromodulator (this is now summarized in Fig. 4C). Finally, we propose that the weighting of sensory inputs and predictions as described here can be used to explain the contraction bias.

We showed that our results do generalize for different input distributions (now Supplementary Fig. 3), hold for a multi-cell population model (now Supplementary Fig. 7), and spatial uncertainty estimation (now Supplementary Fig. 8). To address the reviewer’s concern about robustness, we performed additional simulations. We validated that our network performs well for a continuous input signal instead of a piecewise constant input (Supplementary Fig. 4), discuss the influence of baseline activities in PE neurons (Supplementary Fig. 5), and investigate the robustness of the results by changing several network parameters (Supplementary Fig. 8).

The effect of a continuous input signal (Supplementary Fig. 4) is addressed on line 163: *Moreover, using a continuously changing signal instead of a piecewise constant stimulus yields similar results, where small deviations can be attributed to the PE neurons not reaching their steady state (Supplementary Fig. 4).*

The effect of baseline activities (Supplementary Fig. 5) and the robustness of the results in general are addressed on line 166:

While the results do not strongly depend on the stimulus statistics and distribution, they are affected by the baseline activities of the PE neurons that were assumed to be zero in our network, in line with the low baseline firing rates reported for neurons in primary visual cortex of rodents (Polack et al., 2013; Xue et al., 2014). When the baseline rate of the nPE neuron is increased, the memory neuron underestimates the mean of the sensory input (Supplementary Fig. 5A). In contrast, when the baseline rate of the pPE neuron is increased, the memory neuron overestimates the mean of the sensory input (Supplementary Fig. 5A). However, increasing the baseline for both PE neurons by the same amount does not affect the estimation of the stimulus mean (Supplementary Fig. 5A). In contrast, a non-zero baseline in any of the PE neurons yields an overestimation of the stimulus variance (Supplementary Fig. 5B). This suggests inhibitory interneurons must cancel the baseline activity to ensure an unbiased uncertainty estimation in networks with high-baseline PE neurons. While the baseline activity of PE neurons can bias the estimation of mean and variance, other network connection strengths and neuron properties play a less pivotal role (see Supplementary Fig. 6 and discussed in the Supplementary Discussion).

Furthermore, the results are also mentioned on line 260: *We have shown that the baseline activity of PE neurons can affect the ability of the M and V neuron to encode the mean and the variance of the feedforward input, respectively. In the full network, these biases manifest in a sensory weight that is slightly pushed towards 0.5 (Supplementary Fig. 2C). That is, in an initially sensory-driven regime, the dependence on the sensory inputs is slightly weakened. In contrast, in an initially prediction-driven regime, the dependence on the sensory inputs is slightly strengthened. Similarly, while other properties like the connectivity between the PE neurons and the M/V neurons can affect the estimation of the mean and the variance, the sensory weight is only slightly affected if the changes occur in both the lower- and the higher-level circuit (Supplementary Fig. 6).*

Moreover, in the SI, we discuss the results of the robustness analysis further on line 227: *We showed that the lower-level subnetwork can correctly estimate the mean and the variance of the*

sensory inputs for different stimulus implementations (Supplementary Fig. 4), statistics (Fig. 2) and distributions (Supplementary Fig. 3). Moreover, the results hold for a multi-cell population model used to estimate the sensory uncertainty over time (Supplementary Fig. 7) or space (Supplementary Fig. 8). Furthermore, we showed that the baseline firing rate of nPE and pPE neurons must rather be small or canceled by respective interneurons to ensure an unbiased estimation of the sensory uncertainty (Supplementary Fig. 5).

We further investigated how the estimation of the mean and the variance is influenced by connection strengths and neuron properties (Supplementary Fig. 6). Scaling the connections from the PE neurons onto the memory neuron only affects the speed at which the mean is reached (Supplementary Fig. 6B). If the connections from the M neuron are scaled, the mean is encoded in the product of the memory neuron’s activity times this connection strength (Supplementary Fig. 6C). If additional arbitrary top-down input targets the neurons in the PE circuit that also receive synapses from the memory neuron, the memory neuron encodes the mean reduced by this top-down input (Supplementary Fig. 6E). In contrast, the V neuron is mainly affected by its time constant and the connection it receives from the PE neurons. While the time constant determines how much the estimated variance fluctuates around the true variance, scaling the connections onto the V neuron biases the variance estimation (Supplementary Fig. 6A & D).

In the full network, these biases manifest in a sensory weight that is slightly pushed towards 0.5 (Supplementary Fig. 6). That is, in a former sensory-driven regime, the dependence on the sensory inputs is slightly weakened. In contrast, in a former prediction-driven regime, the dependence on the sensory inputs is slightly strengthened. Similarly, while other properties like the connectivity between the PE neurons and the M/V neurons can affect the estimation of the mean and the variance, the sensory weight is only slightly affected if the changes occur in both the lower- and the higher-level circuit.

We thank the reviewer for suggesting contrasting our model to alternative circuit models. This is an important point and we considered two alternative models which we now discuss in the supporting text (line 251 in SI):

We focus here on one network whose core units are PE circuits. In this network, the nPE and pPE neurons drive both the M and the V neurons, which in turn encode the mean and the variance of the feedforward input (in the lower-level circuit: sensory input, in the higher-level circuit: prediction). This implementation has the advantage, first, of explicit mean and variance representations in excitatory cells, which could be broadcasted to other areas, and second, provides a prediction and an output even in the absence of sensory input. While an exhaustive exploration of different models would be beyond the scope of this work, we will discuss other alternatives in the following.

First of all, we modeled the memory neuron as a perfect integrator. An alternative would be to model it as a leaky integrator with a time constant significantly larger than zero. The leaky memory neuron would directly receive the feedforward input and project to the PE circuit, thereby representing a low-pass filter of the feedforward input. In contrast to our implementation, the memory neuron would be silent without any sensory information. Hence, it can not be interpreted as a "prediction" which should also be available in the absence of sensory evidence. While, in our model, we do need the sensory information to develop a prediction in the first place (here the mean of the sensory input), the input is not necessary after the prediction has been established.

Second, similar to how representation neurons are updated according to the classical predictive coding model by Rao and Ballard (1999), prediction errors from the same level and the level below could be used to calculate what we here call the "weighted output". To ensure Bayes-optimal integration, those errors would need to be weighted based on the variance of each level. In Rao and Ballard (1999), the variance was a fixed parameter. However, in recent work by Wilmes et al. (2023), it has been suggested that the variance could also be dynamically estimated by PV interneurons, which then modulate the prediction error activity to lead to uncertainty-modulated prediction errors.

2. **How pPE and nPE neurons produce responses that depend on the relative strength of prediction compared to stimulus is not clear in this manuscript. While I see that this is described in the cited, previous work of the authors, it will be helpful to include**

some brief description on how pPE and nPE neurons in the model produce the desired response patterns.

Thank you for this suggestion. That is a great idea. We revised the first part of the Results extensively to include a detailed description on the model (section "A circuit model for uncertainty estimation"). Within this section, we also elaborated on the PE neurons, line 100:

[...] We chose the connection strengths in line with our previous work on prediction-error neurons (see Hertäg and Sprekeler, 2020; Hertäg and Clopath, 2022, and Methods). In that work, we showed that response patterns of excitatory cells resemble those of PE neurons when a number of excitatory (E) and inhibitory (I) pathways onto the pyramidal cells were balanced. This multi-pathway E/I balance results in an E/I balance of the inputs to excitatory neurons when the stimulus is perfectly predicted. Depending on the network connectivity, for some excitatory cells, this input E/I balance was preserved for over-predicted stimuli (sensory input < prediction) while it was temporarily broken in favor of excitation for under-predicted stimuli (sensory input > prediction). In contrast, for some excitatory cells, the responses for over- and under-predicted stimuli were reversed. While the former are pPE neurons, the latter are nPE neurons. [...]

- 3. Is there any experimental evidence supporting the existence and the assumed connectivity of the M and V neurons proposed in the model? Are the results from the simulations of neuromodulator effects expected? There are some discussions on biological relevance in Discussions, but it will be helpful to motivate the implementation of M & V neurons and the neuromodulator simulations in the main text.**

To motivate the existence of the M neuron, we now moved the part from the Discussion to the main results, line 135:

In our network, the memory neuron is consistent with the idea of internal representation neurons in predictive processing theories (Bastos et al., 2012; Keller and Mrsic-Flogel, 2018). The neural substrate for these internal representation neurons have been hypothesised to be deep-layer 5 (L5) neurons (Bastos et al., 2012; Heindorf and Keller, 2022). Also, more recently, a group of excitatory L2/3 neurons have been shown to integrate over negative and positive prediction errors (O'Toole et al., 2022).

As variance-encoding neurons in sensory areas are not reported and are therefore an experimentally testable prediction from the model, we left this in the Discussion, and added a sentence, line 457:

However, neurons encoding the variance in primary sensory areas is a prediction of our model that still needs to be validated. It has been shown though that stimulus uncertainty can be encoded in the gain variability of individual neurons recorded from V1 and V2 of macaque monkeys (Hénaff et al., 2020).

Moreover, to motivate the connectivity and the rationale behind the M and V neurons better, we added (and edited) the following to the main part (including the new Supplementary Fig. 2):

We reasoned that if a prediction of a stimulus is the mean of the previously experienced stimuli, it can be modeled through a perfect integrator (here denoted memory neuron) that receives connections from the PE neurons (Fig. 1C). More precisely, following Keller and Mrsic-Flogel (2018), we assume that the pPE neuron excites the memory neuron, while the nPE neuron inhibits this neuron (for instance, through lateral inhibition, here not modeled explicitly). If the activity of the memory neuron is below the sensory input, the pPE neuron is active while the nPE neuron is silent (Supplementary Fig. 2). Hence, the memory neuron receives more excitation. If the activity of the memory neuron is above the sensory input, the nPE neuron is active while the pPE neuron is silent (Supplementary Fig. 2). As a consequence, the memory neuron receives more inhibition. When the memory neuron is roughly at the mean of the sensory inputs, occasionally being below or above, the number of times the nPE and pPE neuron are active balances. [...]

Furthermore, we completely revised Figure 4 and the corresponding SI figures (Supplementary Figs. 11 and 12), as well as the text on the effect of perturbing the INs (lines 273 - 338). We hope that the new version does capture the motivation behind the simulations and the use of different mean-field networks. However, we decided to leave the comparison with experimental results on neuromodulators in the Discussion because we feel that it is rather speculative.

4. Why do the authors use discrete sequences of sensory stimuli, rather than continuous stimulus signals and modulating the temporal and spatial variability?

Thank you for this important comment. The reason for why we originally used piecewise constant stimuli is that we use the PE circuits derived in Hertäg and Clopath (2022). Those PE circuits were derived for steady state stimuli and are hence only strictly mathematically valid in the steady state. However, to follow the reviewer’s suggestion, we performed additional simulations and found that the network still performs very well for continuously changing inputs. To show that, we included a new SI Figure (Supplementary Fig. 4) for which we filtered our piecewise constant stimuli with different smoothing windows. Even for very large smoothing windows, the network estimates the mean and the variance well. We address this now on line 163:

Moreover, using a continuously changing signal instead of a piecewise constant stimulus yields similar results, where small deviations can be attributed to the PE neurons not reaching their steady state (Supplementary Fig. 4).

While we do mainly focus on temporal variability, we also show an example in which the network is used to compute the spatial variability. However, to keep the network simple, we decided to separate the example for spatial variability from the ones with temporal variability.

5. The Figure 4C is motivated on lines 226-228 “To disentangle the effect of nPE and pPE neurons, we perturbed those neurons individually in both the lower or higher subnetwork by injecting either an inhibitory or excitatory additional input.”, but it seems to show that perturbations of both nPE and pPE introduce exactly the same effect on V neuron, so the claim of “disentangling the effect of nPE and pPE neurons” can be misleading.

That is very true. We revised Fig. 4 and the corresponding text completely. The part cited here has been removed as it indeed did not help to disentangle the effect of nPE and pPE neurons further.

6. Is the weighted output hypothesized to be represented by a separate class of neurons?

Yes, we do assume that it is computed in a separate class of neurons. To state that more clearly, we added the following sentence (line 215):

For the sake of simplicity, we assume that the weighted output is encoded in a separate class of neurons not explicitly modeled here, and only compute the sensory weight arithmetically.

7. The authors have proposed a mechanism underlying contraction bias using a similar model in their previous work (Hertäg & Clopath 2022, Fig 5). Some clarification on the novelty and description that distinguish these two models of contraction bias should be included to signify the result in the manuscript.

Absolutely! Thank you for bringing this up. We now added a paragraph in the Discussion (line 428):

Finally, we illustrate that the weighted integration of feedforward and feedback inputs can be interpreted as a neural manifestation of the contraction bias. We have recently proposed another network which implements a neuronal contraction bias (Hertäg and Clopath, 2022). The main difference is that in the previous paper, the output neuron received not only the sensory input but also synapses from the nPE and pPE neurons. While the nPE neurons excited the output neuron, the pPE neuron inhibited it. This way, the contraction bias was a direct result of the opposite connectivity between the PE neurons and the output neuron, and the amount of bias was dictated by the connection strength between those neurons. Hence, the bias is independent of the uncertainties of the sensory input or the prediction, and only scales with the absolute difference between both inputs. In contrast, in our current model, we assume that the network output would not receive direct input from the PE neurons but the prediction itself, and that both the sensory input and the prediction is weighted by their reliabilities (Fig. 5). However, we note that we only consider a neural representation of the stimulus and do not account for other sources of noise, like execution noise, that surely impacts the contraction bias observed in behavioral studies.

8. What is the motivation to use an integrator with indirect inhibition from nPE for M neuron, and a leaky integrator with quadratic activation for V neuron?

We used a perfect integrator for the M neuron because the prediction (here the mean of the sensory stimuli) should also be present in the absence of a sensory stimulus. A leak term would eventually

alter the memory when the sensory stimulus is removed. To explain in more detail how nPE and pPE neurons establish the mean in the M neuron, we added a SI Figure (Supplementary Fig. 2) and a new paragraph in the Results (line 123):

[...] More precisely, following Keller and Mrsic-Flogel (2018), we assume that the pPE neuron excites the memory neuron, while the nPE neuron inhibits this neuron (for instance, through lateral inhibition, here not modeled explicitly). If the activity of the memory neuron is below the sensory input, the pPE neuron is active while the nPE neuron is silent (Supplementary Fig. 2). Hence, the memory neuron receives more excitation. If the activity of the memory neuron is above the sensory input, the nPE neuron is active while the pPE neuron is silent (Supplementary Fig. 2). As a consequence, the memory neuron receives more inhibition. When the memory neuron is roughly at the mean of the sensory inputs, occasionally being below or above, the number of times the nPE and pPE neuron are active balances. Hence, the PE neurons ensure that the memory neuron's activity does not drift too far from the mean. [...]

Here we do not assume that the variance of the sensory input or the prediction needs to be remembered. This is why we decided to include a leak term. For nPE and pPE neurons to establish the variance in the leaky neuron, both must excite the V neuron. However, please note that we could have also used a perfect integrator for the V neuron.

- 9. In the hierarchical model of PE circuits, feedforward signals from the lower PE circuit to the higher PE circuit are relayed only via the M neuron which encodes the mean sensory inputs/expectations. However, in the traditional framework of hierarchical predictive coding, error signals (represented in PE neurons) are transmitted from the lower to the higher areas as feedforward signals. Some comments on this discrepancy would be helpful.**

We thank the reviewer for this important comment. We have now added a short section on this discrepancy in the Discussion (line 464):

We suppose that the lower-level prediction is forwarded to the higher-level subnetwork. This architecture deviates from classical hierarchal predictive coding frameworks in which the prediction errors are sent up the hierarchy (Rao and Ballard, 1999). However, we are not the first to consider this alternative model. For instance, Sprattling (2008) reformulated the predictive coding model by Rao and Ballard (Rao and Ballard, 1999) in terms of a biased competition model in which the prediction instead of the prediction errors are forwarded. However, we note that whether prediction errors or predictions are forwarded between different subnetworks is not mutually exclusive and might differ between brain areas or species.

- 10. The authors modeled the excitatory PE neurons as two-compartmental models representing the soma and the dendrites of pyramidal cells. In a recent paper by Gillon et al (2024, J Neuroscience), the dendrites and the soma of pyramidal neurons show distinct response patterns to expectation violation, which can be relevant for the two-compartmental model proposed in this work.**

Indeed, we should have related our work to this paper. We now do so in the Discussion (line 475):

However, we note that whether prediction errors are encoded in the (spiking) activity of separate neurons and/or in the local voltage dynamics of the dendrites is still an active field of research (Mikulasch et al., 2023). In a recent paper by Gillon et al. (2024), it has been shown that pattern-violation signals are differently processed at the soma and the dendrites over time. This suggests that the compartments of an excitatory neuron play a more intricate function in predictive processing than we can account for in our simplified neuron model.

- 11. Figure 4 is a bit confusing. Fig4A markers have different colors noting sensory weights, but this does not add any new information as the y-axis already denotes Sensory Weight.**

Thank you for pointing that out, the color in 4A was indeed redundant information. We have now revised Figure 4 completely.

- 12. Typo on Line 258, "trail-averaged" → "trial-averaged"**

Corrected, thank you.

13. **Typo on Line 379, “trail-to-trial” → “trial-to-trial”**

Corrected, thank you.

14. **Typo in Supporting Information, Line133, “trail” → “trial”**

Corrected, thank you.

15. **The legend describing colors for “in lower PE circuit” and “in higher PE circuit” of Supplementary Fig. 8 is confusing.**

We changed the legend to make it easier to understand.

16. **While the colors and shapes of the data points in Supplementary Fig. 9 are provided in the main text, it will be nice to have the legend included in Supplementary Fig 9 as well.**

We included now a legend to indicate what the colors and markers mean.

17. **In Fig 5D, the schematics showing different distributions of stimuli seems to be misleading. Shouldn’t a schematic of Gaussian distributions with different widths be shown?**

Thank you for pointing that out. In Figure 5D the illustration was supposed to show a uniform distribution with almost zero width and below two Gaussians with different (non-zero) widths. The Gaussians, however, were pretty broad, so it was hard to recognize them as Gaussian distributions. We changed this to avoid confusion.

18. **(Remarks on code availability): The Github repository provided in the URL is not publicly accessible.**

Thank you for trying to download the code. We apologize for the inconvenience you experienced. The GitHub repository had been set to “public” before submission. To ensure that the repo is indeed publicly accessible we now asked colleagues to use the link provided in the manuscript to download and fork the repo. They confirmed that it is reachable via the link and can be downloaded/forked. If you still face difficulties, we would be happy to send the code in separate files.

Reviewer #4

The authors considered a scenario in which there is an incoming scalar sensory stimulus, s , which has some mean and variance (over trials). The job of the brain is to compute that mean and variance. The mean is what the authors call the prediction. When a new (noisy) stimulus comes in, the brain can infer its mean by combining the prediction and the current stimulus, weighted by their respective variances.

I think that’s right, but I admit that I don’t have a huge amount of confidence in. Assuming it is, though, I have a couple of issues with the paper.

1. **What the authors call a “prediction” is really just a prior over the sensory stimulus, s . That’s fine; it’s certainly something the brain uses. But I don’t think it’s what most people would call prediction. And it confused me for a long time. It would have been a lot easier if they just said the network was computing a prior; then I would have instantly known what the model was supposed to do.**

Thank you for pointing that out. We originally refrained from replacing the term prediction with prior because we had felt that the term prediction is appealing to a broader neuroscience audience. However, we agree that the connection with a prior needs to be highlighted and therefore included a sentence in the first section of the Results (line 140):

The memory neuron in our network projects back to the PE neurons it receives inputs from. We therefore call the input from the memory neuron to all other neurons in the PE circuit feedback input. Please note that the activity of the memory neuron, here considered a prediction, could also be interpreted as a prior for the sensory stimulus.

2. **Trying to explain the network in words doesn't work, especially given that the words are somewhat vague (see lines 97-108). So I looked at Methods. After a couple hours I was able to sketch what the circuit looked like. Which wasn't all that complicated; it's not clear why the authors didn't do that (Figs. 1B-D are kind of close, but, in my opinion, were missing a lot of important detail).**

Thank you for pointing this out. We realized that the description of the network was indeed difficult to follow and that the schematics provided in Figure 1 were not sufficient to understand the rationale behind our network. We have now revised the text on the network completely (line 88) and added a new SI figure that shows all connections in the network (Supplementary Fig. 1):

[...] *To test our hypothesis, we study a rate-based mean-field network: the core network contains two excitatory neurons, two inhibitory parvalbumin-expressing (PV) interneurons, one inhibitory somatostatin-expressing (SOM), and one inhibitory vasoactive intestinal peptide-expressing (VIP) interneuron (Fig. 1B, also see Supplementary Fig. 1). While the excitatory neurons are simulated as two coupled point compartments to emulate the soma and dendrites of elongated pyramidal cells, respectively, all inhibitory cell types were modeled as point neurons. In line with experimental findings (Tremblay et al., 2016), we assume that the PV neurons target the somatic compartment, while the SOM neuron targets the dendritic compartment of the excitatory cells. Moreover, the SOM neuron inhibits both the PV and VIP neurons, while the VIP neuron inhibits both the PV and the SOM neurons (Tremblay et al., 2016) (Fig. 1B, also see Supplementary Fig. 1). In addition, all neurons receive local connections from the excitatory neurons (Supplementary Fig. 1).*

We chose the connection strengths in line with our previous work on prediction-error neurons (see Hertäg and Sprekeler, 2020; Hertäg and Clopath, 2022, and Methods). In that work, we showed that response patterns of excitatory cells resemble those of PE neurons when a number of excitatory (E) and inhibitory (I) pathways onto the pyramidal cells were balanced. This multi-pathway E/I balance results in an E/I balance of the inputs to excitatory neurons when the stimulus is perfectly predicted. Depending on the network connectivity, for some excitatory cells, this input E/I balance was preserved for over-predicted stimuli (sensory input < prediction) while it was temporarily broken in favor of excitation for under-predicted stimuli (sensory input > prediction). In contrast, for some excitatory cells, the responses for over- and under-predicted stimuli were reversed. While the former are pPE neurons, the latter are nPE neurons.

The multi-pathway E/I balance required for PE neurons to emerge was established through the different interneurons. These interneurons provide compartment-specific inhibition to balance the feedforward sensory inputs and the feedback predictions, respectively (for a more detailed discussion on the role of these interneurons in PE circuits, please see Supplementary Discussion). In the present work, we use a PE circuit in which the soma of the excitatory cells, the SOM neuron and one of the PV neurons receives the feedforward sensory input, while the other cells/compartments receive the prediction thereof (other networks were tested in Fig. 4). This is in line with experimental work showing that feedback connections hypothesized to carry information about expectations or predictions (Mumford, 1992; Larkum, 2013; Friston, 2008) target the apical dendrites of pyramidal cells (Larkum, 2013) and interneurons located in superficial layers of the cortex (see, e.g. Tremblay et al., 2016). [...]

We hope that the revised text and the SI figure are sufficient to understand the network.

3. **Based on the equations in Methods, two things jumped out at me. First, the prediction was put into a "memory neuron", rM, which was a perfect integrator. That seems a bit strange, since it would seem hard for the memory neuron to equilibrate at the mean value of the stimulus. It managed to (see Fig. 2D), but I can't figure out why. An explanation would be very useful here. In particular, did it require fine tuning?**

Thank you for raising that question. We added a new paragraph (and a new Supplementary Fig. 2) that more precisely explains how the nPE and pPE neuron establish the mean in the memory neuron (line 121):

We reasoned that if a prediction of a stimulus is the mean of the previously experienced stimuli, it can be modeled through a perfect integrator (here denoted memory neuron) that receives connections from the PE neurons (Fig. 1C). More precisely, following Keller and Mrosovsky (2018), we assume that

the pPE neuron excites the memory neuron, while the nPE neuron inhibits this neuron (for instance, through lateral inhibition, here not modeled explicitly). If the activity of the memory neuron is below the sensory input, the pPE neuron is active while the nPE neuron is silent (Supplementary Fig. 2). Hence, the memory neuron receives more excitation. If the activity of the memory neuron is above the sensory input, the nPE neuron is active while the pPE neuron is silent (Supplementary Fig. 2). As a consequence, the memory neuron receives more inhibition. When the memory neuron is roughly at the mean of the sensory inputs, occasionally being below or above, the number of times the nPE and pPE neuron are active balances. Hence, the PE neurons ensure that the memory neuron's activity does not drift too far from the mean. [...]

Furthermore, to show it did not require fine-tuning, we investigate how the activities of M and V neurons depend on network parameters (Supplementary Fig. 6) and neuron properties (Supplementary Fig. 5) and describe that further in the text (line 166):

While the results do not strongly depend on the stimulus statistics and distribution, they are affected by the baseline activities of the PE neurons that were assumed to be zero in our network, in line with the low baseline firing rates reported for neurons in primary visual cortex of rodents (Polack et al., 2013; Xue et al., 2014). When the baseline rate of the nPE neuron is increased, the memory neuron underestimates the mean of the sensory input (Supplementary Fig. 5A). In contrast, when the baseline rate of the pPE neuron is increased, the memory neuron overestimates the mean of the sensory input (Supplementary Fig. 5A). However, increasing the baseline for both PE neurons by the same amount does not affect the estimation of the stimulus mean (Supplementary Fig. 5A). In contrast, a non-zero baseline in any of the PE neurons yields an overestimation of the stimulus variance (Supplementary Fig. 5B). This suggests inhibitory interneurons must cancel the baseline activity to ensure an unbiased uncertainty estimation in networks with high-baseline PE neurons. While the baseline activity of PE neurons can bias the estimation of mean and variance, other network connection strengths and neuron properties play a less pivotal role (see Supplementary Fig. 6 and discussed in the Supplementary Discussion).

and line 260:

We have shown that the baseline activity of PE neurons can affect the ability of the M and V neuron to encode the mean and the variance of the feedforward input, respectively. In the full network, these biases manifest in a sensory weight that is slightly pushed towards 0.5 (Supplementary Fig. 2C). That is, in an initially sensory-driven regime, the dependence on the sensory inputs is slightly weakened. In contrast, in an initially prediction-driven regime, the dependence on the sensory inputs is slightly strengthened. Similarly, while other properties like the connectivity between the PE neurons and the M/V neurons can affect the estimation of the mean and the variance, the sensory weight is only slightly affected if the changes occur in both the lower- and the higher-level circuit (Supplementary Fig. 6).

4. **Second, I was expecting the positive and negative prediction error neurons to be differentially driven by the stimulus, s , and the prediction, rM . That is, positive prediction error neurons should increase their firing rate when $rM < s$ and negative prediction error neurons should decrease their firing rate when $rM < s$ (I think; I might have gotten that backwards). However, when I looked at the equations (Eq. 2 in Methods), as far as I could tell, both s and rM positively modulated both types of prediction error neurons. But maybe that's not the case? It was somewhat hard to tell from the explanation, even when I looked in SI. This should be clear, as it seems critical.**

Thank you for this comment. To clarify, pPE neurons increase their activity when the activity of the memory neuron is smaller than the stimulus ($rM < s$) but remain at their baseline when the stimulus is perfectly predicted ($rM = s$) or when the activity of the memory neuron is larger than the stimulus ($rM > s$). In contrast, nPE neurons increase their activity when the activity of the memory neuron is larger than the stimulus ($rM > s$) but remain at their baseline when the stimulus is perfectly predicted ($rM = s$) or when the activity of the memory neuron is smaller than the stimulus ($rM < s$). This, however, is not reflected in how the sensory input s and the prediction rM modulate these two excitatory neurons (hence, the equations do not differ). More precisely, both nPE and pPE neurons receive the excitatory feedforward input at the somatic compartment and

the prediction at their dendritic compartment. Whether an excitatory neuron exhibits an nPE-like or pPE-like response profile upon predicted and unpredicted sensory stimuli is only determined by the connections it receives from the interneuron circuit it is embedded in. These interneurons also receive s and rM . Hence, the net effect of s and rM on the nPE or pPE neuron can differ depending on the connection strengths within the network. We now discuss this in more detail on line 100:

We chose the connection strengths in line with our previous work on prediction-error neurons (see Hertäg and Sprekeler, 2020; Hertäg and Clopath, 2022, and Methods). In that work, we showed that response patterns of excitatory cells resemble those of PE neurons when a number of excitatory (E) and inhibitory (I) pathways onto the pyramidal cells were balanced. This multi-pathway E/I balance results in an E/I balance of the inputs to excitatory neurons when the stimulus is perfectly predicted. Depending on the network connectivity, for some excitatory cells, this input E/I balance was preserved for over-predicted stimuli (sensory input < prediction) while it was temporarily broken in favor of excitation for under-predicted stimuli (sensory input > prediction). In contrast, for some excitatory cells, the responses for over- and under-predicted stimuli were reversed. While the former are pPE neurons, the latter are nPE neurons.

The multi-pathway E/I balance required for PE neurons to emerge was established through the different interneurons. These interneurons provide compartment-specific inhibition to balance the feedforward sensory inputs and the feedback predictions, respectively (for a more detailed discussion on the role of these interneurons in PE circuits, please see Supplementary Discussion). In the present work, we use a PE circuit in which the soma of the excitatory cells, the SOM neuron and one of the PV neurons receives the feedforward sensory input, while the other cells/compartments receive the prediction thereof (other networks were tested in Fig. 4). This is in line with experimental work showing that feedback connections hypothesized to carry information about expectations or predictions (Mumford, 1992; Larkum, 2013; Friston, 2008) target the apical dendrites of pyramidal cells (Larkum, 2013) and interneurons located in superficial layers of the cortex (see, e.g. Tremblay et al., 2016).

5. **Based on Eq. 6, it seems that variance was computed on two timescales. (There were two circuits, low and high, but I never actually got to the high one. Given my uncertainty on the low one, I didn't want to try.) The short timescale was for the stimulus. But that was only 60 ms, which seems way too short; most stimuli last for longer than that. But again, maybe I'm missing something?**

The variance is not computed on different time scales. However, the speeds with which the lower-level M neuron and the higher-level M neuron evolve are different (see line 203 in Results or line 482 in Discussion). Their time constant was set to the time constant of the excitatory neurons (60 ms) but the connection strengths from the PE neurons to the M neurons in each subnetwork are different so that the lower-level M neuron evolves faster than the higher-level M neuron. The V neurons, however, have the same time constant (5 seconds) and the connection strengths from the PE neurons onto the V neuron of each subnetwork are equal, too.

For the full network, each stimulus is composed of 10 piecewise linear constant values shown for 500 time steps each (in the network, 500 time steps could be interpreted as 0.5 seconds). Hence, each stimulus would be 5 seconds long. In total, we showed between 100 and 200 stimuli (each trial one stimulus). We now changed the Methods to clarify this point further (Table 1).

6. **Finally, it wasn't clear to me what the take-home message was. Because the computational problem was easy (compute priors), I assume the take-home message was how a circuit implemented it. But if that's the case, trying to figure out what the circuit was from the paper was quite difficult. What is needed is a clear circuit diagram, with clear explanations. In particular, I would have liked to know why there were so many inhibitory cell types. Was it, for instance, impossible to build a circuit with only one kind of inhibitory neuron? Or were they just trying to match data?**

The reviewer is correct that the main contribution of this work is to suggest and investigate a circuit-level implementation that allows the brain to dynamically estimate the mean of the perceived stimuli and the variance of those inputs by utilizing the unique response patterns of nPE and pPE neurons. Furthermore, we wanted to show that when hierarchically combined, these modules may

be able to compute not only the variance of the sensory input but also the prediction thereof, thus, laying the foundation for combining these different inputs flexibly. In addition, we examine how neuromodulators targeting the different interneurons in the PE circuit can affect uncertainty estimation. And, finally, we propose that the weighting of sensory inputs and predictions as described here can be used to explain the contraction bias.

We agree that the paper did not sufficiently describe the circuit model and hence edited this part extensively. As pointed out in our response to the reviewer’s concern #2, we have now added a new section in the Results (starting line 77) and added two supplementary figures that help to explain the circuit model (Supplementary Fig. 1 shows all elements and connections, Supplementary Fig. 2 shows how the nPE and pPE neurons establish the mean in the M neuron).

We also follow the reviewer’s suggestion to elaborate on the interneurons in the model. We indeed can develop PE circuits with less inhibitory interneuron types but those networks would require more constraints on the distribution of sensory inputs and predictions among the remaining interneurons. This, however, might not be in line with the rich spectrum of inputs neurons receive in biological networks. As we aimed at a representative model of the PE circuits in the brain, which are composed of these diverse cell types, we therefore decided to include the three IN types that have been studied intensively. We now include a paragraph in the Results, line 110:

[...] The multi-pathway E/I balance required for PE neurons to emerge was established through the different interneurons. These interneurons provide compartment-specific inhibition to balance the feedforward sensory inputs and the feedback predictions, respectively (for a more detailed discussion on the role of these interneurons in PE circuits, please see Supplementary Discussion). [...]

and the supporting discussion in the SI (line 203):

We include three types of inhibitory interneurons in our network: PV, SOM and VIP interneurons. Generally speaking, these interneurons are required to establish nPE and pPE neurons by balancing the multiple pathways the sensory inputs and predictions can take through the network (Hertäg and Clopath, 2022). More precisely, the PV neurons establish an E/I balance at the soma of the excitatory neurons, while the SOM neurons establish an E/I balance at the dendrites of the same neurons. In addition, we include VIP neurons that are known to receive top-down inputs and to provide disinhibition. These VIP neurons and more importantly the connections they make with other interneurons are necessary to ensure that the E/I balance required for PE neurons is met not only for fully predicted sensory inputs but also for one of the mismatches (sensory input < prediction, or sensory input > prediction). [...]

7. **(Remarks on code availability): As you can see, I’m not too excited about the paper. There may be something interesting there (although I kind of doubt it), but it was pretty much impossible for me to extract it.**

We hope that by clarifying all the issues above, we can convince the reviewer that the work is interesting and provides new insights.

References

- Michael C Avery and Jeffrey L Krichmar. Neuromodulatory systems and their interactions: a review of models, theories, and experiments. *Frontiers in neural circuits*, 11:108, 2017.
- Andre M Bastos, W Martin Usrey, Rick A Adams, George R Mangun, Pascal Fries, and Karl J Friston. Canonical microcircuits for predictive coding. *Neuron*, 76(4):695–711, 2012.
- Jessica A Cardin. Functional flexibility in cortical circuits. *Current opinion in neurobiology*, 58:175–180, 2019.
- Karl Friston. Hierarchical models in the brain. *PLoS computational biology*, 4(11):e1000211, 2008.
- Colleen J Gillon, Jason E Pina, Jérôme A Lecoq, Ruweida Ahmed, Yazan N Billeh, Shiella Caldejon, Peter Groblewski, Timothy M Henley, Eric Lee, Jennifer Luviano, et al. Responses to pattern-violating visual stimuli evolve differently over days in somata and distal apical dendrites. *Journal of Neuroscience*, 44(5), 2024.

- Ryoma Hattori, Kishore V Kuchibhotla, Robert C Froemke, and Takaki Komiyama. Functions and dysfunctions of neocortical inhibitory neuron subtypes. *Nature neuroscience*, 20(9):1199–1208, 2017.
- Matthias Heindorf and Georg B Keller. Reduction of layer 5 mediated long-range cortical communication by antipsychotic drugs. *bioRxiv*, 2022.
- Olivier J Hénaff, Zoe M Boundy-Singer, Kristof Meding, Corey M Ziemba, and Robbe LT Goris. Representation of visual uncertainty through neural gain variability. *Nature communications*, 11(1):2513, 2020.
- Loreen Hertäg and Claudia Clopath. Prediction-error neurons in circuits with multiple neuron types: Formation, refinement, and functional implications. *Proceedings of the National Academy of Sciences*, 119(13):e2115699119, 2022.
- Loreen Hertäg and Henning Sprekeler. Learning prediction error neurons in a canonical interneuron circuit. *Elife*, 9:e57541, 2020.
- Georg B Keller and Thomas D Mrsic-Flogel. Predictive processing: A canonical cortical computation. *Neuron*, 100(2):424–435, 2018.
- Matthew Larkum. A cellular mechanism for cortical associations: an organizing principle for the cerebral cortex. *Trends in neurosciences*, 36(3):141–151, 2013.
- Fabian A Mikulasch, Lucas Rudelt, Michael Wibral, and Viola Priesemann. Where is the error? hierarchical predictive coding through dendritic error computation. *Trends in Neurosciences*, 46(1):45–59, 2023.
- David Mumford. On the computational architecture of the neocortex. *Biological cybernetics*, 66(3):241–251, 1992.
- Sean M O’Toole, Hassana K Oyibo, and Georg B Keller. Prediction error neurons in mouse cortex are molecularly targetable cell types. *BioRxiv*, pages 2022–07, 2022.
- Pierre-Olivier Polack, Jonathan Friedman, and Peyman Golshani. Cellular mechanisms of brain state-dependent gain modulation in visual cortex. *Nature neuroscience*, 16(9):1331, 2013.
- Rajesh PN Rao and Dana H Ballard. Predictive coding in the visual cortex: a functional interpretation of some extra-classical receptive-field effects. *Nature neuroscience*, 2(1):79, 1999.
- Michael W Spratling. Predictive coding as a model of biased competition in visual attention. *Vision research*, 48(12):1391–1408, 2008.
- Olivia K Swanson and Arianna Maffei. From hiring to firing: activation of inhibitory neurons and their recruitment in behavior. *Frontiers in molecular neuroscience*, 12:168, 2019.
- Robin Tremblay, Soohyun Lee, and Bernardo Rudy. Gabaergic interneurons in the neocortex: from cellular properties to circuits. *Neuron*, 91(2):260–292, 2016.
- Jason C Wester and Chris J McBain. Behavioral state-dependent modulation of distinct interneuron subtypes and consequences for circuit function. *Current opinion in neurobiology*, 29:118–125, 2014.
- Katharina A Wilmes, Mihai A Petrovici, Shankar Sachidhanandam, and Walter Senn. Uncertainty-modulated prediction errors in cortical microcircuits. *bioRxiv*, pages 2023–05, 2023.
- Mingshan Xue, Bassam V Atallah, and Massimo Scanziani. Equalizing excitation–inhibition ratios across visual cortical neurons. *Nature*, 511(7511):596, 2014.

We would like to thank all reviewers for the effort and the numerous valuable suggestions.

To address the comments comprehensively, we have included a box with equations to more intuitively explain the computations performed by our network. Additionally, we have added a new section in our Supplementary Information titled "*Model assumptions, simplifications & limitations*". Furthermore, in consideration of the overall comments from all reviewers and following the Nat Comm guidelines, we have made additional modifications where we felt that more clarity or conciseness was needed. Below, we have addressed all comments in detail and hope that we have resolved all concerns.

Detailed responses

Reviewer #2

The authors should be commended for investing a lot of effort into clarifying their model and to making it generally accessible to a wider scientific audience. They have addressed my concerns and have strengthened the manuscript with substantial additional Figures and associated discussion. I especially appreciate the efforts that went into the additional Supp Fig 1 and also the additional discussions and clarifications regarding the influence of the baseline dynamics.

In relation to the minor comment about lines 55-77, indeed the authors are correct – my confusion arose from cross-referencing this text in the introduction to the accompanying Figure 1B caption text:

Lines 55-57: "negative PE (nPE) neurons only increase their activity when the prediction is stronger than the sensory input, while positive PE (pPE) neurons only increase their activity when the prediction is weaker than the sensory input."

Fig 1B caption: 'The nPE neuron only increases its activity relative to a baseline when the sensory input is weaker than predicted, while the pPE neuron only increases its activity relative to a baseline when the sensory input is stronger than predicted.'

Clearly both are actually correct, but on first reading they seemed contradictory since the prediction and the sensory input are switched in reference to each other. While I see the error now, for a first-time reader this was confusing. Personally, I find the order of the Fig 1B caption text more intuitive, but in general I would just suggest they pick one direction of reference and stick to this at least for the first few references of this relationship, and in the early descriptions of the model, for increased clarity and ease.

We thank the reviewer for this comment. We followed their suggestion and changed it accordingly.

Reviewer #3

The revised manuscript has undergone significant edits for improved clarity and now includes additional simulations. Specifically, some of the major edits for improved clarity include 1) modified visualization of the main Fig 4 and Supplementary Figures 11 & 12, and 2) additional Supplementary Figures 1 & 2 which show the circuit model schematics and the relationship between the M neuron and the nPE/pPE neurons. Regarding additional simulations, the revised manuscript now includes three new Supplementary Figures 4, 5, and 6 on new results. In these newly added Supplementary Figures, the authors performed simulations with continuously varying stimulus signals and tested effects of varying some parameters of the circuit model such as nPE and pPE baselines, V neuron time constant, connection weight scaling, etc. Finally, the revisions include additional discussions as a part of the Supporting Information. These revisions addressed many of the suggestions from the first round of reviews, and the manuscript has improved a lot in clarity. However, in my opinion, the major concerns raised in the first round of reviews remain despite the revisions.

This work is an extension of the authors' previously published papers which formulate the circuit model involving nPE & pPE neurons and inhibitory subtypes. The novel aspects of this manuscript are that 1) the M and V neurons are added to the model which encode the mean and the variance of sensory inputs, respectively, 2) this circuit motifs are hierarchically combined, 3) the trial and stimulus statistics are shown to modulate weights on the prediction or the sensory input, 4) neuromodulator effects on the sensory and prediction weights are simulated, and 5) contraction bias is explained by the model. While the originally proposed circuit model introduced in the authors' previously published papers provides a valuable framework and the extensions introduced in this manuscript are interesting, I am yet unconvinced whether this work adds a sufficient amount of novelty and more in-depth insights on top of the authors' previously published works, for the following reasons:

1. In my opinion, the proposed model provides one possible explanation of how cortical circuits can estimate the uncertainty of sensory stimuli and predictions. This point was raised in the previous round of reviews, and in response, the authors added a discussion on two other possible minor modifications of the proposed circuit model in the Supplementary Discussion as well as testing the effects of varying parameters such as time constants and connection weights on the M and V neurons in the model. However, this does not eliminate alternative mechanisms or circuit structures that can carry out similar computations. Testing all different possible circuit mechanisms, of course, may not be realistic, but given that the proposed model is constructed purposefully to output the desired effects without a sufficient ground on experimental data, simply proposing this one circuit mechanism and slight variations of it limits the impact of this work.

We agree that the proposed model is only one potential mechanism, and others may be possible, too. However, we believe that the current paper adds sufficient novelty for the reasons the reviewer listed above. A systematic and thorough comparison with all or many models would be beyond the scope of this project, as it would significantly expand the paper and, more importantly, shift its focus towards becoming more of a review. However, the reviewer raises an important point and to address this concern more thoroughly, we have expanded our Supplementary Information discussion on alternative mechanisms. We incorporated another proposed method for estimating uncertainty and highlighted the distinct predictions made by each alternative model in comparison to ours (line 262 ff).

Second, we propose that the activity of PE neurons is utilized to encode uncertainty in a downstream neuron. However, alternative models offer different mechanisms for this process. One such alternative suggests that a neuron could independently compute the squared error between the sensory input and its mean. Wilmes et al. (2023) demonstrate that a PV neuron with a quadratic activation function can represent variance in its activity without requiring direct connections from PE neurons. In their model, the incoming weights onto the PV neurons learn to store input variability through a local activity-dependent plasticity rule (Wilmes et al., 2023). Moreover, in the model proposed by Wilmes et al. (2023), the PE itself is modulated by uncertainty, a feature absent in our model.

Another alternative involves estimating variance through an error-minimizing learning rule that reduces the discrepancy between the squared magnitude of errors and a variance estimate (Granier et al., 2023). This model predicts that the error on the variance estimate is encoded by a third class of error neurons, but it does not predict the existence of an excitatory cell type that represents the variance itself, as our model does. Additionally, Granier et al. (2023) propose that error activity occurs in the apical dendrites of the representation neurons. [...]

Please note that we have relocated the paragraph comparing the neural implementation for the contraction bias between this paper and our previous work to the "Alternative network architectures" section, as we felt it fits more appropriately there.

While we hope that this new section on alternative mechanisms resolves the reviewer's concern, we would be happy to extend the part further if the reviewer let us know which models or mechanisms they want us to compare our model with (or implement a specific model).

2. Relatedly, the proposed circuit model and the conclusions from the model simulations are not sufficiently motivated. First, given how the model is structured, it is not

surprising to see the desired effects such as the M and V neurons encoding the mean and the variance, or the stimulus and trial statistics determining the sensory & prediction weights. Secondly, some of the key components of the model lack biological grounds. The motivation of the proposed model will be much stronger, for example, if the suggested neuromodulator effects are indeed hinted in some experimental studies, or if the variance- and mean-encoding neurons and their projections onto other cell populations & higher cortical circuit are supported by biological evidence.

We agree that the network model we propose "seems" fairly obvious in hindsight. However, we want to note that this is true for any model and mechanism published. Eve Marder put it nicely in her paper: *"Ironically, a theory paper that 'solves' a difficult problem with a simple, transparent, and elegant solution can be easily devalued. It can take months or years of work to come to a result, which once understood, seems simple, and perhaps obvious."* (Living Science: Theoretical musings, 2020)

Regarding the experimental evidence, we agree with the reviewer. To better ground the model in experimental data, we have revised the section on biological evidence in the Discussion, line 434 ff:

What could be the biological basis for our network model? Sensory information is commonly believed to be channeled through the thalamus and initially arrives in layer 4 of the neocortex (Douglas and Martin, 2004; Bruno and Sakmann, 2006). Neurons in layer 4 subsequently relay the information to layer 2/3 (Douglas and Martin, 2004), where it is further integrated with inputs from higher-order cortical areas entering layer 1 (Larkum, 2013b). From layer 2/3, the information is subsequently forwarded to layer 5 neurons (Douglas and Martin, 2004; Thomson and Bannister, 1998), which integrate it with direct inputs from the thalamus (Constantinople and Bruno, 2013). The core hypothesis of our model is the presence of sensory PE neurons that have been predominately found in layer 2/3, in different brain areas of various species (Eliades and Wang, 2008; Keller and Hahnloser, 2009; Keller et al., 2012; Attinger et al., 2017; Jordan and Keller, 2020; Audette et al., 2021). While we assume these neurons encode PEs in their activity, it remains an active research area whether PEs are encoded in the (spiking) activity of separate neurons and/or in the local voltage dynamics of dendrites (Mikulasch et al., 2023). Recent findings by Gillon et al. (2024) (Gillon et al., 2024) indicate that pattern-violation signals are processed differently at the soma and dendrites over time, suggesting a more complex role for excitatory neuron compartments in predictive processing than our simplified model accounts for.

In our network, memory neurons could correspond to a subset of excitatory L2/3 neurons. Some L2/3 neurons have been shown to develop predictive responses to expected visual stimuli (Fiser et al., 2016). Additionally, a group of L2/3 neurons has been shown to integrate both negative and positive prediction errors (O'Toole et al., 2022), which aligns with our assumption. The weighted output of our network aligns with the concept of internal representation neurons in predictive processing theories (Bastos et al., 2012; Keller and Mrsic-Flogel, 2018), hypothesized to be deep-layer 5 (L5) neurons (Bastos et al., 2012; Heindorf and Keller, 2022). These large pyramidal cells in L5 are ideally situated to integrate top-down information (e.g., predictions) arriving at their apical dendrites in layer 1 with bottom-up information (e.g., sensory inputs) arriving at their basal dendrites in deeper layers (Larkum, 2013a; Harris and Shepherd, 2015).

Neurons encoding variance in primary sensory areas remain a prediction of our model that requires validation. However, it has been shown that stimulus uncertainty can be encoded in the gain variability of individual neurons in V1 and V2 of macaques (Hénaff et al., 2020). Evidence also indicates that populations of neurons can encode uncertainty (Soltani and Izquierdo, 2019). For instance, neurons in the parietal cortex of monkeys encode confidence in perceptual decisions (Kiani and Shadlen, 2009), and neurons in the orbitofrontal cortex encode confidence regardless of sensory modality (Masset et al., 2020). Neural signatures of uncertainty have also been found in regions of the prefrontal cortex (Rushworth and Behrens, 2008), the rat insular and orbitofrontal cortex (Jo and Jung, 2016), and the dorsal striatum in monkeys (White and Monosov, 2016). Additionally, the accuracy of memory recalls is encoded in single neurons of the human parietal and temporal lobes (Rutishauser et al., 2015, 2018).

Please note that to avoid unnecessary repetitions we have decided to move a few sentences from the first results section to the Discussion on biological evidence.

Moreover, we have also edited the paragraph on the neural basis of the weighting, line 468 ff:

In our computation, the relative weights with which the sensory input and the prediction are integrated depend on the activities of the lower-level and higher-level variance neurons. While it is unlikely that these variance neurons can directly modulate the weights, they might trigger the release of neuromodulators that then in turn affect the synaptic plasticity of those weights (Gao and Goldman-Rakic, 2003; Picciotto et al., 2012). For example, deep L5 neurons, which have been hypothesised to act as internal representation neurons (Bastos et al., 2012; Heindorf and Keller, 2022) could be modulated in this manner. Depending on the receptor types present in the apical and basal dendrites, neuromodulators could either decrease or increase the synaptic weights connecting the sensory input and the prediction to these neurons.

Alternatively, the integration of sensory inputs and predictions could occur without changes in synaptic weights, implemented through a network of neurons encoding different aspects of the computation via their activities. For instance, an inhibitory neuron could encode the sum of the variances and exert divisive inhibition on another neuron, which is driven by the sensory input and the higher-order variance neuron in a multiplicative manner. Interneurons such as PV or SOM neurons have been shown to exert divisive inhibition (Lee et al., 2012; Seybold et al., 2015). Moreover, these computations could also be carried out by different compartments within the same neuron. For example, a deep L5 pyramidal cell may receive the sensory input at its basal dendrites and the activity of the higher-order variance neuron at its apical dendrites.

In terms of the neuromodulator results, we show under which conditions our model is in line with documented findings on ACh and NA. However, we refrain from bold, absolute statements on purpose because, in those complex networks with several parameters, many knobs could be tweaked to get a qualitative match with experimental findings. We believe it is of greater value to show under which conditions the model is in line with experimental studies and thereby provide predictions that can be used to verify or falsify our model.

Overall, while this work provides interesting ideas and the revisions have improved it further, the novel contributions made by this work in addition to the authors' previous publications are rather limited.

Here are some other minor suggestions/comments:

- 1. Supporting Information- B. Supporting analyses: References to Supplementary Figure 12 in B2 and B3 should be changed to Supplementary Figure 13.**

Well spotted - Thank you! We have corrected the reference.

- 2. Supplementary Figure 5: Please show the actual mean and the actual variance on panels A & B for clarity.**

Thank you for that suggested. We added the actual mean and variance in the plots.

- 3. Supporting Discussion: Line 208 "In addition, we include VIP neurons that are known to receive top-down inputs and to provide disinhibition.": Please provide a reference supporting this.**

We have added a number of references to support this claim. Thanks for bringing it up.

Reviewer #4

For me, the paper was still quite difficult to read. Equations are still explained in words, which doesn't make sense – to understand what's going on, one has to turn the words into equations, so why not start with the equations? Or at least make them easily accessible. It's true that there's a figure in SI, but it didn't help me that much.

In addition, it was pretty much impossible, from the main text, for me to figure out what problem they were addressing. The focus was on prediction when they in fact meant prior, which is utterly confusing for a theorist; "prediction" does not equate to prior.

Fortunately, I stumbled across Secs. B1 and A1.3 in SI, and I think (but am still not

100% sure) that I figured out what was going on. (Those sections were in the original submission, but I didn't notice them.) Sec. B1 told us what the network did: it low-pass filtered the sensory input to get the mean, and then subtracted the mean from the sensory input, squared it, and low-pass filtered that to get the variance. The mean was then passed to the next network, where the process was repeated – thus producing the mean and variance over trials of the sensory input. This formed the prior. Section A1.3 then told us that on each trial, the means were combined with the variances to produce, on each trial, the minimum variance estimate of the sensory input. It would have been very helpful to have that in the main text.

Equations: Given the journal's broad and diverse audience, we aimed to be inclusive by following a common practice: presenting equations and their supporting derivations or analyses in the Methods section and Supplementary Information (SI), while describing the underlying principles in words in the main text.

However, as theoreticians, we understand and appreciate the clarity provided by including equations directly in the main text. Therefore, following the reviewer's suggestion, we have included a box next to the main text containing the principles of the network's computations described in section B1 of the SI (see line 127 ff). Please also note that the equation from A1.3 is part of Figure 3B. We defer to the editor and reviewers to determine whether this approach is sufficient and in line with the journal's guidelines.

Prior vs. Prediction: To our knowledge, a prior probability distribution of an uncertain quantity (usually a model parameter or a latent variable) represents the "best" probability distribution for that quantity based on prior knowledge or beliefs *before* observing the data. In the context of sensory processing, a prior might represent an initial expectation about the sensory stimulus *before* any sensory information is received. Based on this definition, the activity of the memory neuron cannot be considered a prior because it is continuously shaped by both past and current sensory input. However, extensions have been proposed, such as Empirical Bayes Methods, where the prior is not fixed and can be estimated from the actual data. In this sense, the activity of the memory neuron could also be considered a prior (although the activity of the M neuron cannot be interpreted as a probability distribution per se).

A prediction, on the other hand, is an estimate of the current or a future observation (in our case, the current or next-step stimulus value) given the past data (here, the sensory inputs). Hence, from our perspective, the word prediction is appropriate in the examples we use. To be more precise, given the incoming noisy sensory information and the model at hand, the prediction of the current sensory input is the mean of the previous inputs. Therefore, the activity of the memory neuron, reflecting the mean of the sensory input, can be considered both a prediction for the current step and a prior for the next one. We have updated the sentence on line 130 ff:

While we consider the activity of the memory neuron as a prediction of the current sensory input, it could also be interpreted as a prior of the sensory mean at the next time step.

Given that the model uses "prediction-error neurons," which have been observed in the neocortex, we feel that the term prediction is a less technical and more intuitive description for the activity of the memory neuron.

However, we acknowledge that the reviewer has deep knowledge in this field, and we want to ensure we address their concerns appropriately. We kindly ask the reviewer to further elaborate on their concerns regarding the usage of the term "prediction" in the context of our study if it has not been resolved with our explanations.

In any case, now that I know what's going on, I can review the science. Here I think there are problems.

1. **The goal is standard Bayesian inference: the brain acquires the mean and variance of some sensory signal, and that's combined with the prior (also characterized by a mean and variance) to provide an optimal estimate of the sensory input.**

But details matter. The sensory signal consists of multiple presentations. More precisely, letting x_t be the sensory signal at time t , the sensory signal is x_1, x_2, \dots, x_n . The x_t are drawn from a distribution with mean μ and variance σ^2 . What the circuit does is estimate μ and σ^2 .

That seems problematic to me: given n independent samples from a distribution, the variance is σ^2/n , not σ^2 . It's not totally clear that the brain can do that kind of averaging, but it's almost for sure true that the brain can estimate the mean with lower variance than σ^2 . So in that respect the setup is not very realistic.

Much more realistic is that a single sensory input tells you both the mean and variance (when you hear a sound or see an image, typically you immediately know how uncertain you are about it). For repeated multiple stimuli, figuring out what variance the brain estimates is highly nontrivial, but almost for sure it's not the raw variance of the x_t .

It is correct that the network aims to estimate the mean μ_{in} and the variance σ_{in}^2 of the sensory signal, as well as the uncertainty of the prediction which is caused by unpredictable environment switches (modeled here as changes in the mean of the stimulus distribution). We defined a period during which the stimulus mean remains constant as a trial.

In each trail, we draw N_{in} values (x_1, x_2, x_3, \dots) from a normal distribution with mean μ_{in} and variance σ_{in}^2 to simulate noise. Each value (x_n) is presented N_{step} consecutive time steps. As demonstrated (Fig. 2), the lower-level subnetwork can correctly estimate the mean and the variance of the sensory signal under these conditions. This has also been verified with a continuously changing signal (see Supplementary Fig. 4).

To assess whether our network can track the uncertainty of the prediction caused by switches in the mean of the stimulus distribution, we draw μ_{in} from a uniform distribution $U(a, b)$ for each trial. Thus, theoretically, the variance of the μ_{in} is given by $\sigma_{\text{trial}}^2 = (b - a)/\sqrt{12}$. We have revised and refined the explanation of the stimuli on line 194 ff.

We believe the reviewer is referring to the concept of the squared standard error of the mean (SEM^2). However, we would like to clarify that in our case σ_{trial}^2 would only be equivalent to SEM^2 if the mean of the stimulus distribution remained the same in every trial. In such a scenario, the higher-order variance neuron should indeed estimate the SEM^2 because, in each trial, the lower-order memory neuron would be estimating the mean of the same stimulus distribution. However, when the mean changes in each trial, this equivalence does not hold.

The reviewer's comment raises an important issue that merits a more thorough discussion in our paper. Several factors can contribute to the over- or underestimation of σ_{trial}^2 . For example, the limited duration of trials (and, therefore, the stimulus duration) constrains the accuracy of the variance estimation. Additionally, neurons in our network do not function instantaneously; their responses are defined by time constants that introduce transient phases, which can bias variance estimates. This effect may be particularly pronounced when noise levels are high and the stimulus changes rapidly. Furthermore, the PE neurons in our network exhibit a small baseline activity, leading to an overestimation of small variances.

We now discuss this further in a new section titled "Model assumptions, simplifications & limitations" in the Supporting Information, line 324 ff:

[...] Several factors influence the accuracy of the variance estimation, leading to potential over- or underestimation of the true variance. For instance, the limited duration of trials (and thus the stimulus duration) inherently restricts the precision of variance estimation. Moreover, neurons do not operate instantaneously. Their activity upon a stimulus is governed by time constants that introduce transient responses. These ON/OFF responses impact the variance estimation, especially when the noise levels are high and the stimulus changes rapidly. Additionally, the PE neurons in our network exhibit a small baseline activity, which may result in the overestimation of small variances.

However, we do not expect the brain to be Bayes-optimal at all times. The variances may not always be used directly. As the reviewer pointed out, it is unclear if the brain computes variance precisely. It likely does not need to, as long as it can use some function of uncertainty to judge which signal is more trustworthy.

Moreover, we believe that a stimulus must be perceived for a certain duration to accurately estimate its mean and variance. This is especially true for auditory stimuli – one cannot determine the variance of an audio signal instantaneously but needs to listen to it for some time. This approach is

how we set up our simulated experiments with the additional twist that after some time the mean of the stimulus changes.

2. **Low pass filtering a signal to get the mean works only if the timescale of the filter is long compared to the timescale of the variability in the sensory input. In particular, if the sensory input varies on a timescale long compared to the timescale of the filter, it will track the input almost perfectly and yield no variance at all. Given that the variability is likely to have multiple timescales, having a single filter timescale seems like a bad idea.**

We agree with the reviewer that the time constants of the neurons, particularly the M and V neurons as well as the nPE and pPE neurons, are crucial and must be chosen (or ideally learned) in relation to the timescales of the stimulus. The stimulus can vary on two timescales: Each stimulus is a concatenation of N_{in} piecewise constant values, with each value shown for N_{step} time steps. These values are drawn from a normal distribution whose mean changes after $N_{\text{in}} \cdot N_{\text{step}}$ time steps.

The time constants of the V neurons were chosen to be of the order of a single stimulus presentation ($N_{\text{in}} \cdot N_{\text{step}}$ time steps), that is, 5 seconds. The M neurons, modeled as perfect integrators, change on a timescale defined by the weights of the connections from the PE neurons to the M neurons. To account for the two different timescales at which our stimulus can vary, we selected smaller weights for the PE neurons connecting to the M neurons in the higher-level PE circuit compared to those in the lower-level circuit. We have demonstrated that these timescales significantly impact system performance (see Supplementary Figures 6 and 10) and are critical for the system to function correctly. To further discuss this, we have now added a section on this in "Model assumptions, simplifications & limitations" in the Supporting Information, line 303 ff:

The sensory input in our work can vary on two timescales: changes in the stimulus caused by noise are faster than those caused by switches in the environment. To account for this, the lower-level memory neuron must operate on a faster timescale than the higher-level memory neuron. To achieve this, the weights from the PE neurons onto the memory neuron are larger for the lower than the higher PE circuit. This assumption is consistent with the observation that time constants increase along the cortical hierarchy (Murray et al., 2014; Chaudhuri et al., 2015; Runyan et al., 2017). While we have set these weights in our network, they are more likely subject to plasticity, allowing them to be learned and adjusted according to the stimulus timescales that can change in real life.

3. **Combining likelihood with prior (which is essentially what they did) requires the relative weights to depend on the variance. To set those weights, they plugged in the computed variance (Eq. 7 of SI). Biologically, it's pretty unrealistic to have activity affect synaptic weight in just the right way. Given that this paper is all about biological realism, that seems to be a serious problem.**

We agree that it would be great to have a neural implementation for the weighting of sensory inputs and predictions that we currently only do arithmetically. While an exact implementation is currently beyond the scope of this project, we would like to deepen our discussion on potential mechanisms that may be involved:

Pre- and postsynaptic activity indeed modulates synaptic weights/efficacy (Turrigiano and Nelson, 2004; Citri and Malenka, 2008). In our computation, the relative weights with which the sensory input and the prediction are integrated depend on the activities of the lower-level and higher-level variance neurons, rather than the presynaptic inputs (i.e., the sensory input or the prediction), which is more challenging to justify. However, it is conceivable that the activity of the variance neurons could trigger the release of different neuromodulators that have been shown to strongly affect plasticity (Gao and Goldman-Rakic, 2003; Picciotto et al., 2012).

For instance, excitatory elongated neurons that receive top-down predictions on their apical dendrites and feedforward sensory input on their basal dendrites could be modulated in this way. Depending on the types of receptors present in these distinct compartments, neuromodulators could either decrease or increase the synaptic weights connecting the sensory input and the prediction to the internal representation neurons. In this scenario, variance-encoding neurons would not directly affect the activity of the internal representation neurons – instead, they would regulate the release of neuromodulators, which in turn adjust the synaptic weights on these neurons.

Furthermore, we would like to mention that the weighting of the sensory input and the prediction could also be implemented without changes in synaptic weights. This approach would require a network of neurons encoding different aspects of the computation through their activities. For instance, an inhibitory neuron could encode the sum of the two variances and exert divisive inhibition on another neuron. This target neuron would also be driven by the sensory input and the higher-order variance neuron in a multiplicative manner. It has been shown that some interneurons, such as PV or SOM neurons, can indeed exert divisive inhibition (Lee et al., 2012; Seybold et al., 2015). Moreover, these computations could also occur within the same neuron. For example, an elongated layer 5 pyramidal cell might receive sensory input at its basal dendrites while its apical dendrites are driven by the output of the higher-order variance neuron. Research has shown that top-down dendritic input can increase the gain of L5 cells (Larkum et al., 2004). Therefore, in this scenario, the activity at the soma of an L5 neuron, driven by sensory input to the basal dendrites, could be modulated by the variance neuron’s activity arriving at the apical dendrites, resulting in a representation of sensory input modulated by uncertainty.”

We have now added two paragraphs on this in the Discussion, starting line 468 ff:

In our computation, the relative weights with which the sensory input and the prediction are integrated depend on the activities of the lower-level and higher-level variance neurons. While it is unlikely that these variance neurons can directly modulate the weights, they might trigger the release of neuromodulators that then in turn affect the synaptic plasticity of those weights (Gao and Goldman-Rakic, 2003; Picciotto et al., 2012). For example, deep L5 neurons, which have been hypothesised to act as internal representation neurons (Bastos et al., 2012; Heindorf and Keller, 2022) could be modulated in this manner. Depending on the receptor types present in the apical and basal dendrites, neuromodulators could either decrease or increase the synaptic weights connecting the sensory input and the prediction to these neurons.

Alternatively, the integration of sensory inputs and predictions could occur without changes in synaptic weights, implemented through a network of neurons encoding different aspects of the computation via their activities. For instance, an inhibitory neuron could encode the sum of the variances and exert divisive inhibition on another neuron, which is driven by the sensory input and the higher-order variance neuron in a multiplicative manner. Interneurons such as PV or SOM neurons have been shown to exert divisive inhibition (Lee et al., 2012; Seybold et al., 2015). Moreover, these computations could also be carried out by different compartments within the same neuron. For example, a deep L5 pyramidal cell may receive the sensory input at its basal dendrites and the activity of the higher-order variance neuron at its apical dendrites.

4. **The memory neuron has linear tuning, something that’s highly unusual – most neurons have ”bump” (as for orientation) or sigmoidal tuning curves. There’s a reason for that: neurons don’t have much dynamic range (even at the population level). As far as I can tell, their scheme will not apply to more standard population tuning curves.**

We agree with the reviewer that the exact input-output (activation) function for the memory neuron plays an important role in correctly estimating the mean. However, as the reviewer pointed out, neurons are feature-specific. Thus, even if one neuron has a limited dynamic range, an ensemble of cells with uniformly distributed feature selectivity can collectively cover the necessary input range.

Moreover, we would like to emphasize that the weights from the PE neurons onto the M neuron can be learned and adjusted to ensure that the mean of the inputs maps onto the mean of the neuron’s dynamic range. Similarly, the weights from the M neuron onto the PE neurons can be adjusted so that the mean input value equals the product of the weights and the activity of the M neuron. There are several plasticity rules that adjust synaptic weights to map the input range onto the dynamic range of a neuron. Homeostatic plasticity helps neurons adjust their sensitivity to inputs, thereby aligning their dynamic range with the statistical properties of the input signals (Turrigiano, 2012, 2017). An example of this is synaptic scaling (Turrigiano and Nelson, 2004; Turrigiano, 2012), where the strengths of all synapses on a neuron are scaled up or down to maintain overall activity within a target range. Additionally, neurons can adjust their gain, or sensitivity to inputs, based on the statistical properties of the inputs they receive (Fairhall et al., 2001).

To address this, we have now also added a paragraph in our section ”*Model assumptions, simplifications & limitations*” in the Supporting Information, line 290 ff:

As with any computational model, we simplify certain biological details to maintain the model’s

simplicity and interpretability. However, those details, while beyond the scope of this study, may be well investigated in future work. For instance, we have simplified the input-output activation function of the neurons. The memory neurons are modeled with a linear tuning curve and a broad dynamic range. This contrasts with typical neurons, which are usually feature-specific and respond to a limited range of inputs. However, homeostatic plasticity can help neurons adjust their sensitivity to inputs, thereby aligning their dynamic range with the statistical properties of the input signals (Turrigiano, 2012, 2017). One example is synaptic scaling (Turrigiano and Nelson, 2004; Turrigiano, 2012), where the strengths of all synapses on a neuron are scaled up or down to maintain overall activity within a target range. Additionally, neurons can adjust their gain, or sensitivity to inputs, based on the statistical properties of the inputs they receive (Fairhall et al., 2001).

5. **Performing Bayesian inference (on a range of problems, including the simple case here of combining the likelihood with the prior) has a long history in this field. The authors alluded to it on lines 47-48, where they say "However, how the variance of both the sensory input and the prediction can be computed on the circuit level is not resolved yet". Although it's not resolved, it's not for lack of trying. There's a pretty big literature on this (which mainly consists of sampling – Lengyel and colleagues – and probabilistic population codes – Pouget and colleagues). Essentially none of it was referenced. The work in this paper needs to be placed in the context of that previous work.**

Thank you for bringing this to our attention. We regret the oversight in not adequately referencing the significant body of work by Lengyel and colleagues on sampling-based approaches and by Pouget and colleagues on probabilistic population codes.

To address this, we have expanded our literature review to include these key contributions and provide a more comprehensive context for our study, line 557 ff in the Discussion:

Our model suggests one potential neuronal circuit mechanism for the uncertainty estimation of sensory inputs and predictions. Modeling specific neurons that encode the variance of feedforward sensory inputs and predictions aligns with the concept that neurons can explicitly represent parameters of a probability distribution, such as the mean or variance (see also Wilmes et al., 2023; O'Neill and Schultz, 2010; O'Reilly et al., 2012). However, the representation of variances in the brain is still not comprehensively understood, and several alternative models have been proposed. For instance, uncertainty might be decoded from the collective activity of neuron populations (Pouget et al., 2000; Knill and Pouget, 2004; Ma et al., 2006; Dehaene et al., 2021). Some theories suggest that uncertainty is represented by the amplitude (Ma et al., 2006), the width (Fischer and Peña, 2011) or the variability of a neuron's response (Hoyer and Hyvärinen, 2002; Ma et al., 2006). Another prominent theory is the neural sampling hypothesis, which suggests that neural circuits encode probability distributions rather than precise values. In this framework, the variability in neural responses is interpreted as samples drawn from these distributions (Fiser et al., 2010; Buesing et al., 2011; Berkes et al., 2011). [...]

References

- Alexander Attinger, Bo Wang, and Georg B Keller. Visuomotor coupling shapes the functional development of mouse visual cortex. *Cell*, 169(7):1291–1302, 2017.
- Nicholas J Audette, WenXi Zhou, and David M Schneider. Temporally precise movement-based predictions in the mouse auditory cortex. *bioRxiv*, pages 2021–12, 2021.
- Andre M Bastos, W Martin Usrey, Rick A Adams, George R Mangun, Pascal Fries, and Karl J Friston. Canonical microcircuits for predictive coding. *Neuron*, 76(4):695–711, 2012.
- Pietro Berkes, Gergő Orbán, Máté Lengyel, and József Fiser. Spontaneous cortical activity reveals hallmarks of an optimal internal model of the environment. *Science*, 331(6013):83–87, 2011.
- Randy M Bruno and Bert Sakmann. Cortex is driven by weak but synchronously active thalamocortical synapses. *Science*, 312(5780):1622–1627, 2006.
- Lars Buesing, Johannes Bill, Bernhard Nessler, and Wolfgang Maass. Neural dynamics as sampling: a model for stochastic computation in recurrent networks of spiking neurons. *PLoS computational biology*, 7(11):e1002211, 2011.

- Rishidev Chaudhuri, Kenneth Knoblauch, Marie-Alice Gariel, Henry Kennedy, and Xiao-Jing Wang. A large-scale circuit mechanism for hierarchical dynamical processing in the primate cortex. *Neuron*, 88(2):419–431, 2015.
- Ami Citri and Robert C Malenka. Synaptic plasticity: multiple forms, functions, and mechanisms. *Neuropharmacology*, 33(1):18–41, 2008.
- Christine M Constantinople and Randy M Bruno. Deep cortical layers are activated directly by thalamus. *Science*, 340(6140):1591–1594, 2013.
- Guillaume P Dehaene, Ruben Coen-Cagli, and Alexandre Pouget. Investigating the representation of uncertainty in neuronal circuits. *PLOS Computational Biology*, 17(2):e1008138, 2021.
- Rodney J Douglas and Kevan AC Martin. Neuronal circuits of the neocortex. *Annu. Rev. Neurosci.*, 27(1):419–451, 2004.
- Steven J Eliades and Xiaoqin Wang. Neural substrates of vocalization feedback monitoring in primate auditory cortex. *Nature*, 453(7198):1102, 2008.
- Adrienne L Fairhall, Geoffrey D Lewen, William Bialek, and Robert R de Ruyter van Steveninck. Efficiency and ambiguity in an adaptive neural code. *Nature*, 412(6849):787–792, 2001.
- Brian J Fischer and José Luis Peña. Owl’s behavior and neural representation predicted by bayesian inference. *Nature neuroscience*, 14(8):1061–1066, 2011.
- Aris Fiser, David Mahringer, Hassana K Oyibo, Anders V Petersen, Marcus Leinweber, and Georg B Keller. Experience-dependent spatial expectations in mouse visual cortex. *Nature neuroscience*, 19(12):1658–1664, 2016.
- József Fiser, Pietro Berkes, Gergő Orbán, and Máté Lengyel. Statistically optimal perception and learning: from behavior to neural representations. *Trends in cognitive sciences*, 14(3):119–130, 2010.
- Wen-Jun Gao and Patricia S Goldman-Rakic. Selective modulation of excitatory and inhibitory microcircuits by dopamine. *Proceedings of the national academy of sciences*, 100(5):2836–2841, 2003.
- Colleen J Gillon, Jason E Pina, Jérôme A Lecoq, Ruweida Ahmed, Yazan N Billeh, Shiella Caldejon, Peter Groblewski, Timothy M Henley, Eric Lee, Jennifer Luviano, et al. Responses to pattern-violating visual stimuli evolve differently over days in somata and distal apical dendrites. *Journal of Neuroscience*, 44(5), 2024.
- Arno Granier, Mihai A Petrovici, Walter Senn, and Katharina A Wilmes. Precision estimation and second-order prediction errors in cortical circuits. *arXiv preprint arXiv:2309.16046*, 2023.
- Kenneth D Harris and Gordon MG Shepherd. The neocortical circuit: themes and variations. *Nature neuroscience*, 18(2):170, 2015.
- Matthias Heindorf and Georg B Keller. Reduction of layer 5 mediated long-range cortical communication by antipsychotic drugs. *bioRxiv*, 2022.
- Olivier J Hénaff, Zoe M Boundy-Singer, Kristof Meding, Corey M Ziemba, and Robbe LT Goris. Representation of visual uncertainty through neural gain variability. *Nature communications*, 11(1):2513, 2020.
- Patrik Hoyer and Aapo Hyvärinen. Interpreting neural response variability as monte carlo sampling of the posterior. *Advances in neural information processing systems*, 15, 2002.
- Suhyun Jo and Min Whan Jung. Differential coding of uncertain reward in rat insular and orbitofrontal cortex. *Scientific reports*, 6(1):24085, 2016.
- Rebecca Jordan and Georg B Keller. Opposing influence of top-down and bottom-up input on excitatory layer 2/3 neurons in mouse primary visual cortex. *Neuron*, 108(6):1194–1206, 2020.
- Georg B Keller and Richard HR Hahnloser. Neural processing of auditory feedback during vocal practice in a songbird. *Nature*, 457(7226):187, 2009.

- Georg B Keller and Thomas D Mrsic-Flogel. Predictive processing: A canonical cortical computation. *Neuron*, 100(2):424–435, 2018.
- Georg B Keller, Tobias Bonhoeffer, and Mark Hübener. Sensorimotor mismatch signals in primary visual cortex of the behaving mouse. *Neuron*, 74(5):809–815, 2012.
- Roозbeh Kiani and Michael N Shadlen. Representation of confidence associated with a decision by neurons in the parietal cortex. *science*, 324(5928):759–764, 2009.
- David C Knill and Alexandre Pouget. The bayesian brain: the role of uncertainty in neural coding and computation. *TRENDS in Neurosciences*, 27(12):712–719, 2004.
- Matthew Larkum. A cellular mechanism for cortical associations: an organizing principle for the cerebral cortex. *Trends in neurosciences*, 36(3):141–151, 2013a.
- Matthew E Larkum. The yin and yang of cortical layer 1. *Nature neuroscience*, 16(2):114–115, 2013b.
- Matthew E Larkum, Walter Senn, and Hans-R Lüscher. Top-down dendritic input increases the gain of layer 5 pyramidal neurons. *Cerebral cortex*, 14(10):1059–1070, 2004.
- Seung-Hee Lee, Alex C Kwan, Siyu Zhang, Victoria Phoumthippavong, John G Flannery, Sotiris C Masmanidis, Hiroki Taniguchi, Z Josh Huang, Feng Zhang, Edward S Boyden, et al. Activation of specific interneurons improves v1 feature selectivity and visual perception. *Nature*, 488(7411):379–383, 2012.
- Wei Ji Ma, Jeffrey M Beck, Peter E Latham, and Alexandre Pouget. Bayesian inference with probabilistic population codes. *Nature neuroscience*, 9(11):1432–1438, 2006.
- Paul Masset, Torben Ott, Armin Lak, Junya Hirokawa, and Adam Kepecs. Behavior-and modality-general representation of confidence in orbitofrontal cortex. *Cell*, 182(1):112–126, 2020.
- Fabian A Mikulasch, Lucas Rudelt, Michael Wibral, and Viola Priesemann. Where is the error? hierarchical predictive coding through dendritic error computation. *Trends in Neurosciences*, 46(1):45–59, 2023.
- John D Murray, Alberto Bernacchia, David J Freedman, Ranulfo Romo, Jonathan D Wallis, Xinying Cai, Camillo Padoa-Schioppa, Tatiana Pasternak, Hyojung Seo, Daeyeol Lee, et al. A hierarchy of intrinsic timescales across primate cortex. *Nature neuroscience*, 17(12):1661–1663, 2014.
- Martin O’Neill and Wolfram Schultz. Coding of reward risk by orbitofrontal neurons is mostly distinct from coding of reward value. *Neuron*, 68(4):789–800, 2010.
- Jill X O’Reilly, Saad Jbabdi, and Timothy EJ Behrens. How can a bayesian approach inform neuroscience? *European Journal of Neuroscience*, 35(7):1169–1179, 2012.
- Sean M O’Toole, Hassana K Oyibo, and Georg B Keller. Prediction error neurons in mouse cortex are molecularly targetable cell types. *BioRxiv*, pages 2022–07, 2022.
- Marina R Picciotto, Michael J Higley, and Yann S Mineur. Acetylcholine as a neuromodulator: cholinergic signaling shapes nervous system function and behavior. *Neuron*, 76(1):116–129, 2012.
- Alexandre Pouget, Peter Dayan, and Richard Zemel. Information processing with population codes. *Nature Reviews Neuroscience*, 1(2):125–132, 2000.
- Caroline A Runyan, Eugenio Piasini, Stefano Panzeri, and Christopher D Harvey. Distinct timescales of population coding across cortex. *Nature*, 548(7665):92–96, 2017.
- Matthew FS Rushworth and Timothy EJ Behrens. Choice, uncertainty and value in prefrontal and cingulate cortex. *Nature neuroscience*, 11(4):389–397, 2008.
- Ueli Rutishauser, Shengxuan Ye, Matthieu Koroma, Oana Tudusciuc, Ian B Ross, Jeffrey M Chung, and Adam N Mamelak. Representation of retrieval confidence by single neurons in the human medial temporal lobe. *Nature neuroscience*, 18(7):1041–1050, 2015.
- Ueli Rutishauser, Tyson Aflalo, Emily R Rosario, Nader Pouratian, and Richard A Andersen. Single-neuron representation of memory strength and recognition confidence in left human posterior parietal cortex. *Neuron*, 97(1):209–220, 2018.

- Bryan A Seybold, Elizabeth AK Phillips, Christoph E Schreiner, and Andrea R Hasenstaub. Inhibitory actions unified by network integration. *Neuron*, 87(6):1181–1192, 2015.
- Alireza Soltani and Alicia Izquierdo. Adaptive learning under expected and unexpected uncertainty. *Nature Reviews Neuroscience*, 20(10):635–644, 2019.
- AM Thomson and AP Bannister. Postsynaptic pyramidal target selection by descending layer iii pyramidal axons: dual intracellular recordings and biocytin filling in slices of rat neocortex. *Neuroscience*, 84(3):669–683, 1998.
- Gina Turrigiano. Homeostatic synaptic plasticity: local and global mechanisms for stabilizing neuronal function. *Cold Spring Harbor perspectives in biology*, 4(1):a005736, 2012.
- Gina G Turrigiano. The dialectic of hebb and homeostasis. *Philosophical transactions of the royal society B: biological sciences*, 372(1715):20160258, 2017.
- Gina G Turrigiano and Sacha B Nelson. Homeostatic plasticity in the developing nervous system. *Nature reviews neuroscience*, 5(2):97–107, 2004.
- J Kael White and Ilya E Monosov. Neurons in the primate dorsal striatum signal the uncertainty of object–reward associations. *Nature communications*, 7(1):12735, 2016.
- Katharina A Wilmes, Mihai A Petrovici, Shankar Sachidhanandam, and Walter Senn. Uncertainty-modulated prediction errors in cortical microcircuits. *bioRxiv*, pages 2023–05, 2023.

We are grateful for the reviewers' thoughtful feedback, which has helped us identify key areas for improvement. We are confident that the proposed revisions will strengthen the manuscript and make it more accessible and compelling to the broader research community.

To address the comments comprehensively, we have revised the main text and the Supporting Information, conducted additional simulations now summarized in a new Supplementary Figure (Supplementary Figure 10), and re-ordered the SI Figures to ensure they appear in the correct sequence. Additionally, in light of the collective feedback from all reviewers and following the *Nature Communications* guidelines, we have made further modifications to enhance clarity and conciseness where appropriate.

Below, we provide detailed responses (blue) to all comments (bold black) and hope that we have satisfactorily addressed all concerns (line numbers correspond to the lines in the document with highlighted changes).

Detailed responses

Reviewer #3

In this second revision, the authors have mainly updated the text for clarification. In particular, they included an additional discussion on alternative network architectures and biological grounds of the proposed circuit model. The added discussion on the experimental evidence is especially helpful and informative, as they provide speculations on the location and identity of the proposed M and V neurons. This update has alleviated my previous concerns regarding biological relevance. My only remaining concern, however, is on the novelty and additional contributions of this work in comparison to the authors' previous published works (Hertag & Clopath, 2022; Hertag & Sprekeler, 2020) as noted in both the first and the second reviews. As the authors point out, I do recognize that the current manuscript made several new extensions to the previous circuit model, such as incorporating M and V neurons for computing the mean and the variance of the sensory inputs and the model predictions of neuromodulator effects on weighting of sensory inputs and predictions. I am not sure, however, whether these would be sufficient innovations compared to the authors' previous papers; yet, I think this work is a solid and interesting study suggesting a possible circuit mechanisms for computing the uncertainty using prediction-error circuits.

We thank the reviewer for their thoughtful comments and for acknowledging the improvements in the manuscript, particularly the additional discussion on biological plausibility and experimental evidence for our circuit model.

Regarding novelty: As the reviewer pointed out, the major contributions of our work lie in proposing a specific circuit-level implementation for estimating the variance and mean of sensory stimuli, as well as the uncertainty of predictions. To the best of our knowledge, an implementation at this level of detail has not been proposed previously. Central to our proposal is the idea that negative and positive prediction-error neurons form the backbone of this computation. While our work builds on previously discussed PE circuits, we propose a new framework by introducing a hierarchical model that incorporates both lower and higher PE circuits, alongside memory neurons and variance-encoding neurons to enhance its functionality. Furthermore, we explore how neuromodulators influence the weighting of sensory and prediction inputs by targeting specific interneurons, such as PV, SOM, and VIP neurons. This investigation establishes a critical link between neuromodulation and changes in predictive processing.

In response to the comments from Reviewer 4, we now conducted additional simulations, summarized in Supplementary Figure 10. These findings further strengthen our manuscript by providing a comparison of our reliability-based weighting approach with two alternative methods brought to our attention. The first method combines sensory input and predictions using sensory variance alone. The second is based on the similarity between lower and higher memory neurons. While both approaches perform well in low-noise conditions, neither effectively filters noise in high-noise regimes. These findings have been incorporated

into the main manuscript (lines 235–250):

This observation highlights that the approach is suboptimal immediately after a change point, as predictions based on previous sensory inputs become incorrect following an environmental change. In such cases, the system should promptly adapt to the new sensory input. Alternative approaches that detect potential change points and allow the system to prioritize sensory input after a change are possible and have been discussed (see, e.g., Meijer et al., 2024). However, identifying change points can be challenging, especially when the level of sensory noise varies. While it is common to focus on changes in the environment (i.e., μ), changes in sensory noise levels (i.e., σ) can also occur. In this work, we considered a spectrum of scenarios encompassing both environmental changes and fluctuations in noise levels. Although the weighting strategy used here is less accurate immediately after a change point, it performs well in steady-state conditions (Supplementary Fig. 10A). Furthermore, the reliability-weighted input approach effectively handles scenarios where sensory noise undergoes abrupt changes (Supplementary Fig. 10A). In contrast, simpler methods designed to minimize inaccuracies after a change point may struggle with high-noise scenarios. For example, while approaches based solely on sensory input variance (Supplementary Fig. 10B) or the disparity between the lower and the higher memory neuron (Supplementary Fig. 10C) improve the output estimate in low-noise conditions, they struggle in high-noise conditions..

We have also discussed these results in the Discussion (lines 397-406):

Relying more on predictions at the onset of a new trial, immediately after a change point, can be suboptimal. It was found that subjects tended to discard their predictions immediately following a change point in a sound-localization task where subjects were asked to predict the next stimulus after observing a series of stimuli (Krishnamurthy et al., 2017; Meijer et al., 2024). However, as noted in the study, participants were informed about the nature of the task, which could have influenced their responses. In situations where the underlying task is not explicitly known, the strategy may be less clear. This is particularly true when changes occur in sensory noise rather than in the environment (that is, σ than μ). In such cases, a reliability-based weighting of sensory input and predictions might offer an advantage. Nevertheless, our model could be extended to include a change-point detection mechanism (see, e.g., Meijer et al., 2024), which could help reduce the observed discrepancy immediately after a change point.

Reviewer #4

OK, I think I finally understand what's going on. Mainly because I talked to two of the authors, Claudia Clopath and Loreen Hertäg. After they explained it, I realized it's all in the paper. But not where I was expecting, and somewhat spread out.

But first, a general comment: this paper was more about the circuit than the algorithm, and that part was good. So, although this seems like kind of a long review, it shouldn't be taken as negative; more as guidance on how to make the paper better.

We thank the reviewer for his detailed feedback, insightful analysis, and constructive suggestions for improving the clarity and focus of the manuscript. We will address each of the reviewer's points below.

So let me summarize. Subjects observe a stimulus, $s(t)$, which has the following behavior:

- $s(t)$ consists of a series of steps
- each step lasts 500 ms
- the height of each step is drawn from $N(\mu, \sigma^2)$.
- every 5 seconds μ is redrawn from a uniform distribution.

The goal is to estimate μ . So basically subjects get 10 noisy samples at each μ .

This is a classic change detection task. Which is possibly why I was so confused (and why comment 1 in my previous review was pretty far off): nowhere in the paper is change

detection mentioned. In fact, line 40 says "A common hypothesis is that the brain weights different inputs according to their reliabilities". Technically that's true, but it's a bit misleading given the prior literature in the field (OK, I was misled), for which the major emphasis has been on an uncertain signal and a prior.

By the way, almost *exactly* this task has been considered before; see Krishnamurthy et al., Nature Human Behaviour 2017,

<https://www.nature.com/articles/s41562-017-0107>

And the task was analyzed theoretically by Meijer et al., bioRxiv 2024,

<https://www.biorxiv.org/content/10.1101/2024.10.29.620874v1>).

Both those papers should be referenced.

Given that this is a change detection task, we can ask: is the algorithm used by the authors a reasonable one? For their algorithm, they updated two sets of variables, corresponding to a lower and higher level circuit. Using L and H for lower and higher, and M and V for mean and variance, the update rules were

lower:

$$\begin{aligned} \tau_{LM} dr_{LM}/dt &= -r_{LM} + s(t) \\ \tau_{LV} dr_{LV}/dt &= -r_{LV} + (r_{LM}(t) - s(t))^2 \end{aligned}$$

higher:

$$\begin{aligned} \tau_{HM} dr_{HM}/dt &= -r_{HM} + r_{LM}(t) \\ \tau_{HV} dr_{HV}/dt &= -r_{HV} + (r_{HM}(t) - r_{LM}(t))^2 \end{aligned}$$

The estimate of the mean, *muhat*, is

$$\text{muhat} = (s(t)/r_{LV} + r_{LM}(t)/r_{HV}) / (1/r_{LV} + 1/r_{HV}).$$

According to the equation for *muhat*,

- the stimulus, $s(t)$, is weighted more heavily when the variance of the lower circuit is small compared to the variance of the higher circuit.
- the filtered version of the stimulus, $r_{LM}(t)$, is weighted more heavily when it's the other way around.

Given this, if the circuit is close to optimal, we would expect the variance of the lower circuit to be small compared to the variance of the higher circuit right after a change. In fact, I believe it's the other way around: because the higher circuit lags the lower circuit, the variance of the lower circuit goes up faster than the variance of the higher circuit after a change. So this algorithm seems backwards: it weights the stimulus less heavily after a change, not more heavily. This paper was more about the circuit than the algorithm, and that part was good, so this isn't the end of the world. But I don't think it would be a great idea to pass this off as semi-optimal (on lines 36-7 the authors ask "And how do neural networks in the brain combine both input streams wisely?"). The authors could simply admit that their algorithm is suboptimal. (With a caveat: assuming it is suboptimal; it's possible that I'm totally wrong.) But this doesn't seem like a great idea. A better alternative, in my opinion, is to use a more optimal algorithm. There is (to my knowledge) no closed form solution to this problem, so all reasonable algorithms are approximate. Here are a couple suggestions:

1. Filter the signal at two timescales. The difference between the two filters provides information about whether or not there has been a change: if the two filters are about the same, the output should be a filtered version of the stimulus; otherwise, the output should rely much more heavily on the stimulus. Note, though, that the variance neurons disappear.

2. Use the variance as an indicator of change. When the variance goes up, rely more heavily on $s(t)$; when it's not so large, rely more heavily on $r_{LM}(t)$. Not sure if you need the higher circuit any more, but maybe it can be used.
3. And there are a lot of other options; see Meijer et al., referenced above.

Alternatively, they could show that their algorithm actually is a good one (see caveat above).

Thank you for this valuable suggestion. We appreciate the reviewer's observation that the task in our study can be interpreted as a change detection task, which we had not explicitly emphasized in the initial draft. Furthermore, we are grateful for the references that were pointed out and apologize for the oversight. To address this, we now mention the link to a change detection task and the references immediately after describing the stimulation protocol (lines 202-203):

This setup aligns with a change detection task and has been previously studied (see, e.g., Krishnamurthy et al., 2017; Meijer et al., 2024).

Furthermore, we agree that the reliability-based weighting of sensory input and predictions used in this study is suboptimal immediately after an environmental change, as illustrated in Figure 3C. In this example, the stimulus noise is low, but trial-to-trial variability is high. Following a change, prior predictions should ideally be discarded, and the weighted output should closely align with the sensory input. However, in our example, the weighted output gradually converges to the new sensory input while still incorporating the prediction initially. Although this is suboptimal in this specific context, it is important to note that adopting a more conservative approach can be advantageous in situations where it is unclear whether an abrupt change reflects a shift in the environment or a sudden increase in sensory noise. To demonstrate this, we conducted new simulations where the stimulus transitions from a low-noise to a high-noise scenario (Supplementary Figure 10). As suggested by the reviewer, we compared our network to a method that integrates sensory input and predictions based on: 1) filtering the signal at two timescales, and 2) considering the variance of the sensory input only. While the reliability-based weighting does not capture the state perfectly immediately after the change point (but performs well in the steady state) under low-noise conditions, it does well in high-noise scenarios. In such cases, it effectively filters out noise and appropriately relies more on predictions. By contrast, the alternative approaches (see 1–2 above) perform well in low-noise conditions but exhibit significant difficulties in high-noise scenarios. We have included this analysis and comparison in the main text (lines 235–250):

This observation highlights that the approach is suboptimal immediately after a change point, as predictions based on previous sensory inputs become incorrect following an environmental change. In such cases, the system should promptly adapt to the new sensory input. Alternative approaches that detect potential change points and allow the system to prioritize sensory input after a change are possible and have been discussed (see, e.g., Meijer et al., 2024). However, identifying change points can be challenging, especially when the level of sensory noise varies. While it is common to focus on changes in the environment (i.e., μ), changes in sensory noise levels (i.e., σ) can also occur. In this work, we considered a spectrum of scenarios encompassing both environmental changes and fluctuations in noise levels. Although the weighting strategy used here is less accurate immediately after a change point, it performs well in steady-state conditions (Supplementary Fig. 10A). Furthermore, the reliability-weighted input approach effectively handles scenarios where sensory noise undergoes abrupt changes (Supplementary Fig. 10A). In contrast, simpler methods designed to minimize inaccuracies after a change point may struggle with high-noise scenarios. For example, while approaches based solely on sensory input variance (Supplementary Fig. 10B) or the disparity between the lower and the higher memory neuron (Supplementary Fig. 10C) improve the output estimate in low-noise conditions, they struggle in high-noise conditions.

We furthermore discuss these results in the Discussion (lines 397-406):

Relying more on predictions at the onset of a new trial, immediately after a change point, can be suboptimal. It was found that subjects tended to discard their predictions immediately following a change point in a sound-localization task where subjects were asked to predict the next stimulus after observing a series of stimuli (Krishnamurthy et al., 2017; Meijer et al., 2024). However, as noted in the study, participants were informed about the nature of the task, which could have influenced their responses.

In situations where the underlying task is not explicitly known, the strategy may be less clear. This is particularly true when changes occur in sensory noise rather than in the environment (that is, σ than μ). In such cases, a reliability-based weighting of sensory input and predictions might offer an advantage. Nevertheless, our model could be extended to include a change-point detection mechanism (see, e.g., Meijer et al., 2024), which could help reduce the observed discrepancy immediately after a change point.

In addition, I have a some minor suggestions.

- 1. In the intro, be clear that you're modeling a change detection task. The stairs example doesn't seem to fall into this category. And, given the timescales, this seems relevant to cognitive tasks; if you framed it as a cognitive task from the beginning, it would make a lot more sense. At least to me.**

Thank you for your idea. We have revised the example (lines 31-44):

For instance, when walking down an unfamiliar staircase in a well-lit basement, your brain may rely almost entirely on feedforward (bottom-up) sensory input (Fig. 1A, left), gradually forming a model of the step sizes. As the step sizes become more predictable, the brain can increasingly rely on this model. However, if the step sizes suddenly change, it will need to revert to the sensory input for guidance. Later, when walking down the same staircase and the lights suddenly turn off, your brain may rely entirely on feedback (top-down) signals derived from the staircase model you previously formed (Fig. 1A, middle), as the sensory information becomes too noisy to trust. But how do neural networks switch between a feedforward-dominated and a feedback-dominated processing mode in an ever-changing environment? For instance, if you hike down an unexplored mountain in very foggy conditions, your brain receives unreliable visual information. In addition, it can only draw on a shaky prediction about what to expect (Fig. 1A, right).

- 2. I would strongly suggest having, somewhere in the paper a succinct description of the model, as given above, along with the parameters used in all the simulations. In particular, it was hard to go from the Table 1 to the time constants, which are critical for understanding what's going on. I personally would put this in the main text (you have already started with the Box), but I'll leave that up to you.**

Thank you for this suggestion. We have clarified in the main text (Methods) that the effective time constant for the memory neurons are different between the lower and the higher subnetwork (lines 558 - 561)

Please note that although the time constants for the lower and higher M neurons are identical, their effective time constants differ due to variations in the weights connecting the PE neurons with the M neurons (the effective time constant of the higher subnetwork is between 4 and 64 times larger than that of the lower subnetwork, see 'Connectivity').

and in the SI Methods (lines 79 - 82):

Please note that although the time constants for the lower and higher M neurons are identical, their effective time constants differ due to differences in the weights connecting the PE neurons with the M neurons. (the effective time constant of the higher subnetwork is between 4 and 64 times greater than that of the lower subnetwork, see A.2.2).

Furthermore, we have refined the description of the model in the SI Methods, ensuring it now includes all relevant information and parameters. We have also ensured that references to the sections containing the parameters are provided immediately after each parameter is mentioned for the first time (see, for example, section A.1).

- 3. What are the dark lines in Figs. 2D and E? I don't believe they were described in the figure caption. But I might have missed it.**

Thank you for bringing this to our attention. We have now added the descriptions to the captions.

References

- Kamesh Krishnamurthy, Matthew R Nassar, Shilpa Sarode, and Joshua I Gold. Arousal-related adjustments of perceptual biases optimize perception in dynamic environments. *Nature human behaviour*, 1(6):0107, 2017.
- David Meijer, Roberto Barumerli, and Robert Baumgartner. How relevant is the prior? bayesian causal inference for dynamic perception in volatile environments. *bioRxiv*, pages 2024–10, 2024.

Reviewer #3

In this revision, the main update is Supplementary Fig 10 where the authors performed additional simulations comparing alternative weighted outputs. These additional results show that different strategies of weighing can result in distinct reconstruction results under different noise conditions. This is a nice addition to the manuscript. The authors also mention that the manuscript is novel as it adds a hierarchical extension and introduces M and V neurons to the previously proposed circuits. I am still not entirely convinced that these are sufficiently significant extensions of the models proposed in the authors' previous works. I do not think further simulations or modifications to the model are needed for this manuscript at this point, as my concerns are more on the general impact and novelty, which of course, is just my opinion. Nevertheless, I agree that the manuscript explores interesting additional mechanisms and simulations that make the circuit model more holistic, which is a valuable contribution.

I briefly looked at the Github repository and it includes a README file with instructions and code files. I have not installed and run the code.

We thank the reviewer for their thoughtful feedback and for recognizing the value of our additional simulations. We believe our work represents a significant advancement by unifying key mechanisms and demonstrating their functional implications. We are also pleased that the reviewer finds no need for further simulations or modifications at this stage.

Reviewer #4

I'm happy with the changes!

We appreciate the reviewer's feedback and are pleased that the revisions meet their expectations. We also want to thank them for their time and valuable input!